Corrected: Author correction

# WWP2 regulates pathological cardiac fibrosis by modulating SMAD2 signaling

Huimei Chen[1,22], Aida Moreno-Moral [1,22], Francesco Pesce[2], Nithya Devapragash[1], Massimiliano Mancini[3], Ee Ling Heng[4], Maxime Rotival[5], Prashant K. Srivastava[6], Nathan Harmston[1], Kirill Shkura[6], Owen J.L. Rackham[1], Wei-Ping Yu[7,8], Xi-Ming Sun[9], Nicole Gui Zhen Tee[10], Elisabeth Li Sa Tan[1], Paul J.R. Barton [4,11], Leanne E. Felkin[4,11], Enrique Lara-Pezzi[12], Gianni Angelini[4,13], Cristina Beltrami[4], Michal Pravenec[14], Sebastian Schafer [1,10], Leonardo Bottolo [15,16,17], Norbert Hubner[18,19,20,21], Costanza Emanueli[4,11], Stuart A. Cook [1,9,10] & Enrico Petretto [1,9]

Cardiac fibrosis is a final common pathology in inherited and acquired heart diseases that causes cardiac electrical and pump failure. Here, we use systems genetics to identify a pro-fibrotic gene network in the diseased heart and show that this network is regulated by the E3 ubiquitin ligase *WWP2*, specifically by the *WWP2-N* terminal isoform. Importantly, the *WWP2*-regulated pro-fibrotic gene network is conserved across different cardiac diseases characterized by fibrosis: human and murine dilated cardiomyopathy and repaired tetralogy of Fallot. Transgenic mice lacking the N-terminal region of the WWP2 protein show improved cardiac function and reduced myocardial fibrosis in response to pressure overload or myocardial infarction. In primary cardiac fibroblasts, WWP2 positively regulates the expression of pro-fibrotic markers and extracellular matrix genes. TGFβ1 stimulation promotes nuclear translocation of the WWP2 isoforms containing the N-terminal region and their interaction with SMAD2. WWP2 mediates the TGFβ1-induced nucleocytoplasmic shuttling and transcriptional activity of SMAD2.

[1] Programme in Cardiovascular and Metabolic Disorders, Duke-NUS Medical School, Singapore 169857, Republic of Singapore. [2] Department of Emergency and Organ Transplantation (DETO), University of Bari, 70124 Bari, Italy. [3] SOC di Anatomia Patologica, Ospedale San Giovanni di Dio, 50123 Florence, Italy. [4] National Heart and Lung Institute, Imperial College London, London SW7 2AZ, UK. [5] Unit of Human Evolutionary Genetics, Institute Pasteur, 75015 Paris, France. [6] Division of Brain Sciences, Imperial College Faculty of Medicine, London W12 0NN, UK. [7] Animal Gene Editing Laboratory, BRC, A*STAR20 Biopolis Way, Singapore 138668, Republic of Singapore. [8] Institute of Molecular and Cell Biology, A*STAR, 61 Biopolis Drive, Singapore 138673, Republic of Singapore. [9] MRC London Institute of Medical Sciences (LMC), Imperial College, London W12 0NN, UK. [10] National Heart Centre Singapore, Singapore 169609, Republic of Singapore. [11] Cardiovascular Research Centre, Royal Brompton and Harefield NHS Trust, London SW3 6NP, UK. [12] Centro Nacional de Investigaciones Cardiovasculares – CNIC, 28029 Madrid, Spain. [13] Bristol Heart Institute, Bristol Medical School, University of Bristol, Bristol BS2 89HW, UK. [14] Institute of Physiology, Czech Academy of Sciences, 142 00 Praha 4, Czech Republic. [15] Department of Medical Genetics, University of Cambridge, Cambridge CB2 0QQ, UK. [16] The Alan Turing Institute, London NW1 2DB, UK. [17] MRC Biostatistics Unit, University of Cambridge, Cambridge CB2 0SR, UK. [18] Cardiovascular and Metabolic Sciences, Max Delbrück Center for Molecular Medicine in the Helmholtz Association (MDC), 13125 Berlin, Germany. [19] DZHK (German Centre for Cardiovascular Research), Partner Site Berlin, 13347 Berlin, Germany. [20] Charité-Universitätsmedizin, 10117 Berlin, Germany. [21] Berlin Institute of Health (BIH), 10178 Berlin, Germany. [22] These authors contributed equally: Huimei Chen, Aida Moreno-Moral. Correspondence and requests for materials should be addressed to E.P. (email: enrico.petretto@duke-nus.edu.sg)

Pathological cardiac fibrosis is a process characterized by an excessive deposition of extracellular matrix (ECM) which in turn can lead to the development of cardiac dysfunction, arrhythmias, and heart failure (HF)[1]. Prolonged and extensive cardiac fibrosis is associated with poor clinical outcome and is an independent predictor of sudden cardiac death and overall mortality[2]. Cardiac fibrosis can have several and sometimes concurring triggers, including systemic hypertension, diabetes, and native cardiac ischemic insults causing ischemic cardiomyopathies and other forms of dilated cardiomyopathy (DCM). Despite diverse etiologies (e.g., ischemic vs non-ischemic origin), different heart conditions such as DCM and hypertrophic cardiomyopathy develop myocardial fibrosis and may share common pathogenic pathways that lead to the development of HF[3,4]. The fibrotic remodeling of the myocardium is also a feature of congenital heart diseases and their surgical repair, such as in repaired tetralogy of Fallot (rTOF)[5] or in other conditions causing progressive right ventricular dilation and failure[6]. Therefore, there is a growing body of evidences supporting cardiac fibrosis as a pathophysiological pathway common to different heart diseases[7].

The cardiac fibroblast is a key effector cell in cardiac fibrosis, responsible for homeostasis of the ECM in the heart. Upon triggering of the fibrogenic response, activated cardiac fibroblasts ultimately differentiate into myofibroblasts, increase the synthesis of ACTA2 and other ECM components, and reduce their proliferation rate[8]. At the molecular level, the development of tissue fibrosis comprises a complex signaling cascade, for which many regulators have been proposed such as angiotensin II (AngII), connective tissue growth factor (CTGF), bone morphogenetic protein (BMP), WNT, and cytokines such as interleukin 11 (IL-11)[9–12], and the transforming growth factor beta (TGFβ) superfamily[13]. In particular, TGFβ1 binds directly to receptors for signal transduction via downstream effector proteins known as SMADs, which leads to the transcription of ECM proteins[14].

Therapeutic targeting of cardiac fibrosis has been proposed for alleviating the progression of cardiovascular diseases and improving cardiac function[15]. Systemic and localized delivery of drugs modulating TGFβ, endothelin-1, AngII, CCN2, and PDGF are examples of anti-fibrotic therapies[16]. Moreover, targeting multiple pro-fibrotic pathways may provide an additional therapeutic approach to control cardiac fibrosis[17]. Since the physiological fibrosis process is required for normal wound healing and tissue repair, the identification of targets that regulate specifically cardiac fibrosis under pathological conditions is important to develop new therapies to improve clinical outcomes for patients with heart disease.

Here, using systems genetics[18] in the fibrotic heart we identify WWP2, a E3 ubiquitin protein ligase, as a positive genetic regulator of a transcriptional ECM gene network, which is associated with pathological cardiac fibrosis. We demonstrate a previously unappreciated role for WWP2 in the pathophysiological cardiovascular system, and elucidate how this gene regulates SMAD function and the downstream fibrogenic response in the diseased heart. These findings suggest WWP2 as a potential target for treating heart diseases.

## Results

**Coordinated regulation of ECM genes in the diseased heart.** We set out to identify transcriptional programs conserved across species and associated to cardiac fibrosis. We first used a panel of 30 rat recombinant inbred (RI) strains[19], which allows integrative analyses of cardiac gene expression with quantitative pathophysiological traits (e.g., cardiac fibrosis), and genome-wide genetic data[20–22]. This RI strains panel is an established model of

cardiovascular traits, including cardiac hypertrophy[20], blood pressure (BP)[23], and heart remodeling[21,24]. We performed gene co-expression network analysis in the rat RI strains left ventricle (LV) transcriptome using RNA-sequencing (RNA-seq) data. This identified 41 distinct gene co-expression networks (Fig. 1a, Supplementary Data 1a, b). We then tested the association of these gene co-expression networks with quantitative histopathologic measurements of interstitial and perivascular fibrosis in the rat heart (Supplementary Fig. 1). We identified five gene co-expression networks associated with both interstitial and perivascular cardiac fibrosis in the rat (adjusted $P < 0.05$, permutation test in Gene Set Enrichment Analysis [GSEA][25]) (Fig. 1b). To uncover transcriptional programs relevant to the fibrogenic processes in human heart disease, we performed a separate gene network analysis using LV RNA-seq data generated from a cohort of patients with DCM ($n = 126$) and a cohort of control heart samples ($n = 92$ organ donors whose hearts were explanted for transplantation[26]). We inferred gene co-expression networks in the human DCM LV transcriptome, which resulted in 48 distinct networks (Supplementary Data 2a, b). Then, we assessed which of these human DCM gene networks was conserved in the rat LV. This identified 14 human DCM networks as having some degree of conservation with the rat networks (adjusted $P < 0.05$, Fisher's exact test [FET]) (Fig. 1c). To obtain additional evidences supporting a role for these networks in human heart disease, we formally tested which gene co-expression pattern was present only in the LV from human DCM patients and not in LV from controls, i.e., we tested for differential co-expression between DCM and control hearts (see Methods). The differential co-expression paradigm assumes that the disease state is linked to perturbations of the structure of the regulatory network itself, and might reflect the dysregulation of the underlying transcription factors (TFs) in disease[27]. Here, we found that eight human gene networks were both conserved across species and differentially co-expressed between human DCM and controls (adjusted $P < 0.05$, permutation test). Despite the fact that several networks were significantly conserved between rat and human DCM heart (see Extended Analyses in Supplementary Note 1), only one human network (*Hs M47*), containing 683 genes, was (1) significantly conserved in the rat (sharing 72 genes with the rat network *M1*, adjusted $P = 8.4 \times 10^{-45}$ in FET, Fig. 1d), (2) associated with both interstitial and perivascular fibrosis in the rat heart (Fig. 1b), and (3) differentially co-expressed in human DCM heart (Fig. 1c, right panel).

The *Hs M47* was functionally relevant for ECM regulation, as it was significantly enriched for genes belonging to the specific biological pathways and processes: "ECM-receptor interaction", "TGFβ signaling pathway", and "focal adhesion" (Fig. 1e). Henceforth, we will refer to *Hs M47* as human ECM-network (or hECM-network). This hECM-network contains 237 strongly co-expressed genes annotated to encode for extracellular ECM region proteins (Supplementary Data 3b), and among these we highlight: 21 genes encoding for collagens; focal adhesion molecules such as *ITGB5*, *COMP*, *MAPK10*, and *THBS4*; several extracellular genes involved in TGFβ-signaling (e.g., *DCN*, *CHRD*, *TGFβ3*); three members of the BMP family (*BMP4*, *BMP6*, and *BMP8B*), and other important matricellular proteins, such as *CTGF* and *PDGFD*, which contribute to the fibrogenic response[11,28].

We further investigated whether the hECM network was specific to the ECM remodeling processes undergoing in DCM and/or in LV tissue. To this aim, we considered a separate heart condition, rTOF, which has a very different etiology from DCM but is characterized by the presence of cardiac dysfunction and diffuse and pathologic myocardial fibrosis of both the right ventricle (RV) and LV[29]. We analyzed RNA-seq data from RV in

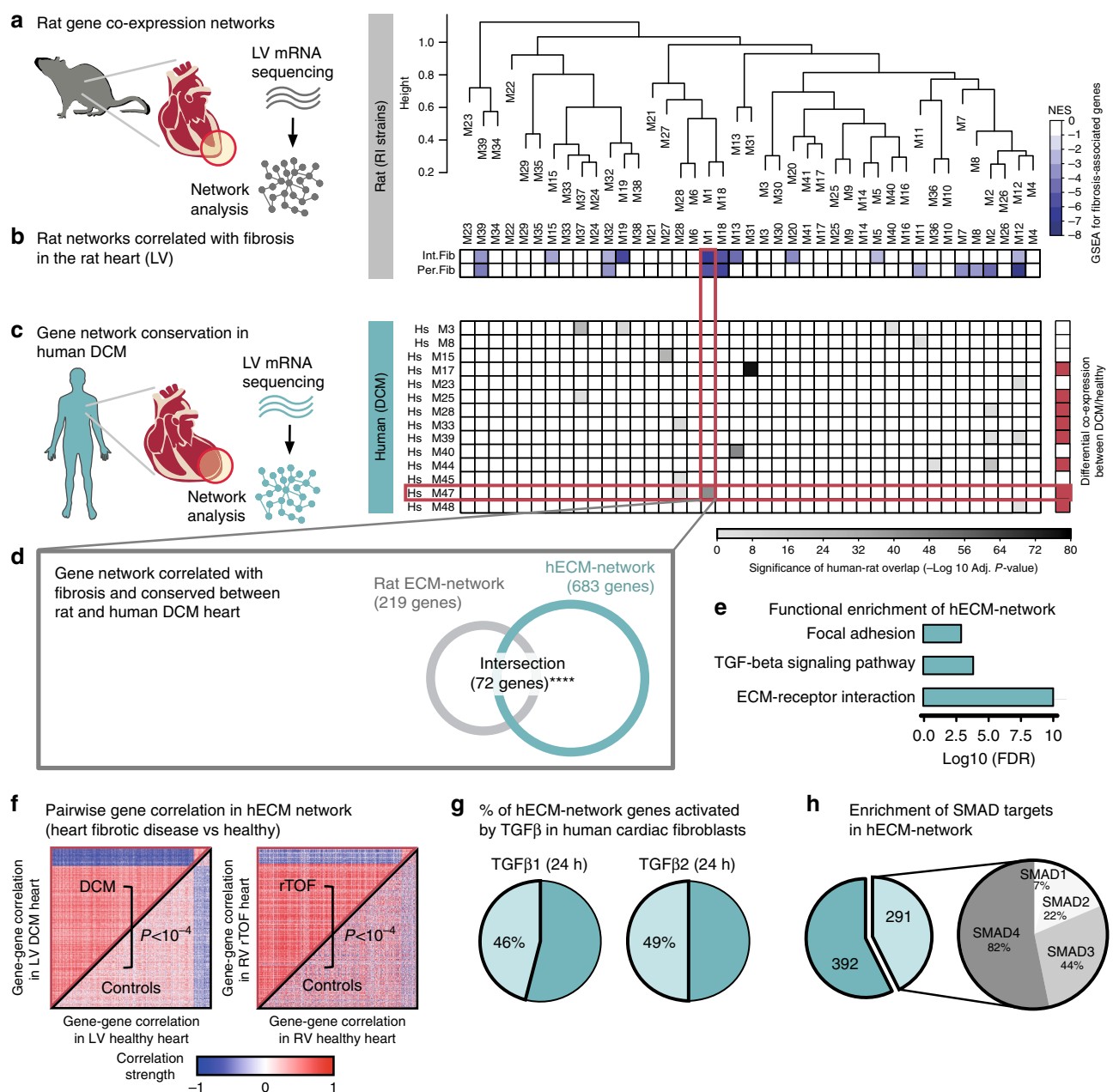

**Fig. 1** Co-expression networks analysis in rats and humans identifies a disease-associated pro-fibrotic ECM transcriptional program. **a** Hierarchical clustering of the first principal component of the rat co-expression networks (M1 to M39) inferred in the LV of the RI rat strains. Each branch denotes a distinct rat network with a different number of genes (full list in Supplementary Data 1a, b). **b** Level plot displaying the GSEA results with the significance of the association of each network to interstitial and perivascular fibrosis. This level plot is colored by GSEA NES, representing the strength of enrichment normalized by gene set size and white color denoting non-significant enrichment. **c** Each row represents a human co-expression network inferred in the LV of DCM patients. Only DCM networks that are significantly intersecting (adjusted $P < 0.05$, FET) to any rat network are shown (full list of networks in Supplementary Data 2a, b). The right panel highlights in red the human DCM modules that are significantly differentially co-expressed between DCM and control LV samples (red denotes adjusted $P < 0.05$, permutation test). **d** Shortlisted rat and human networks: rat network M1 (rat ECM-network) and human network Hs M47 (hECM-network). These two networks have a significant overlap and share 72 genes (adjusted $P = 8.39 \times 10^{-45}$, FET). **e** Functional enrichment of the hECM-network shows significantly overrepresented terms (FDR < 0.05). **f** Heatmap of pairwise gene-gene expression correlation estimates of the 683 genes included in the hECM-network in: left, LV cohort (DCM patients, $n = 126$ and controls, $n = 92$) and right, RV cohort (rTOF patients, $n = 27$ and controls, $n = 11$). In both cases, patients are displayed in the upper-left triangular matrix whereas controls are displayed in bottom-right triangular matrix. The $P$ is from the differential co-expression test (see Methods for details). **g** Percentage of genes in the hECM-network with significant differential expression (FDR < 0.05) in human ventricular cardiac (myo)fibroblasts after 24 h induction with either TGFβ1 or TGFβ2. **h** Breakdown of SMAD targets enrichment in hECM-network. The network contains 683 genes; 43% of these genes are targets of any SMAD (i.e., considering SMAD1/2/3/4; full list of SMADs targets in Supplementary Data 3d)

a cohort of adult patients with rTOF ($n = 27$) who underwent surgery for pulmonary valve replacement and age-matched control donor samples ($n = 11$), full details in Supplementary Table 1. The hECM gene co-expression network found in DCM LV was significantly conserved in rTOF RV ($P < 1 \times 10^{-5}$, permutation test) but not in control RV ($P \sim 1$, permutation test). The gene co-expression pattern of the hECM-network in rTOF RV mirrored the pattern observed in DCM LV (i.e., strongest gene-gene co-expression in disease), and this was significantly different between rTOF patients and controls (Fig. 1f). The consistent differential gene co-expression in DCM and rTOF heart thus suggests that the hECM network is capturing common ECM remodeling processes taking place across a wider range of human cardiac fibrotic diseases, irrespective of the specific disease etiology and the heart tissue, i.e., LV and RV. In addition, using longitudinal cardiac transcriptome data from a mouse genetic model of HF[30], we show that this hECM network is enriched for genes upregulated in DCM that progresses to HF. Compared with control heart, 54% of the hECM-network genes are upregulated in DCM ($P = 5.4 \times 10^{-59}$, FET) and 53% are upregulated in HF ($P = 2.4 \times 10^{-49}$, FET). These results in human and mouse DCM/HF suggest that the identified hECM-network recapitulates the maladaptive fibrotic remodeling that promotes HF and adverse cardiovascular outcomes.

Since the hECM network was enriched for genes involved in the TGFβ signaling (Fig. 1e), we looked at whether the hECM network was downstream of TGFβ-receptor activation. We analyzed RNA-seq data generated from primary cultures of human atrial cardiac (myo)fibroblasts exposed to TGFβ1 (24 h), TGFβ2 (24 h) or control media. TGFβ1 and TGFβ2 induced the differential expression of 46% and 49% of the genes in the hECM-network: 315 and 335 genes, respectively (Fig. 1g, Supplementary Data 3c). Given the key role of SMAD TFs downstream of TGFβ-receptors activation[31], we tested whether the hECM-network was enriched for SMAD target genes. Using published chromatin immunoprecipitation-sequencing data[32], we found that the hECM network was significantly enriched for SMAD-regulated genes ($n = 291$, 43% of the network genes), especially for SMAD2, SMAD3, and SMAD4 (Fig. 1h, Supplementary Data 3d). Together, these analyses revealed a cross-species conserved gene network, which might recapitulate the pathological ECM remodeling undergoing downstream of TGFβ/SMAD signaling activation in the diseased heart.

**WWP2 regulates the pro-fibrotic gene network in heart**. Gene-gene co-expression suggests coordinated gene regulation by one or more master regulator gene. To identify master regulators of the hECM-network, we used advanced Bayesian genetic mapping approaches[33] to pinpoint genomic loci regulating the whole ECM-network. In this, we consider the expression of the rat (or human) genes in the ECM-network as a multivariate quantitative trait, and then we test if the joint expression levels of the network genes are associated with genome-wide genetic variants (i.e., single-nucleotide polymorphisms, SNPs). This Bayesian expression quantitative trait locus (eQTL) mapping (or network-eQTL mapping) method was developed by our group to discover *trans*-acting master genetic regulators of networks in disease, including type 1 diabetes[34], epilepsy[35] and inflammatory disease[36].

Here, we first mapped the ECM-network in the rat, and identified a single locus on rat chromosome 19 regulating 219 genes of the network (median Bayes factor [BF] = 181.7, where the BF represents the strength of genetic regulation versus no genetic control of the network) (Fig. 2a). Then, we investigated whether the regulatory locus for the rat ECM-network was

conserved in humans. In the DCM patient cohort where the network genes are strongly co-regulated (Fig. 1f), we tested whether the hECM-network genes were jointly mapping to the human locus syntenic to the regulatory locus found in the rat (the human syntenic locus is located on chromosome 16, Fig. 2b). Network-eQTL mapping in human DCM detected a single regulatory SNP (rs9936589) located within an intron of the *WWP2* gene, an E3 ubiquitin ligase. This SNP was strongly associated with the expression of the hECM-network in DCM heart (Fig. 2c) (median BF = 2004 for the 683 hECM-network genes). This network-eQTL was not detectable in the heart from control organ donors (Supplementary Data 3a), suggesting that the genetic regulation of the hECM network is present (or is detectable) only in diseased heart. This regulatory SNP (rs9936589) and *WWP2* have not been previously associated with any fibrotic or heart disease.

These systems genetics analyses revealed a coordinated pro-fibrotic ECM transcriptional program in the diseased heart, which is regulated by a genetic variant within the *WWP2* locus (Fig. 2a–c). Therefore, we investigated whether the *WWP2* gene was a potential regulator of the hECM network in human fibrotic heart disease. We correlated the expression levels of *WWP2* to the expression levels of the 683 hECM-network genes in LV and RV fibrotic hearts (i.e., in DCM and rTOF), and control hearts separately. In both LV (DCM) and RV (rTOF) fibrotic hearts, we observed a positive and significant shift in the distribution of the correlations between *WWP2* and the expression of the hECM-network genes (Fig. 2d, e), suggesting a positive association between *WWP2* cardiac expression and the hECM-network genes in disease. However, *WWP2* cardiac expression was only moderately increased in heart disease: human rTOF vs control, fold change (FC) = 1.23; human DCM vs control[26], FC = 1.02; mouse HF vs WT control[30], FC = 1.86. We also identified a core set of genes in the hECM network that were positively and consistently correlated with *WWP2* expression in the heart (false discovery rate [FDR] < 1%), irrespective of heart tissue of origin (i.e., LV or RV) or disease (i.e., rTOF or DCM) (Fig. 2f, Supplementary Data 3e). This core gene set comprises known regulators of the pathological ECM remodeling, including matrix metalloproteinases (e.g., *MMP14*, *MMP2*)[37] and their tissue inhibitors (e.g., *TIMP2*)[38], several collagens (*COL1A1*, *COL1A2*, *COL5A1*, *COL6A2*, *COL8A2*, and *COL14A1*) and their binding partners (e.g., *TGFβI*)[39], microfibrillar-associated proteins, and pro-fibrotic cytokines (e.g., *TGFβ3*)[30]. In summary, these findings show that increased expression of *WWP2* is associated with an elevated pro-fibrotic gene expression program in the diseased heart.

Analysis of the cardiac expression levels of hECM-network genes stratified by the genotypes of the regulatory SNP rs9936589 showed increased expression associated with the *TT* genotype (Fig. 2g). To investigate whether the effect of regulatory SNP on the hECM network was mediated by *WWP2*, we tested whether *WWP2* cardiac expression was similarly regulated by the same SNP. Three main *WWP2* gene isoforms have been characterized containing different protein domains: full-length isoform (*WWP2*-FL, covering the entire gene), N-terminal isoform (*WWP2*-N, containing the 5′ end of the protein) and C-terminal isoform (*WWP2*-C, containing the 3′ end of the protein)[40] (Fig. 2h). All three isoforms are expressed in the heart of DCM patients (Supplementary Data 3f). Isoform-specific *cis*-eQTL mapping for each *WWP2* isoform showed that only the *WWP2*-N was regulated by the SNP rs9936589 in DCM heart (Fig. 2i, right). Increased *WWP2*-N cardiac expression was associated with the *TT* genotype (Fig. 2i, left), matching the hECM-network regulation by SNP rs9936589 in DCM heart (Fig. 2g). This concordance of genetic regulation suggests that the

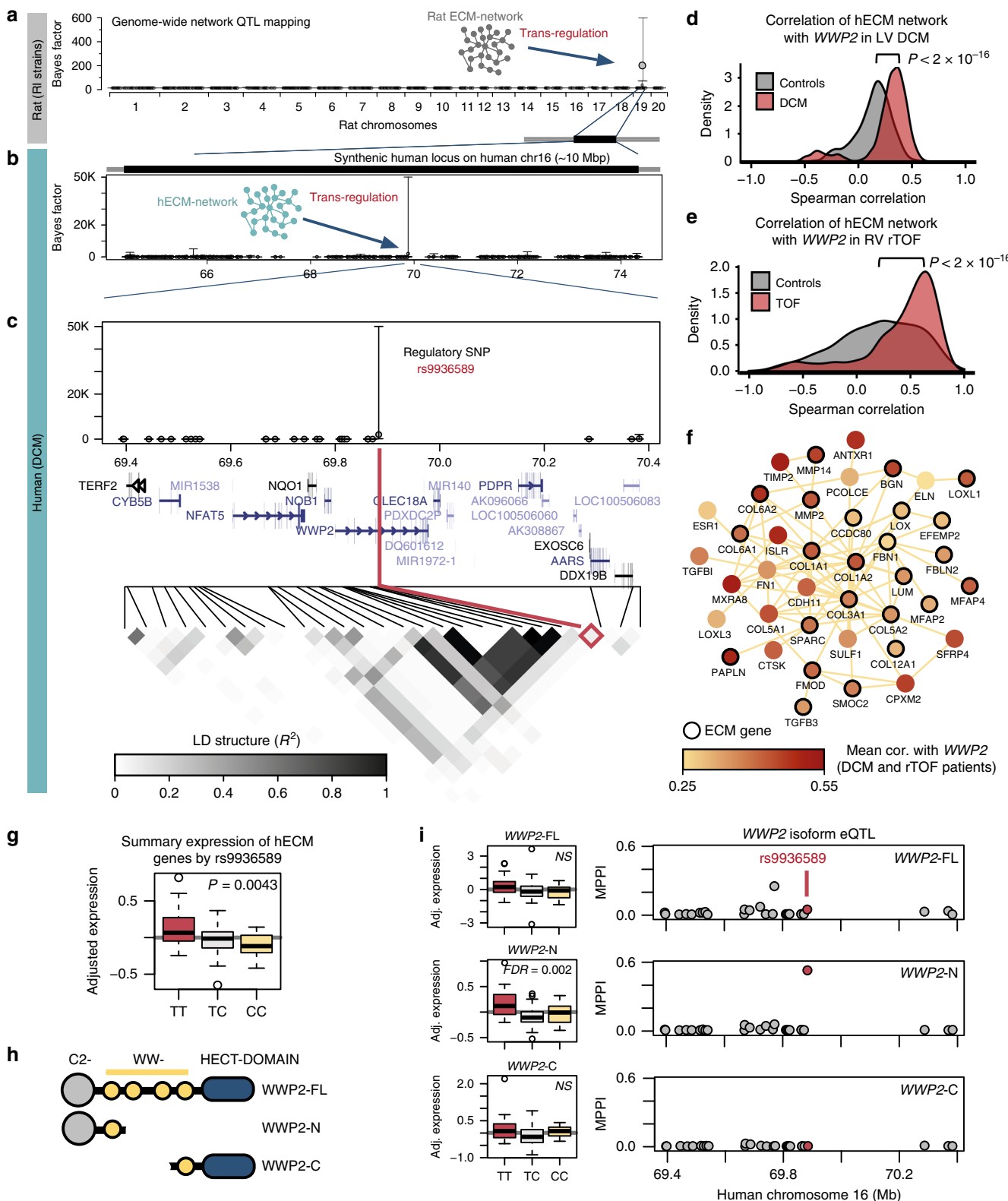

WWP2-N isoform, containing the N-terminal C2 and WW1 domains of WWP2, can be a positive regulator of the hECM network in the diseased heart.

**WWP2 regulates cardiac fibrosis in vivo**. To identify the pathophysiological processes regulated by the N-terminal region

of WWP2, we generated Wwp2 mutant mice (WWP2[Mut/Mut]) by introducing a 4-bp deletion in exon 2 of Wwp2, which would lead to disruption of WWP2-FL and WWP2-N isoforms (Fig. 3a, Supplementary Fig. 2a–c). Consistent with previously reported phenotype in Wwp2-null mice[41], the WWP2[Mut/Mut] mice generated in our study showed reduced body weight, abnormal craniofacial development and elongated teeth (Fig. 3b, c,

**Fig. 2** Network-eQTL mapping in rats and humans reveals WWP2 as a positive regulator of the ECM transcriptional program. **a** Genome-wide mapping of the rat ECM network ($n = 219$ genes) to the genome (1384 SNPs) identifies a *trans*-acting regulatory locus on rat chromosome 19q12 (spanning 6.22 Mb). **b** Genetic mapping of the hECM network in DCM patients ($n = 96$) to a ~10 Mbp region (475 human SNPs) on human chromosome 16 that is syntenic to the rat locus identified in panel **a**. **c** Zoom-in of panel **b** to the locus on human chromosome 16 with the SNP regulating the hECM network (rs9936589, in red), and the linkage disequilibrium (LD) blocks in the genotyped samples ($n = 187$). **d, e** Density plot showing the distribution of the Spearman's pairwise correlations ($\rho$) between WWP2 expression levels and each of the genes in the hECM network in DCM (**d**, red) or rTOF hearts (**e** in red) and control hearts (in gray). *P* for significance was calculated by Mann–Whitney *U* test. **f** Network graph showing the core set of hECM-network genes with strongest significant correlation (FDR < 0.01) to WWP2 in both DCM and rTOF patients. Full results in Supplementary Data 3e. Nodes represent genes and edges denote protein–protein interaction or co-expression in STRING database (https://string-db.org). Node color is mapped to the average correlation of each gene with WWP2 in patients. Genes that are annotated as "extracellular matrix" (ECM) are highlighted with thicker border. **g** Boxplot of the hECM-network genes (683 genes) summarized by median expression level of the hECM network in the DCM patients and broken down by genotype at the regulatory SNP rs9936589. **h** Graphical overview of the three main WWP2 protein isoforms. **i** Left, boxplot of WWP2-FL, WWP2-N, and WWP2-C gene isoform expression levels broken down by genotype at the regulatory SNP rs9936589 (following Kruskal–Wallis test for each WWP2 isoform, FDR corrects for the 18 isoforms detected in DCM heart). Right, Bayesian eQTL mapping, reported as marginal posterior probability of association (MPPI, *y*-axis) shows the WWP2-N terminal isoform at the WWP2 regulatory locus (1 Mb region centered around the WWP2 gene). NS not significant (FDR > 0.05)

---

Supplementary Fig. 2d). This 4-bp deletion in *Wwp2* resulted in ablation of WWP2 isoforms containing the Wwp2 N-terminal region (i.e., *Wwp2-FL* and *Wwp2-N* isoforms, detected at the mRNA level by isoform-specific primer pairs P1; Fig. 3a, d), and lack of WWP2-FL and WWP2-N proteins (Fig. 3e, Supplementary Fig. 2e). The cardiac expression of WWP2-C protein was not affected.

Our systems genetics analysis indicated that the *WWP2*-N isoform positively regulates a pro-fibrotic transcriptional network in diseased heart (Figs. 1 and 2). Therefore, we hypothesized that loss-of-function (LOF) of WWP2-N/FL protein isoforms might regulate the in vivo fibrogenic response. Upon chronic (4 weeks) AngII treatment, AngII infusion resulted in increased tissue fibrosis, ventricular remodeling, and worsened cardiac function (Supplementary Fig. 3a–d). We also detected increased WWP2 transcripts and proteins levels in LV tissue (Supplementary Fig. 3e–g). Compared with WT mice, WWP2$^{Mut/Mut}$ mice showed a significant improvement of AngII-induced cardiac fibrosis as shown by the lower percentage of tissues fibrosis (Fig. 3f), which was accompanied by an attenuation of cardiac hypertrophy (Fig. 3g, Supplementary Fig. 4). Cardiac function was also improved in WWP2$^{Mut/Mut}$ mice, which showed increased fractional shortening (FS%) and ejection fraction (EF%) as compared with AngII-treated WT mice (Fig. 3h).

Bulk RNA-seq analysis in AngII-treated WT ($n = 8$) and WWP2$^{Mut/Mut}$ mice ($n = 8$) hearts revealed that the mouse orthologs of the hECM-network genes detected in DCM heart have a different co-expression pattern with increased gene co-expression in WT compared with WWP2$^{Mut/Mut}$ ($P = 0.003$ by permutation test, Fig. 3i). This pattern of differential co-expression (WWP2$^{Mut/Mut}$ vs WT) was remarkably similar to the pattern observed in humans (i.e., between DCM or rTOF and controls, Fig. 1f), where the WWP2$^{Mut/Mut}$ mice showed a co-expression pattern similar to control heart. Consistent with human data (Fig. 1e, g), the hECM network was significantly enriched for differentially expressed (DE) genes between AngII-treated WT and WWP2$^{Mut/Mut}$ mice (Supplementary Data 4b). Among others, we detected "TGFβ signaling" and "extracellular matrix" as two of the major downregulated pathways in WWP2$^{Mut/Mut}$ mice upon AngII treatment (Fig. 3j, Supplementary Data 4a). WWP2$^{Mut/Mut}$ mice had reduced levels of fibroblast activation and ECM protein markers, as shown by α-smooth muscle actin (ACTA2), collagen1 (COL1A1), fibronectin extracellular domain A (FN-EDA), and periostin (POSTN) abundance in the heart after AngII treatment (Fig. 3k–m, Supplementary Fig. 5).

We used a second in vivo fibrosis disease model (myocardial infarction [MI] model, see Methods) and tested the potential protective effect of the WWP2-N/FL LOF on cardiac fibrosis and function post-MI. Histological analysis showed less post-MI fibrotic remodeling in the WWP2$^{Mut/Mut}$ hearts compared with WT (Fig. 3n). This was associated with reduced chamber dilation and greater preservation of contractile function in WWP2$^{Mut/Mut}$ mice compared with WT (Fig. 3o, p). Taken together, these data suggest that LOF of the *WWP2* isoforms containing N-terminal region reduced cardiac fibrosis and improved cardiac function following AngII treatment or MI. This is in keeping with a role for *WWP2* as a positive regulator of fibrosis in the diseased heart.

**WWP2 regulates the TGFβ1-induced fibrotic response In vitro.** To investigate the regulation of WWP2 in cardiac cells, we imaged WWP2-expressing cell(s) in heart sections by immunofluorescence. WWP2-positive cells did not show the morphology typical of a sarcomere-containing cardiomyocyte, and some WWP2-positive cells expressed fibroblast-specific protein 1 (FSP1) (Fig. 4a, Supplementary Fig. 6a). Cultured (myo)fibroblasts isolated from the LV of WT mice showed co-expression of WWP2 and FSP1 (Supplementary Fig. 6b). Single-cell RNA-seq analysis in the WT heart following AngII treatment provided additional evidence of WWP2 expression in (myo)fibroblasts, but also in endothelial and immune cells (Fig. 4b). By contrast, and consistent with our immunofluorescence data, WWP2 was not expressed in cardiomyocytes.

TGFβ1 stimulation of primary murine LV (myo)fibroblasts induced robust induction of *Wwp2* transcription at 72 h of treatment (Supplementary Fig. 7). We then investigated the impact of WWP2 LOF in the response to prolonged (72 h) TGFβ1 treatment in primary murine LV (myo)fibroblasts (Fig. 4c). In (myo)fibroblasts, TGFβ1 stimulation increased pro-fibrotic activity and ECM production (measured by ACTA2, COL1A1, and POSTN), but the TGFβ1-induced pro-fibrotic expressional changes at both the mRNA and protein levels were largely prevented in WWP2$^{Mut/Mut}$ (myo)fibroblasts (Fig. 4d, e). TGFβ1-stimulated WT (myo)fibroblasts presented a clear organization of ACTA2 into stress fibers, while WWP2$^{Mut/Mut}$-derived cells showed a diffuse expression of ACTA2 with rare incorporation into stress fibers (Fig. 4f). TGFβ1 mildly increased vimentin protein expression, which was reduced in WWP2$^{Mut/Mut}$ cells (Supplementary Fig. 8a, b). However, we did not detect differences in the mRNA level of vimentin and Transcription Factor 21 (*Tcf21*) (Fig. 4d, Supplementary Fig. 8c). Upon TGFβ1 treatment we also observed induction of TGFβ receptors (*Tgfbr1* and *Tgfbr2*) in cardiac (myo)fibroblasts from WWP2$^{Mut/Mut}$ mice (Supplementary Fig. 8d), suggesting a potential compensatory effect of WWP2 on TGFβ-signaling activation. In addition, WWP2$^{Mut/Mut}$ cardiac (myo)fibroblasts

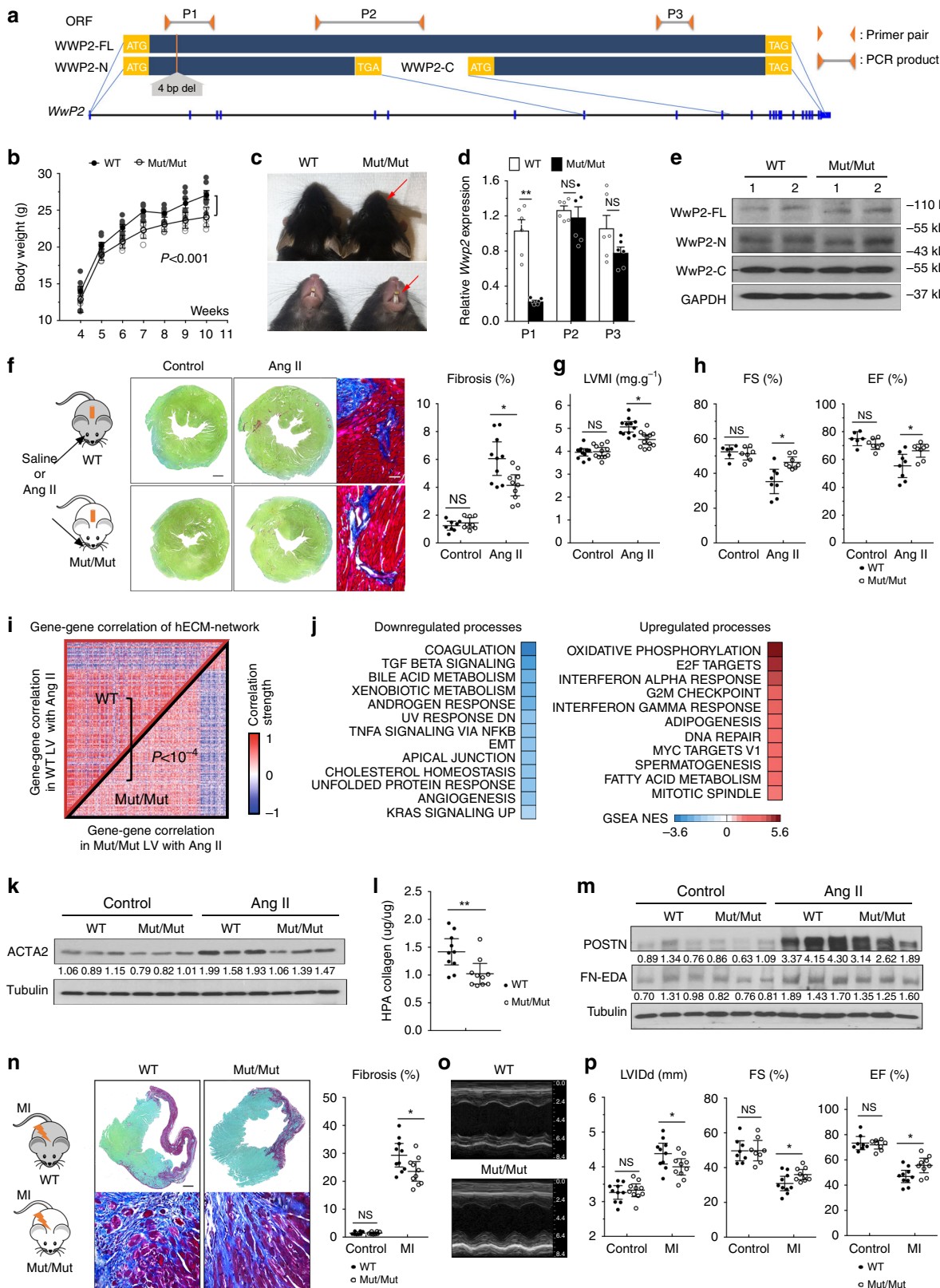

showed higher cell proliferation and migration compared to controls (Supplementary Fig. 9), which is consistent with reduced maturation in myofibroblasts[42].

Our analysis of *WWP2* in DCM heart suggested a role for the *WWP2* isoforms containing N-terminal region in fibrosis (Fig. 2i).

In line with this, murine cardiac (myo)fibroblasts responded to TGFβ1 treatment with increased protein levels of the WWP2-FL/N isoforms, but not of WWP2-C (Fig. 4g). Such induction of WWP2-FL/N isoforms was consistent with the findings in mouse heart following AngII infusion in vivo (Supplementary Fig. 3a–c).

**Fig. 3** WWP2$^{Mut/Mut}$ mice are protected from cardiac fibrosis progression. **a** Representation of the open reading frame (ORF) of *Wwp2*-FL, *Wwp2*-N, and *Wwp2*-C isoforms. The position of the 4 bp deletion introduced in the WWP2$^{Mut/Mut}$ mouse is shown alongside the position of the primer pairs (P1–P2–P3, orange triangles) used for the qPCR analysis. **b** Body weight change of male WT and WWP2$^{Mut/Mut}$ mice (age range: 4–11 weeks). Repeated-measures *t*-test ($n = 8$ for each group). **c** Presence of shortened snout (top, arrow) and overgrown mandibular incisor (below, arrow) in WWP2$^{Mut/Mut}$ mice. **d** *WWP2* transcript level in WT and WWP2$^{Mut/Mut}$ hearts ($n = 6$ for each group). P1-P2-P3 as in panel **a**. **e** Western blot (WB) showing lack of right molecular weight band for WWP2-FL (~110 kDa) and WWP2-N (~50 kDa) protein isoforms in WWP2$^{Mut/Mut}$ mice. **f** Left, schematic of AngII experiment. Middle, representative Sirius red and Masson's Trichrome staining of short-axis sections in LV. Scale bar: 0.5 mm. Right, Quantification of fibrosis area in transverse histological sections with Sirius red staining at the mid-ventricular level. **g** LV mass index (LVMI), **h** echocardiogram-based quantification of LV ejection fraction (EF%) and fractional shortening (FS%) after Saline (Control) and AngII infusion. **i** Heatmap of pairwise gene–gene expression correlation (red, positive; blue, negative) of the hECM-network genes in mouse LV. Gene–gene correlations in WT and WWP2$^{Mut/Mut}$ mice (both with AngII treatment) are shown in the upper and lower triangular matrix, respectively. **j** Top dysregulated biological processes in LV of WWP2$^{Mut/Mut}$ vs WT mice after 28 days AngII treatment (full list in Supplementary Data 4a). After AngII infusion in WT and WWP2$^{Mut/Mut}$ mouse heart, **k** representative WB showing ACTA2, (**l**) collagen content (HPA assay), and **m** representative WB showing Fibronectin extracellular domain A (FN-EDA, ~220 kDa) and Periostin (POSTN, ~94 kDa). **n** Left, schematic of myocardial infarction (MI) experiment. Middle, representative Sirius red (top) and Masson's Trichrome staining (below) of short-axis sections in LV from WT and WWP2$^{Mut/Mut}$ mice after MI. Scale bar: 0.5 mm. Right, fibrosis quantification with Sirius red staining in transverse histological heart sections at the infarct level. **o** Representative M-mode echocardiograms (middle LV long-axis) in WT and WWP2$^{Mut/Mut}$ mice after MI. **p** Cardiac echocardiogram-based analysis of LV inner diameter (LVIDd), LV ejection fraction (EF%), and fractional shortening (FS%) of WT and WWP2$^{Mut/Mut}$ mice after MI. Unless otherwise indicated, *P* values calculated by Mann–Whitney *U* test; *$P < 0.05$, **$P < 0.01$, NS not significant; data reported as mean ± SD

WWP2$^{Mut/Mut}$ mice lack WWP2-FL and WWP2-N protein isoforms (Fig. 3d, e, Supplementary Fig. 2e), and this is sufficient to alter the co-regulation of the hECM-network genes, reduce cardiac fibrosis in vivo (Fig. 3f–p), and decrease the maturation of myofibroblasts in vitro (Fig. 4c–f). These data combined suggest a primary role for *WWP2* gene isoforms containing N-terminal region of the protein in regulating a transcriptional program associated with cardiac fibrosis.

To confirm the differential effects of the *WWP2*-N-terminal region on cardiac (myo)fibroblast activity, we designed siRNA sequences to match either the 5′-terminal (siRNA-Wwp2-N′) or 3′-terminal (siRNA-Wwp2-C′) regions of the *Wwp2* mRNA, targeting different transcripts (Fig. 4h, Supplementary Fig. 10a). We successfully decreased the expression of WWP2-FL/N and WWP2-FL/C isoforms in cardiac (myo)fibroblasts, respectively (Fig. 4j, Supplementary Fig. 10b). Compared with scrambled control, both siRNAs mitigated the expression of ACTA2 in WT primary cardiac (myo)fibroblasts treated with TGFβ1 (Fig. 4i, j). In keeping with the WWP2$^{Mut/Mut}$ cardiac (myo)fibroblasts data (Supplementary Fig. 8d), after *WWP2* knockdown, the (myo)fibroblasts showed increased mRNA expression of *Tgfbr1* and *Tgfbr2* following TGFβ1 treatment (Fig. 4i). In human primary cardiac (myo)fibroblasts, siRNA experiments targeting the 5′-terminal of *WWP2* mRNA (siRNA-WWP2-N′) confirmed that *WWP2*-N/FL knockdown is able to reduce pro-fibrotic gene expression (Fig. 4k). We then performed a rescue experiment in mice, by re-introducing separately each of the two isoforms containing WWP2 N-terminal region (i.e., *WWP2*-FL and *WWP2*-N) in primary cardiac (myo)fibroblasts from WWP2$^{Mut/Mut}$ mice (Fig. 4l). Both *Wwp2*-FL and *Wwp2*-N individually increased the expression of pro-fibrotic genes in the WWP2$^{Mut/Mut}$ (myo)fibroblasts treated with TGFβ1 (Fig. 4m, n), supporting the role of these isoforms in regulating the fibrotic response.

**WWP2 regulates the nucleocytoplasmic shuttling of SMAD2.**
We investigated the mechanisms through which WWP2 regulates the fibrotic response downstream of TGFβ-signaling activation. WWP2 is present in both the cytoplasm and nuclei in untreated cardiac (myo)fibroblasts; however, upon TGFβ1 stimulation (16 h) we observed increased WWP2 expression localized predominantly to the nucleus (Fig. 5a). While the WWP2-FL and WWP2-N protein isoforms were increased in the nucleus after

TGFβ1 stimulation, the WWP2-C isoform remained in the cytoplasm (Fig. 5b, Supplementary Fig. 11a).

This TGFβ1-induced nuclear relocalization of WWP2 isoforms suggest that WWP2 may be involved in regulating gene expression, possibly targeting TFs for ubiquitination as shown for other E3 ubiquitin ligases[43]. We have showed that the pro-fibrotic transcriptional program (i.e., the hECM network) regulated by WWP2 is enriched for transcriptional targets of the TGFβ-signaling transducer SMAD TFs (Fig. 1h). In keeping with this, the genes downregulated by WWP2 in vivo following AngII treatment were enriched for "SMAD binding" (FDR = 0.003), "SMAD protein signal" (FDR = 0.006), and "transcriptional activity of SMAD2/3/4 heterotrimer" (FDR = 0.015) (Supplementary Data 4). We did not observe any difference at the mRNA level of SMAD2 in WWP2$^{Mut/Mut}$ cardiac (myo)fibroblasts or in cells overexpressing or silencing the WWP2-N/FL isoforms (Supplementary Fig. 11b, c), suggesting that WWP2 function on cardiac fibrosis could be exerted through SMAD2 interaction and ubiquitination at the protein level. When ectopically expressed in NIH-3T3 mouse embryonic fibroblast cells, we found that Flag-tagged WWP2-FL and WWP2-N, but not WWP2-C, co-immunoprecipitated with SMAD2 protein (Supplementary Fig. 11d). Mouse WT cardiac (myo)fibroblasts responded to TGFβ1 stimulation with SMAD2 binding to WWP2-FL and WWP2-N, but not to WWP2-C (Supplementary Fig. 11e). These data are suggestive of an endogenous physiological interaction between SMAD2 and WWP2-FL/N protein isoforms. Further analysis of inhibitory SMAD7, a preferred substrate for WWP2-FL and WWP2-C[40], showed co-immunoprecipitation with WWP2-FL and WWP2-C following TGFβ1 stimulation (Supplementary Fig. 11d), which we confirmed in primary (myo)fibroblasts (Supplementary Fig. 11g, h). We also found that WWP2 directly interacts with p-SMAD2 (Supplementary Fig. 11f). Notably, the levels of SMAD2 and p-SMAD2 proteins were similar in WT and WWP2$^{Mut/Mut}$ (myo) fibroblasts treated with TGFβ1 (Fig. 5c). This would be consistent with the monoubiquitination of SMAD2 by WWP2, resulting in a post-translational modification that does not affect SMAD2 protein levels. Ubiquitination assays followed by SMAD2 immunoprecipitation detected monoubiquitinated SMAD2 within 16 h of TGFβ1 stimulation (Supplementary Fig. 11i). We confirmed SMAD2 monoubiquitination by WWP2 in mouse primary cardiac (myo)fibroblasts, and show that this was reduced in WWP2$^{Mut/Mut}$ cells (Fig. 5d).

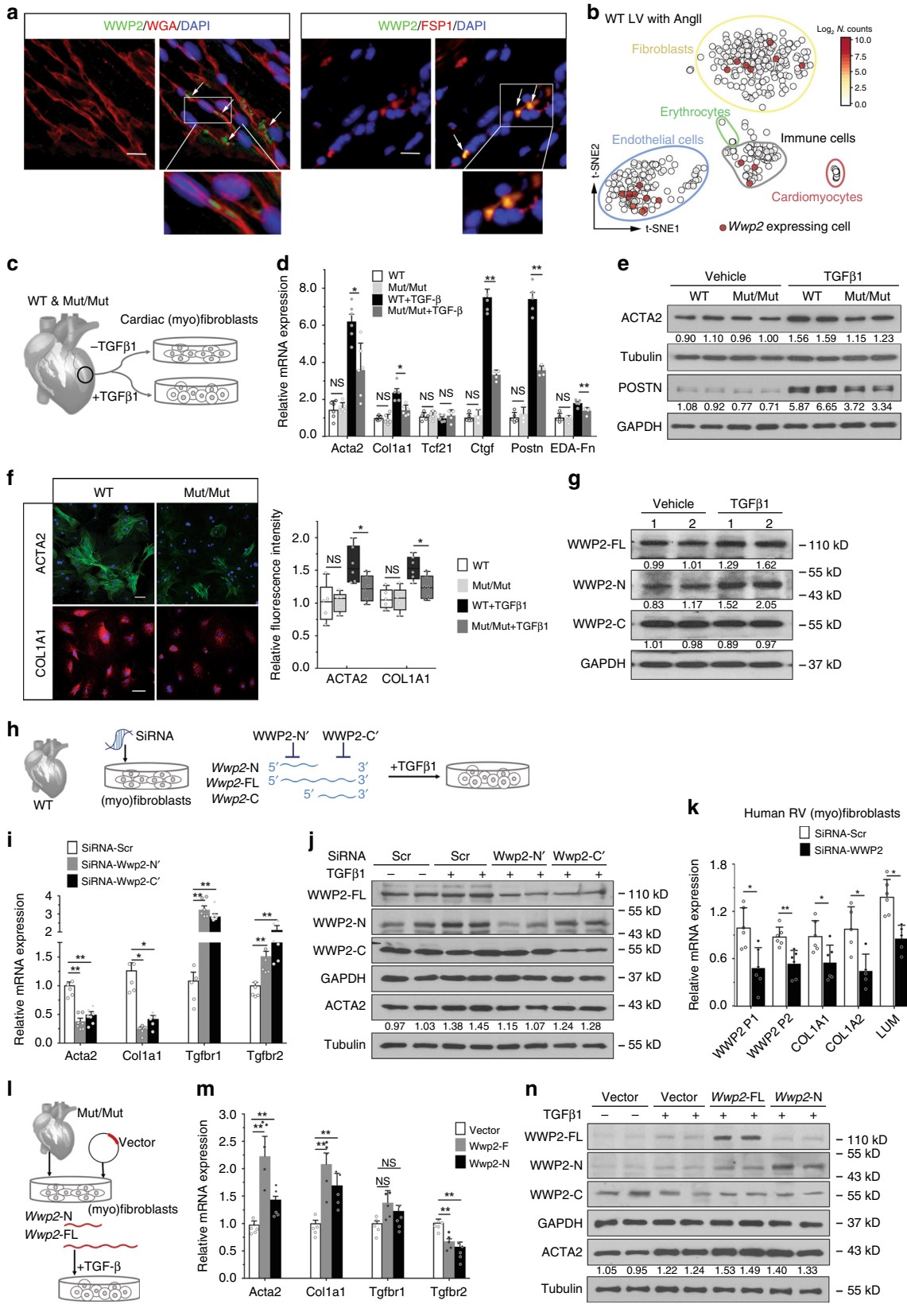

Monoubiquitination of SMADs by E3 ubiquitin ligases reportedly affects their transcriptional activity[44,45]. To test if WWP2 similarly affects SMAD2 transcriptional activity, we performed SMAD2-dependent luciferase reporter activity assay in primary cardiac (myo)fibroblasts from WT and WWP2[Mut/Mut] mice. TGFβ1-dependent SMAD2 reporter activity was signifi- cantly lower in WWP2[Mut/Mut] cardiac (myo)fibroblasts compared to WT cells (Fig. 5e). TGFβ-receptor activation promotes the

**Fig. 4** WWP2 regulates pro-fibrotic gene activity in primary cardiac fibroblasts. **a** Immunofluorescence images of LV-section staining from WT mice after AngII infusion, showing WWP2 expression (green, arrow) in non-myocytes (Left); WWP2 co-localization with FSP1-positive cells (red, arrow; Right). Scale bar: 40 μm. **b** t-SNE displaying LV single-cell RNA-seq data (508 cells detected) in WT mice after AngII infusion. Each dot corresponds to a single-cell, colored by $Wwp2$ expression level ($Log_2$ normalized counts). Different cell subpopulations detected in mouse heart are indicated. **c** Primary cardiac (myo) fibroblasts were taken from LV in WT and WWP2$^{Mut/Mut}$ mice and modeled in vitro by incubation with TGFβ1 (5 ng/ml, 72 h). **d** Relative mRNA expression (normalized to 18S) of ECM and pro-fibrotic marker genes ($Acta2$, $Col1a1$, $Tcf21$, $Ctgf$, $Postn$, and $EDA-Fn$) in primary cardiac (myo)fibroblasts ($n = 5–6$ for each group). **e** WB of ACTA2 and POSTN protein expression in WT and WWP2$^{Mut/Mut}$ cardiac (myo)fibroblasts. **f** Representative microscopy images (left) and quantification analysis (right) with immunostaining for ACTA2 and COL1A1 after TGFβ1 stimulation in WT and WWP2$^{Mut/Mut}$ cardiac (myo)fibroblasts (five biological replicates, nine images each; box-and-whisker plots). **g** Representative WB of WWP2-FL, WWP2-N and WWP2-C isoforms expression in cardiac (myo)fibroblasts after 72 h TGFβ1 stimulation. **h** WT cardiac (myo)fibroblasts were incubated with TGFβ1 (72 h) and siRNA pools (SiRNA-$Wwp2$-N′ and SiRNA-$Wwp2$-C′) against 5′- or 3′-region of $Wwp2$ mRNA. **i** WT (myo)fibroblasts, SiRNAs-$Wwp2$-N′/C′ attenuated TGFβ1-induced pro-fibrotic responses, as shown by qPCR analysis of $Acta2$, $Col1a1$, and TGFβ receptors ($n = 5–6$ for each group). **j** Representative WB showing the effect of siRNA-$Wwp2$-N′ and siRNA-$Wwp2$-C′ on each WWP2 isoform expression in WT (myo)fibroblasts and decrease of TGFβ1-induced ACTA2 protein level by siRNA-$Wwp2$-N′. **k** Cultured human RV primary cardiac (myo)fibroblasts: siRNA-mediated knockdown of $WWP2$ reduced the expression of pro-fibrotic genes ($COL1A1$, $COL1A2$, and $LUM$). P1 tags both WWP2-N and WWP2-FL, P2 tags WWP2-FL only ($n = 3$ for each group). **l** Cardiac (myo) fibroblasts from WWP2$^{Mut/Mut}$ were transfected with either $Wwp2$-FL or $Wwp2$-N plasmid expression (separately) and incubated with TGFβ1 (72 h). **m**, **n** After transfection of either $Wwp2$-FL or $Wwp2$-N, transcript levels of $Acta2$, $Col1a1$, and TGFβ receptors were assessed by qPCR, in WWP2$^{Mut/Mut}$ (myo)fibroblasts ($n = 5$ for each group) and ACTA2 protein levels. These fibrogenic markers were enhanced by either $Wwp2$-FL or $Wwp2$-N transfection after 72 h TGFβ1 treatment. $P$ values calculated by Mann–Whitney $U$ test; *$P < 0.05$, **$P < 0.01$, NS not significant; data reported as means ± SD

nuclear accumulation of SMAD2/3/4 and this process is not necessarily accompanied by SMAD degradation in the nucleus[46], as SMADs are exported out of the nucleus upon dephosphorylation and dissociation of the SMAD complexes[47]. We obtained nuclear and cytoplasmic fractions from cardiac (myo) fibroblasts, and show the nuclear accumulation of SMAD2 upon TGFβ1 stimulation (<16 h). Compared to WT cells, SMAD2 is more localized to the nucleus in WWP2$^{Mut/Mut}$ cells (Fig. 5f). SMAD4, which forms a heteromeric complex with SMAD2 after TGFβ1 activation, showed similar protein level and subcellular localization in WT and WWP2$^{Mut/Mut}$ (myo)fibroblasts. Thus, despite the fact the SMAD2 protein is more abundant in the nucleus of WWP2$^{Mut/Mut}$ (myo)fibroblasts, lack of WWP2-N/FL was associated with a reduced transcriptional activity of SMAD2 downstream of TGFβ-receptor activation (Fig. 5e).

Monoubiquitination is also important for the proper subcellular localization of SMADs, which in turn might regulate their transcriptional activity in the nucleus[44]. We used SB431542, a selective inhibitor of TGFβ superfamily type I activin receptor-like kinase (ALK) receptors[48], to study the nucleocytoplasmic shuttling of SMAD2 (ref. [49]). Using primary (myo)fibroblasts from WWP2$^{Mut/Mut}$ and WT mice, we observed a delay in the nuclear export of SMAD2 in WWP2$^{Mut/Mut}$ cells (Fig. 5g). Compared with WT cells, WWP2$^{Mut/Mut}$ cardiac (myo)fibroblasts maintained sizeable levels of p-SMAD2 at 3 h treatment with SB431542 (Supplementary Fig. 12). Taken together, these data show that WWP2 interacts with SMAD2, promoting its monoubiquitination, and modulates the nucleocytoplasmic shuttling and transcriptional activity of SMAD2 downstream of TGFβ-signaling activation (Fig. 5h).

## Discussion

The WW domain containing E3 ubiquitin protein ligase 2 ($WWP2$) gene, also known as atrophin-1-interacting protein 2 (AIP-2), is a multifunctional ubiquitin E3 ligase. It has been previously reported to regulate various processes, including palatogenesis[50], craniofacial development[41], TLR3-mediated innate immune and inflammatory responses[51], cell death and tumorigenesis[52], and tumor growth[53]. Common genetic variants within the $WWP2$ locus have been associated with plantar fascial disorders and osteoarthritis[54,55], supporting the heterogeneous functions associated with $WWP2$. Working in cancer cell lines,

Soond et al.[40] implicated WWP2 in oncogenic TGFβ-induced epithelial–mesenchymal transition (EMT), reporting the selective and TGFβ-dependent targeting of SMADs by specific ectopic WWP2 isoforms. The WWP2-N isoform was shown to enhance the activity of WWP2-FL, which in turn degrades SMAD2/3. Given the primary role of TGFβ-SMAD2/3 signaling in activating tissue-resident cardiac fibroblasts and the fibrotic response[56], the data by Soond et al.[40] might suggest that increased WWP2-N protein isoform expression with concurrent increased activity of WWP2-FL would lead to reduced fibrogenesis. However, any possible fibrotic or anti-fibrotic function of WWP2 has never been investigated, especially in cardiac pathophysiology.

Here, starting from the identification of a pro-fibrotic gene network conserved in rat and human heart disease characterized by diffuse myocardial remodeling and fibrosis, we identified $WWP2$ as a regulator of pathological cardiac fibrosis. The $WWP2$-reguated pro-fibrotic gene network was also conserved across different heart fibrotic diseases and cardiac tissues, and was upregulated in DCM that progresses to HF in mice[30]. The lack of regulation of the pro-fibrotic gene network by $WWP2$ in human control heart suggests that WWP2 exerts its regulatory role on cardiac fibrosis upon disease. Interestingly, the hECM network regulated by $WWP2$ in DCM heart is enriched for genes highly expressed in fibroblasts (see Supplementary Fig. 13, Supplementary Methods). This might suggest that the power to detect genetic regulation of the pro-fibrotic network in DCM was enhanced because the heart tissues underwent strong remodeling with concurrent changes in relative cellular composition (i.e., more fibroblasts). $WWP2$ mRNA expression was only marginally increased the fibrotic heart disease (less than 2 folds in either human DCM, rTOF, or mouse HF[30]). This might explain why $WWP2$ passed undetected to GWAS, eQTL mapping, and genetic screening studies of fibrotic diseases. Detailed systems genetics analyses in the DCM heart allowed us to hypothesize that increased expression of the WWP2 N-terminal isoform was associated with the activation of a pro-fibrotic gene program downstream of TGFβ/SMAD signaling activation (Figs. 1 and 2). We tested and corroborated this hypothesis in primary cardiac fibroblasts and in two preclinical models of fibrotic heart disease. Our results provide the first indication of a role for WWP2 in regulating pathophysiological processes in the heart. In detail, we have demonstrated that WWP2-N/FL LOF improved cardiac function and reduced myocardial fibrosis in vivo (Fig. 3). Upon

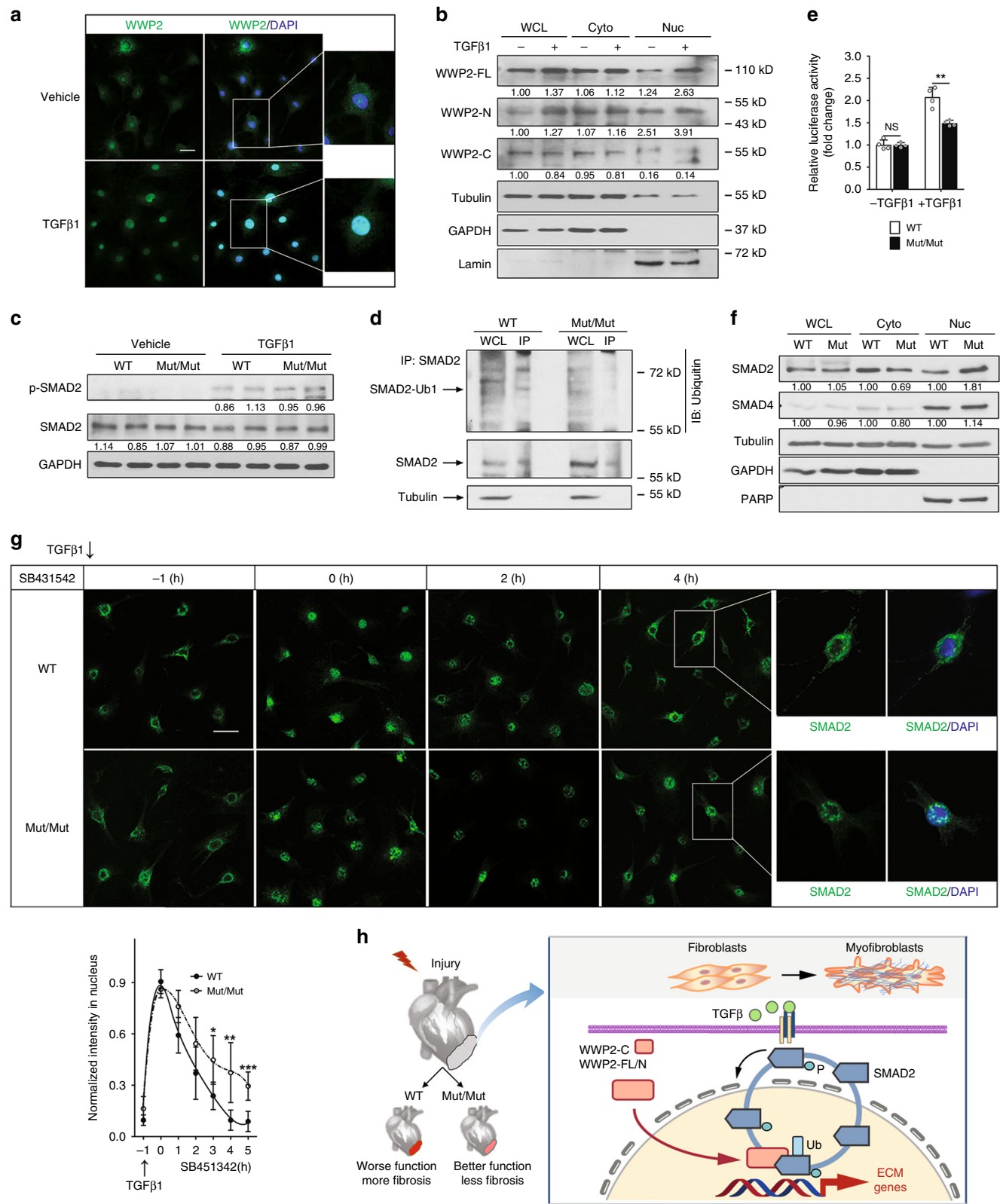

AngII treatment, a typical promoter of cardiac fibrosis, the WWP2$^{Mut/Mut}$ mouse showed increased oxidative phosphorylation, and exacerbated cell cycle and interferon alpha and gamma response in the heart. This suggests that depletion of the N-terminal part of WWP2 might have a wider role in myocardial fibrosis, preventing the glycolytic metabolic reprogramming

required for myofibroblast maturation[57,58], and potentially blocking the cell cycle arrest that has been shown to be a feature of fibrosis[59]. We have also shown that WWP2 positively regulates the expression of established pro-fibrotic markers in primary (myo)fibroblasts cultures (Fig. 4). However, one limitation of our study relates to the maintenance of a quiescent cardiac fibroblast

**Fig. 5** WWP2 regulates the nucleocytoplasmic shuttling and TGFβ-signal transduction activity of SMAD2. **a** Immunofluorescence analysis of WWP2 in WT primary cardiac (myo)fibroblasts shows nuclear localization after 16 h TGFβ1 stimulation. Scale bar: 20 μm. **b** Immunoblotting for WWP2 isoforms in WT (myo)fibroblasts reveals that upon 16 h TGFβ1 stimulation, there is an increased amount of WWP2-FL and -N isoforms in the nuclear extraction. **c** Representative WB of p-SMAD2 and SMAD2 protein levels in WT and WWP2[Mut/Mut] (myo)fibroblasts with or without TGFβ1. **d** In-cell ubiquitylation of SMAD2 in primary (myo)fibroblasts from both WWP2[Mut/Mut] and WT mice. Cells were treated with MG132 (10 μM, 3 h) followed by TGFβ1 (5 ng/ml, 6 h). Lysates were prepared from WWP2[Mut/Mut] and WT (myo)fibroblasts and then were subjected to immunoprecipitation with anti-SMAD2 antibodies, followed by western blotting probed with antibodies as indicated. **e** Quantification of TGFβ1-induced luciferase reporter activity of SMAD2 in WT and WWP2[Mut/Mut] (myo)fibroblasts (P values calculated by Mann–Whitney U test; n = 4 for each group; data reported as means ± SD) **f** WB showing SMAD2 and SMAD4 protein distribution in cytoplasmic and nuclear fractions of WT and WWP2[Mut/Mut] (myo)fibroblasts after TGFβ1 stimulation (16 h). WCL whole-cell lysis, Cyto cytoplasmic, Nuc nuclear. **g** Top: representative immunostaining of SMAD2 in WT and WWP2[Mut/Mut] (myo)fibroblasts treated with TGFβ1 for 1 h and then with SB431542 for 5 h. Scale bar: 50 μm. Bottom: quantification analysis shows delayed exportat of SMAD2 from the nucleus in the WWP2[Mut/Mut] group (five biological replicates, eight images each; P values calculated by Mann–Whitney U test; data reported as means ± SD). **h** Schematic of the proposed model for the WWP2-mediated regulation of SMAD2 nucleocytoplasmic shuttling in cardiac fibrosis

phenotype using common cell culture methods, and the cultured fibroblasts are prone to be converted to myofibroblasts before TGFβ1 treatment[60]. In our study, we notice that the cultured cells presented pro-fibrotic phenotype without TGFβ1 treatment, which also explains why the pro-fibrotic response to TGFβ1 was relatively modest. Therefore, we referred to the in vitro cultured cardiac fibroblasts as "(myo)fibroblasts". However, TGFβ1 stimulation increased (myo)fibroblasts maturation, inducing the expression of pro-fibrotic markers and ECM genes, which was significantly reduced in WWP2[Mut/Mut] cells. The differential fibrotic response between WT and WWP2[Mut/Mut] cardiac (myo)fibroblasts was corroborated by WWP2 siRNA experiments and gain-of-function rescue experiments.

Poly-ubiquitination and proteasome-mediated degradation of nuclear SMAD2 have been documented[61]. In addition, mono-ubiquitination can regulate nuclear accumulation and the nucleocytoplasmic shuttling of SMAD complexes, which are crucial for transduction of TGFβ-superfamily signals[62] and there is mounting evidence supporting the role of monoubiquitination of SMADs by E3 ubiquitin ligases in this context[45,47]. Here, we reveal that TGFβ1 stimulation promotes the translocation of WWP2 to the nucleus, where it interacts with SMAD2, possibly promoting its monoubiquitination, as shown for other E3 ubiquitin ligases[44,45]. The regulatory consequences of E3 ligase-mediated monoubiquitination appear to be complex and context specific. Monoubiquitination can regulate protein localization, activity, and protein interactions with binding partners[63]. Monoubiquitination is also required for the activity and the intrinsic nuclear import of target TFs[41,64], and to disrupt specific TFs interactions and their transcriptional activity[65]. In our studies, we did not observe SMAD2/3 degradation by endogenous WWP2 in primary (myo)fibroblasts, suggesting a mechanism of degradation-independent repression of SMAD2 activity downstream of TGFβ-signaling activation. Our data suggest that blocking WWP2 function can delay the TGFβ1-induced nucleocytoplasmic shuttling of SMAD2 (Fig. 5h). However, further studies are required to prove whether WWP2-mediated monoubiquitination of SMAD2 disrupts or directly regulates the activity SMAD complex[47,66] and to explain mechanistically how this WWP2-monoubiquitination affects the nuclear export and regulation of SMAD2 (ref. [67]).

Ubiquitin ligases are involved in a wide range of diseases, and targeting E3 ubiquitin ligases is thought to yield higher specificity and less toxicity than other ubiquitins[68]. E3 ubiquitin ligases can both degrade and activate specific substrates, and so may allow for a specific regulation of currently "undruggable" protein targets[69]. To date, only a handful of small molecules for cancer therapy (e.g., targeting MDM2 (ref. [70]) and SKP2 (ref. [71])) have been successfully developed using this type of enzymes. However, several E3 ubiquitin ligases such as *SMURF1*, *SMURF2*,

*ARKADIA*, *SYNOVIOLIN*, *NEDD4*, and *PELLINO1* (reviewed in ref. [72]) have been proposed as regulators of tissue fibrosis, through several mechanisms including regulation of TGFβ-signaling. This study adds to our understanding of the regulation of cardiac fibrosis and uncovers a specific E3 ubiquitin ligase, *WWP2*, as a regulator of ECM accumulation in the diseased heart downstream of TGFβ/SMAD signaling. Since cardiac fibrosis has been proposed as an important therapeutic target in HF patients[73,74], the identification of druggable targets that regulate pathological fibrosis may provide new avenues to control the progression to HF. Here, using our systems genetics approach followed by functional validation studies in several cellular and preclinical animal models, we have first identified and then mechanistically explained that WWP2 is a novel and druggable therapeutic target with the potential to control fibrosis in pathological cardiac remodeling.

## Methods

**Data generation and processing in the rat**. This cardiac expression data set consists of RNA-sequencing in the heart (LV) from the 30 rat RI strains, which were previously published[75]. Data were processed with TopHat 1.2.0 (https://ccb.jhu.edu/software/tophat/), which was used to map the reads to the Brown Norway (BN) reference genome RGSC 3.4 (known splice junctions in the Ensembl reference database were supplied, de novo splice junction detection was also enabled). After mapping the reads to the reference genome, Cufflinks 1.0.2 (http://cole-trapnell-lab.github.io/cufflinks/releases/v1.0.2/) was used to assemble the aligned RNA-seq reads into transcripts by using the Ensembl annotation of the rat transcriptome. Cufflinks is able to reconstruct the set of transcripts that "explains" the reads observed in our RNA-seq experiment, in our case, by using a transcriptome reference annotation. The reason for using a reference-based transcriptome assembly approach is that, comparative transcriptome assembly analyses have shown that, overall, reference-based transcriptome assembly approaches have a better performance than de novo approaches, in both sensitivity and the ability to discover splicing patterns[76]. Cufflinks measures transcripts abundances in Fragments Per Kilobase of transcript per Million mapped reads (FPKM). One of the assumptions of reference-based transcriptome assembly approaches is that all the isoforms of all the genes are known. However, transcriptomic reference sets are still incomplete, more specially in the case of the rat transcriptome[76]. Hence, the transcriptomic assembly is not very accurate in rat. Therefore, all the RNA-seq expression data analyses in the rat have been carried out by summarizing expression at the gene level. Six steps followed for gene level FPKM computation and filtering of the data, as follows. (1) Replace the values of all the isoforms with isoform quantification status not *OK* by missing values. The isoform quantification status is part of the output from Cufflinks and it measures the successful deconvolution of each isoform. (2) Remove all isoforms with more than 10 missing values across all samples. (3) Impute all the missing values by using R function *pca* from the *pcaMethods* R package 1.56.0 (https://bioconductor.riken.jp/packages/3.0/bioc/html/pcaMethods.html). The initial number of isoforms was 39,492. After removing those with more than 10 missing values across all samples, the number of isoforms got reduced to 39,181. (4) Sum up all the FPKM values of all the isoforms of each gene to get the FPKM value at the gene level (number of genes: 29,469). (5) Filter out genes that did not have FPKM >1 in at least a 5% of the samples (i.e, in at least two samples). (6) Replace 0s in the data by the minimum number in the data set higher than 0. Afterwards, Log2 transform the data. By following these steps, we ended up with a filtered data set of 12,061 genes. We also inspected the presence of unmeasured confounding factors (i.e., potential sources of expression variation that are not being measured in the study and they may be affecting gene expression)[77].

We computed the variability explained by each of the principal components (PCs) of the transcriptomic data. We found out that the first PC was explaining a large proportion of the variance present in the data (38%) and it was significantly correlated to the library concentration of the Bioanalyser (a library quantification measure, Spearman's rank-order correlation = 0.70, $P = 1.5 \times 10^{-5}$). To account for possible confounding effects introduced by the differences in library concentration, we adjusted the gene expression data for this first PC using a linear regression model from which the residuals were computed. These residuals obtained from the regression model represent the variability present in the data after removing the effect of this first PC. These adjusted expression levels were the ones considered in all analyses.

Histomorphometric measures of both interstitial and perivascular fibrosis were collected in the LV of the 30 RI strains (total $n = 180$; $n = 5–7$ per strain, males at 30 weeks of age). After fixation, short-axis heart slices were processed for paraffin embedding. Multiple 4-μm-thick sections were deparaffinized, rehydrated, and picrosirius red-stained sections were prepared for evaluating fibrosis. The presence, type, and extent of both interstitial and perivascular fibrosis was quantified via thresholding automated analysis using ImageJ 1.43 (ref. [21]). BP measurements were published in ref. [20]. Indwelling aortic radiotelemetry transducers (Data Sciences International) at 8 weeks of age were implanted to measure arterial pressure in conscious, unrestrained rats. Radiotelemetry BP was collected in 5-s bursts every 10 min and recorded over a period of 8 days, using 6–12 rats for each RI strain. The obtained BP measurements were averaged within each RI strain and across eight sequential readings. Median BP effects were removed from the interstitial and perivascular fibrosis measurements by performing standard multiple linear regression and then taking the residuals of the model. Fibrosis and BP measurements are included in Supplementary Data 1c.

Within the rat RI panel, SNPs are a comprehensive source of genetic diversity available for genetic association studies. The genetic map of the rat BXH/HXB RI strains was generated by the STAR Consortium as described in ref. [20]. This genetic map was generated from over 13,000 SNPs that led to 1,384 unique blocks of adjacent SNPs with identical strain distribution patterns.

**Data generation and processing in the human cohorts**. These DCM and control heart data represent a retrospective cohort of patients diagnosed with DCM (obtained from the Royal Brompton and Harefield NHS Foundation Trust Tissue bank: EC Ref: 09/H0504/104 +5) and healthy LV donors (healthy with respect to myocardial diseases such as DCM and HCM). Control left ventricular samples (healthy LV donors) were collected from healthy human hearts of non-related organ donors whose hearts were explanted to obtain pulmonary and aortic valves for transplant, valve replacement surgery, or explanted for transplantation but not used due to logistical reasons. In both DCM patients ("cases") and controls only adult subjects (≥16 years old) were selected. The study was done in compliance with ethical regulation and was approved by the UK National Research Ethic Service (NRES) Committee London-Fulham. All patients gave informed consents.

Human LV RNA-seq data were collected in 128 DCM patients and 106 controls. In addition, genotyping data were collected in 96 DCM and 91 controls samples. We performed a quality control step (QC) in the RNA-seq data and removed outlier samples (see section "RNA extraction, sequencing and RNA-seq data processing"). After QC, the final cohort size used for co-expression analyses was 126 DCM patients and 92 control samples. The number of samples used for genetic mapping was 96 DCM and 91 controls (i.e., samples for which we have both RNA-seq and genotyping data). These data were published in Heinig et al.[26] and are available at the European Genome-phenome Archive under the accession number EGAS00001002454.

Reads were mapped to the human genome with TopHat 1.4.1. TopHat was run using annotation from Gencode release 19 (GRCh37.p13) and allowing only two mismatches per 100 bp. In addition, TopHat was run with option -r 0, which specifies as zero the expected mate inner distance (as fragment size was 200 bp and read length was 100 bp). Option -M was also set, which removes multimapping reads before aligning to the transcriptome. Default options were chosen for the rest of parameters. TopHat mapping resulted in a mean of 173M reads per sample being uniquely mappable (a mean of 92% of the total number of reads with all the samples having a mapping percentage higher than 60%) in the samples from DCM patients. Controls samples resulted in a mean of 185M reads per sample being uniquely mappable (a mean of 90% of the total number of reads with all the samples had a mapping percentage higher than 60%). RNA-seq read gene counts were computed with HTSeq software 0.5.3p3 (https://htseq.readthedocs.io/). In HTSeq the mode "intersection-nonempty" was selected (mode suitable to quantify overlapping transcripts on different strands). To compute gene counts with HTSeq, we used the Gencode human annotation version 19 with a custom TTN annotation[26]. Steps followed for processing of the data: (1) *Gene selection*: From HTSeq output, only "*protein coding*" genes with "*known*" status in Ensembl genes (GRCh37.p13 data set) were consider in the analyses ($n = 19,456$). (2) *Filtering of low expressed genes*: Transcript lengths were downloaded from Biomart Ensembl genes (GRCh37.p13 data set). FPKM was computed with DESeq2 1.6.3R package (https://bioconductor.riken.jp/packages/3.0/bioc/html/DESeq2.html) by considering for each gene, the average length of all the transcripts. An FPKM-based gene filtering criterion was then applied (we only kept genes with FPKM >1 in at least 5% of the samples, considering DCM and controls samples together). This

yielded a final number of 14,281 genes that were considered to be expressed in the samples cohort. (3) *Normalization and data transformation*: After the FPKM filtering, gene raw counts were normalized and variance-stabilized transformed (VST) by using DESeq2 1.6.3. (4) *Covariates adjustment*: VST data was split into DCM patients and healthy controls. Then, the data were adjusted separately for relevant technical ("RIN score" and "library preparation day") and clinical ("sex" and "age at tissue collection") covariates by using a multivariate linear model. In the case of the healthy controls, the data were also adjusted for the center from which the tissue had been collected by adding this information as an additional covariate. (5) *Outlier samples removal (samples QC)*: Outlier samples were inspected by clustering the samples' gene expression levels. The DCM and control samples were independently clustered by hierarchical clustering with the R function *flashClust* from flashClust 1.01-2R package (https://rdrr.io/cran/flashClust/, the agglomeration method used was set to "*average*"). The obtained dendrograms were cut at the 99th percentile of the dendrogram height distribution removing all the samples assigned to small clusters: 2 DCM patients and 14 healthy donors. By following these steps, we ended up with two LV adult cohorts of 126 DCM patients (cases) and 92 healthy donors (controls). Within the 126 DCM patients, 106 were males and 20 were females. In the healthy donors group there were 55 males and 37 females. The mean age in DCM patients was 41.3 ± 13.3 years (16–64); mean age in the controls was 42.6 ± 13.6 years (17–72).

A subset of the DCM and control LV samples from the CEU population (96 DCM samples and 91 control samples[26]) were typed by using genotyping arrays. After imputation and quality control, the final number of SNPs in the DCM patients and healthy controls LV samples was 1,309,892 SNPs. In order to run genetic mapping, we carried out a pairwise linkage disequilibrium (LD) filtering (with an $R^2$ threshold of 0.8) based on the 1000 Genomes pilot 1 CEU population. Pairwise SNPs LD was computed using SNAP 2.2 (http://www.broad.mit.edu/mpg/snap/) using the SNP identifiers as input. The SNP data set chosen to run SNAP was the 1000 Genomes pilot 1 in the CEU reference population with an $R^2$ threshold of 0.8 and a distance limit of 500 bp. From SNAP output, SNPs returning a warning message as not present in the SNAP database were removed and not considered further analyses (the number of missing SNPs was 101,020 SNPs). SNAP web tool outputs the input SNPs clustered in LD blocks (a total number of 126,922 LD blocks were returned by SNAP). To obtain a genomic map filtered by LD, we selected one single SNP for each of the LD blocks ($R^2 = 0.8$). SNPs that were not included in any of the LD blocks (not clustered by SNAP) were added at the end. For each LD block we followed these steps: (1) Get all the SNPs in the LD block and compute their pairwise Spearman's ranked correlation. (2) As representative of the LD block, take the SNP that has the highest average Spearman's ranked correlation with the rest of SNPs included in the LD block. With this procedure we obtained a set of 126,922 SNPs, each of them representing a SNP-LD block. Finally, we added to these SNPs the ones that were not included by SNAP in any of the LD blocks (41,315 SNPs). Then, the final number of SNPs in our LD-pruned ($R^2 = 0.8$) genetic map was 168,237 SNPs.

The rTOF cohort consists of 27 patients with rTOF (mean age 32.0 ± 10.6 years, 78% male). Summary of the clinical information of these patients is included in Supplementary Table 1. All the patients aged ≥16 years old and were scheduled for elective pulmonary valve replacement. They were also under the care of the Adult Congenital Heart Disease service at the Royal Brompton Hospital, UK. For the rTOF patients, the clinical data from non-invasive investigations (performed as part of the surgical work-up) were available. This clinical information included electrocardiography, chest radiograph, echocardiography, and CMR. RV myocardial tissue samples from the 27 patients were snap-frozen in liquid nitrogen at the time of tissue sampling intra-operatively. RV tissue from 11 structurally normal hearts donated for cardiac transplantation was also collected (age donors: 34.0 ± 13.0, sex, male/female: 6/5). RV myocardial biopsies were made available via the Cardiovascular Biomedical Research Unit Biobank of the Royal Brompton & Harefield NHS Foundation Trust. Donor RV control tissue was collected and stored at the time of surgery following similar procedures as with the TOF tissue samples.

rTOF RV expression data are a cohort of RV RNA-seq samples with rTOF patients (fibrotic RV) and control samples. TOF is a congenital heart disease (i.e., problem in the structure of the heart) characterized by pulmonary artery stenosis and ventricular septal defect (a hole between the LV and RV and an overriding aorta, which allows blood from both ventricles to enter the aorta leading to cyanosis and RV hypertrophy). TOF needs to be treated surgically in the first year of life to increase the size of the pulmonary valve and arteries and repairing the septal defect. All eligible patients were under the care of the Adult Congenital Heart Disease service at the Royal Brompton Hospital, UK. The Royal Brompton & Harefield NHS Trust and National Heart & Lung Institute Ethics Committee approved this protocol, with informed consent obtained from all study participants. Tissue studies were compliant with UK Human Tissue Act guidelines. Patients provided specific signed permission for RV myocardial biopsies to be taken intra-operatively under direct vision during open-heart surgery when judged technically feasible by the operating surgeon. However, surgical repaired TOF is usually followed by cardiac fibrosis in both ventricles. Additionally, TOF cases are operated several times during heart post-natal development because the prostatic material used at surgeries is unable to grow with the growing of the heart and valve. Therefore, surgery leftover samples can be collected at different ages of the patients.

TRIzol (Life Technologies) was used for total RNA extraction from the frozen samples by following the manufacturer's protocol. RNA was quantified by ultraviolet spectrophotometry and RNA quality was assessed on the Agilent 2100 bioanalyser. RINs ranged from 6.3 to 9.1 (mean $8.2 \pm 0.6$). One microgram of total RNA was used to prepare the RNA-Seq libraries. RNA-Seq libraries were prepared with Illumina TruSeq RNA sample preparation kits by using the protocol for poly-A enriched mRNA. To avoid batch effects, samples were pooled (4–5 samples/pool, 2 lanes per pool). Finally, paired-end $2 \times 100$ bp sequencing was performed on the Illumina Hi-Seq platform (mean sequencing depth of 196M). TopHat 2.0.12 (https://ccb.jhu.edu/software/tophat/) with Bowtie2 2.2.3 (http://bowtie-bio.sourceforge.net/bowtie2/) and Samtools 0.1.18 (https://sourceforge.net/projects/samtools/files/samtools/0.1.18/) was run by using human genome version GRch38 (hg38.78) reference genome. RNA-seq read counts were computed with HTSeq 0.6.1 (https://htseq.readthedocs.io/). The percentage of reads mapping to the human genome was higher that 80% (above 70% is considered an acceptable mapping percentage for paired-end sequenced reads). Steps followed for the normalization, filtering, and adjustment of the data: (1) *Gene selection*: From HTSeq output, only *"protein coding"* genes with status *"known"* in Ensembl genes (GRCh37.p13 data set) were selected (18,964 genes). (2) *Filtering of low expressed genes*: FPKMs were computed with DESeq2 1.6.3R package using the average transcript length of each gene, which were retrieved from Ensembl Biomart (GRCh37 version). We used a FKPM-based filtering criterion by keeping only those genes with a value of FPKM >1 in at least 5% of the samples (in this case 2 samples). Following these criteria, the number of genes got reduced from 18,964 genes to 13,936 genes. (3) *Data transformation*: Size factors normalization and VST was applied to the raw gene counts by using DESeq2 R package 1.6.3. Then, the gene counts that passed the filtering criteria described in the previous step were selected. (4) *Covariates adjustment*: After normalization and filtering, the data were split into TOF patients and controls. Gene expression counts of TOF patients were adjusted for: (1) age at which the tissue was collected (age of operation) and (2) sex. This adjustment was performed by taking the residuals of a multivariate linear model, in which both age and sex were added as predictors. Among the control samples, some had with missing values for age. More specifically, three samples had missing age. The sex of the sample with missing value was imputed by clustering the expression levels of selected sex specific genes (as described in ref. [78]). Therefore, the gene expression counts of the control samples were adjusted only for sex by taking the residuals of a linear model in which sex was added as a predictor.

Differentially expressed genes between rTOF patients ($n = 27$) and controls samples ($n = 11$) were computed using the R package DESeq2 1.6.3 and adding sex as covariate in the model (age was not added as there were several RV control samples with missing age values). *CooksCutoff* DeSeq2 parameter was set to *False*. The rest of parameters were left to default.

**Gene co-expression network analysis**. Co-expression networks were inferred by using weighted gene correlation network analysis (WGCNA) 1.42 (https://horvath.genetics.ucla.edu/html/CoexpressionNetwork/Rpackages/WGCNA/). Two independent WGCNA runs were carried out in the processed RNA-seq LV data (as described in previous sections): one in rat (RI panel, $n = 30$) and one in the DCM cohort ($n = 126$). In both rat and human runs, WGCNA was run using Turkey's biweight midcorrelation. Biweight midcorrelation estimates are more robust to outliers than the standard Pearson correlation as it assigns lower weights to points further from the center of the distribution[79]. In the rat run, among all the soft threshold values ($\beta$) with $R^2 > 0.8$, we chose the $\beta$ that presented the highest mean connectivity ($\beta = 8$). For the human run, the automatic value of $\beta$ returned by the WGCNA function *pickSoftThreshold* was selected ($\beta = 6$). In both cases, a network merge height of 0.25 was chosen as suggested in the original WGCNA guidelines. For the rest of WGCNA parameters, default settings were used. Once the WGCNA networks were obtained, they were alphabetically sorted by name and renamed as M1-M (number last network, Hs-M in the case of human networks) removing the gray cluster (the gray cluster contains all the non-clustered genes). The full list of rat and human co-expression networks can be found in Supplementary Data 1a, b and 2a, b. Gene-annotation enrichment analysis of the rat ($n = 41$) and human ($n = 48$) networks was performed by carrying out functional gene list enrichment analysis using the R DAVID Web Service 1.4.0 (https://bioconductor.riken.jp/packages/3.0/bioc/html/RDAVIDWebService.html).

Rat networks were queried independently from human networks for Gene Ontology (Biological Process, BP, Molecular Function, MF, and Cellular Component, CC) and Kyoto Encyclopedia of Genes and Genomes (KEGG) annotation categories. The gene reference background was set to the group of genes from which the gene co-expression networks were inferred. In the rat data this background was constituted by the genes robustly expressed in the rat LV data (i.e., after the filtering procedure described in previous sections, $n = 12,061$), whereas in the human data the background was the set of genes robustly expressed in the human LV RNA-seq cohort (i.e., after the filtering procedure previously described, $n = 14,281$). In this analysis results were deemed significant if DAVID FDR/100 < 0.05 (i.e., % FDR). For each network, the results for the top significant GO (BP only) and KEGG terms can be seen in Supplementary Data 1a (rat networks) and Supplementary Data 2a (human DCM networks). The full list of KEGG terms enriched in the Hs-M47 (human ECM-network, hECM-network) is found in Fig. 1e.

Genome-wide gene expression levels in the RI strains LV were correlated with both interstitial and perivascular fibrosis using Spearman's ranked correlation (after correcting for average BP effects as described in the previous sections). Here we used the measurements of interstitial and perivascular fibrosis in the rat after normal transformation (qqnorm R function used) and adjustment for mean BP. Student $P$ values were obtained for each correlation estimate by using the function *corAndPvalue* from the R package WGCNA 1.42. All the rat co-expression networks were tested for enrichment of genes varying with fibrosis in the rat heart by carrying out GSEA[25]. In this analysis, each of the rat co-expression networks was considered as a gene set. The corresponding Student $P$ values of the Spearman's ranked correlation between each of the robustly expressed rat genes and each of the fibrosis measurements were used for ranking all the rat genes ($n = 12,061$) and run GSEA. GSEA 2.1.0 was run in classic, pre-ranked mode with 10,000 iterations. To consider all the co-expression networks, maximum gene set size was set to 5,000 and minimum gene set size was set to 10. As the genes were ranked by $P$ value, in this GSEA test a significant (FDR < 0.05) negative normalized enrichment score (which corresponds to weighted Kolmogorov–Smirnov-like statistic) denotes significant association. The results of the association between the rat co-expression networks and interstitial or perivascular fibrosis measurements can be found in Supplementary Data 1c and are displayed in Fig. 1b.

To assess whether the co-expression networks inferred in the rat were also conserved in the DCM patients, we computed the intersection between rat and human co-expression networks by carrying out a Fisher's exact test (FET). The gene background used in these tests was composed by the genes with one-to-one human-rat ortholog relationships that were both robustly expressed in rat and human LV. For the generation of this gene background, rat–human one-to-one orthologs relationships were downloaded from Ensembl archive (Ensembl 69). The common set of genes in both the rat LV expressed genes set ($n = 12,061$) and human LV expressed genes ($n = 14,281$) yielded a common set of 8,840 genes (this will be the rat–human orthologs background). Genes in the rat and human networks that were not included in this rat–human orthologs background were removed. For each rat ($n = 41$), human ($n = 48$) network pair, a FET was computed with the R function *fisher.test* and setting the *"alternative"* parameter to *"greater"* (as we are interested in overrepresentation). The gene background used for computing the contingency table was the rat–human *orthologs background* (8,840 genes). Nominal FET $P$ values were adjusted for the number of tests carried out (number of rat networks × number of human networks = 1,968 tests) by using the R function *p.adjust* and the B&Y method. Figure 1c shows only the human DCM co-expression networks that had some degree of conservation with the rat (adjusted $P < 0.05$), the full list of human DCM co-expression networks can be found in Supplementary Data 2a, b.

Differentially co-expressed networks (i.e., gene networks where the genes as a whole present divergent pattern of co-expression between cases and controls) can point to different disease response. We carried out the following empirical differential co-expression test to assess the differential co-expression of the human networks between DCM patients and controls samples. First, we computed Tukey's biweight pairwise gene–gene correlations from the DCM and controls expression matrices. Then, for each of the DCM co-expression networks these steps were carried out: (1) Compute the network's dispersion value (DCM versus controls samples) as described in the Section 3 of Supplementary Material of ref. [80]. The dispersion value quantifies the difference between the co-expression of the network in cases and controls. (2) Generate a null distribution of dispersion values for the network by randomly sampling networks with the same number of genes as the network being tested and then compute the corresponding dispersion value of each randomly sampled network as described in step 1. (3) Compute the empirical $P$ value for the network under testing: $P = (r + 1)/(n + 1)$, where $n$ is the number of simulated dispersion values (number of permutations) and $r$ is the number of simulated dispersion values that are higher than the actual dispersion value of the network of interest. This differential co-expression test was run with 100,000 permutations, which yields a minimum nominal significance level of $1 \times 10^{-5}$. Bonferroni-adjusted $P$ values were computed by correcting the nominal empirical $P$ values for the number of DCM co-expression networks tested ($n = 48$). *p.adjust* R function was used to adjust the $P$ values for multiple testing. The complete list of differentially co-expressed networks (Bonferroni-adjusted $P < 0.05$) can be found in Supplementary Data 2a. A gene–gene correlation heatmap for the hECM-network can be seen in Fig. 2f. In this heatmap, the top triangular matrix shows the correlation levels for each pair of genes in the hECM network in the DCM patients whereas the lower triangular matrix shows the correlation levels in the LV controls. This differential co-expression test was also carried out for the hECM-network in the rTOF/controls RV cohort. In this case the input was the sex-adjusted VST counts for the genes in the hECM-network. The test was also performed for 100,000 permutations. The gene–gene correlation heatmap of the hECM network can be seen in Fig. 2f. In this heatmap, the top triangular matrix shows the correlation levels in the rTOF patients whereas the lower triangular matrix shows the correlation levels in the controls. Additional details on enrichment analyses and functional annotations of the human networks is reported in Supplementary Note 2.

**Bayesian network-expression QTL (network-eQTL) analysis**. Co-expression networks suggest coordinated genetic regulation, which can be exploited to uncover

genetic regulators of these transcriptional programs. Moreover, conserved genetic regulation can be driving fundamental biological mechanisms. In keeping with previous studies in the rat, where the BXH/HXB rat panel yielded increased power to carry out genetic mapping of gene networks[34], we used multivariate Bayesian genetic mapping approaches to map the rat and human ECM-networks to the rat and human genomes. We first considered the expression of the rat ECM-network genes as a multivariate quantitative trait and jointly mapped this to the rat genome. Then, we inspected whether the regulatory locus identified in the rat was independently replicated in human DCM heart by joint mapping of the genes in the ortholog human network (e.g., hECM network) to the human locus that is syntenic to the rat regulatory loci. This two-step strategy (mapping in rats first followed by mapping in humans) has been previously used to identify *trans*-acting genetic regulators of transcriptional networks underlying complex disease[34]. The mapping of the rat and human networks was carried out by using HESS[33]. HESS is a sparse Bayesian multiple linear regression method in which mRNA expression levels for multiple genes are regressed against all SNPs to identify the minimum (non-redundant) set of SNPs that predicts the mRNA expression variability. This method has the following features: (1) it takes into account the LD structure of the genotype data (the dependence of the genetic determinants or predictors), allowing to reduce the number of tests to be carried out and pinpoint the putative causal genetic variant; (2) it makes possible to map several responses in one single test; therefore, with HESS is possible to map to the genome (i.e., all genome-wide genetic markers) expression levels of several genes jointly (for instance, map the expression levels of genes included in a co-expression network, without having to summarize their variability by PC analysis); and (3) it exploits multidimensional dependencies within the responses (i.e., correlation of the gene expression levels). This can be used to boost detection of moderate *trans*-acting eQTL effects. The output of this method is a marginal posterior probability of inclusion (MPPI) for each gene-SNP pair tested, which represents the posterior probability of association of each SNP given the data. From this MPPI, the BF can be computed. BF represents the evidence of genetic regulation versus no genetic control and it is defined as the ratio between the posterior odds and the prior odds or ratio between the strengths of these models[34]. In our case, the prior probability ($\pi$) for the *j*th SNP associated with the *g*th gene is defined as: $\pi = \frac{E(p_g)}{p}$, where *p* is the number of SNPs we are testing and $E(p_g)$ is the a priori expected number of control points for the *g*th gene, in our case we fix $E(p_g) = 2$. For instance, in the case of the rat RI strains, as the number of genome-wide SNPs is $p = 1384$ and the prior probability becomes $\pi = 1.4 \times 10^{-3}$. The BF is defined as the ratio between the posterior odds and the prior odds: $BF = \frac{MPPI_{gj}/(1-MPPI_{gj})}{\pi/(1-\pi)}$, where $MPPI_{gj}$ represents the marginal posterior probability of inclusion for the *g*th gene and the *j*th SNP. By using this BF formula, we can compute the BF for each response-predictor pair (i.e., gene and SNP under testing) from the output MPPIs of HESS. *Genome-wide mapping of the rat networks in the RI strains*. The gene expression levels of the genes included in each of the rat co-expression networks built in the LV of interest were jointly mapped to the rat genome with HESS, i.e., rat networks conserved in human, overrepresented for genes correlating with fibrosis and with a pattern of co-expression not present in human control LV tissue: M1, M2, and M12 rat networks. The expression data used was the RNA-seq data in the RI strains (log2 transformed FPKM adjusted for the first PC). These runs were carried out with 1,384 genome-wide SNPs markers in 29 RI strains (instead of 30, as there is one RI strain with RNA-seq expression but no available genotype information).

For each of the rat networks that had regulatory loci with median BF of the genes in the rat networks >100 (i.e., rat networks M1 and M2), the rat regulatory SNPs were selected. In humans, two independent HESS runs were carried out, one for the DCM patients and one for controls. The human expression data input to HESS was the expression level of the genes in the human networks that were significantly intersecting the rat modules of interest (RNA-seq data in the DCM/controls after the processing described above). This was done in the $n = 96$ DCM and $n = 91$ control heart samples for which we had both gene expression and genotype information. The genetic data input to HESS was the set of SNPs tagging the human locus syntenic to the identified rat locus in each case. To identify each the human syntenic locus, we follow these steps. (1) Obtain the start and end positions of the rat haplotype that contained the regulatory SNP (rat Ensembl version 69). (2) Compute the central genetic coordinate of the rat haplotype as: center haplotype = start haplotype + (end haplotype-start haplotype)/2. (3) Get the closest rat gene to that coordinate, then get the human start/end coordinates of the human ortholog gene. (4) In humans (human Ensembl version GRCh37), compute the center of the selected gene: center gene = start gene + (end gene-start gene)/2. Take a window of 10 Mb (±5 Mb) around the *center gene*, which in the case of the hECM network yielded the region Hs-chr16: 64415969..74415969 (human Ensembl version GRCh37). This was the region that we mapped in the human DCM data (which comprises 475 SNPs from our LD-pruned genetic map). Additional details on the HESS runs are reported in Supplementary Methods.

### Correlation analyses between *WWP2* and hECM-network genes.

*WWP2* transcript levels in the LV cohorts (DCM/control) and RV cohorts (rTOF/control) were correlated separately in patients and controls to the genes included in the hECM network. We computed the Spearman's ranked correlation and *P* value using the WGCNA R package function corAndPvalue WGCNA 1.4. The resulting

correlation distributions were plotted as a density plot and tested for differences by using a two-sample non-parametric Mann–Whitney *U* test by using the R function wilcox.test. See the density plots with the obtained *P* values in Fig. 2d, e. The correlation between *WWP2* and all the genes in the hECM-network in LV DCM and RV rTOF patients is included in Supplementary Data 3e. Nominal *WWP2*-gene correlation *P* values were corrected for multiple testing by using the R function *p.adjust* (method "fdr"). In this multiple testing correction, we corrected for the number of genes in the hECM-network, i.e., 683. A core set of genes correlated to WWP2 was extracted by taking all the genes with correlation FDR < 0.01 in both DCM and rTOF patients (326 genes). STRING protein–protein interaction database 10.0 (https://string-db.org) was queried with this set of genes on the 05/09/18. The resulting network was retrieved and imported into Cytoscape 2.8.1 (https://cytoscape.org/) to generate a network visualization graph (see Fig. 2f, only the largest connected component is displayed, 40 genes). To obtain this network graph from STRING database, we only considered experimental connections and co-expression interactions with a minimum interaction score of 0.4. In the network graph, each node corresponds to a gene and color was mapped to the correlation between *WWP2* and each gene (the average correlation in DCM and rTOF patients). Genes annotated in STRING with Gene Ontology Cellular Component (CC) term "extracellular matrix" were also retrieved and highlighted with thicker node border.

### Expression QTL (eQTL) mapping of WWP2 isoforms in DCM.

*WWP2* isoform expression levels were quantified in the DCM patients from the TopHat output with Sailfish 0.6.3 (http://www.cs.cmu.edu/~ckingsf/software/sailfish/). Sailfish was run with default parameterizations. The expression levels in the DCM patients of all inferred *WWP2* isoforms (in Reads Per Kilobase Million, RPKM) are included in Supplementary Data 3. These data were adjusted for relevant technical ("RIN score" and "library preparation day") and clinical ("sex" and "age at tissue collection") covariates by using a multivariate linear model and then mapped to the regulatory locus identified in human chromosome 16 (1 Mb centered around *WWP2*). In this case we mapped the 27 SNPs tagging this region in our genetic map. From the output MPPIs, BFs were computed as described above. Results of this test for the three *WWP2* isoforms with highest expression in the DCM heart are included in Fig. 2i along with a boxplot showing the expression level by genotype at the regulatory SNP rs9936589 (the expression level is displayed after controlling for clinical covariates effects as described above). In each boxplot a non-parametric Kruskal–Wallis test was computed by using the R function *kruskal.test*. The obtained *P* values were corrected for the number tests carried out (i.e., the number of *WWP2* isoforms inferred in the DCM heart with an average RPKM level higher than 0.18 isoforms). *P* values were corrected by using the R function *p.adjust* and the correction method "fdr". The obtained FDRs are displayed in the boxplots in Fig. 2i.

### Animal studies.

Mice were bred and maintained in a specific pathogen-free (SPF) environment. We complied with all relevant ethical regulations according to the guidelines issued by the National Advisory Committee on Laboratory Animal Research. Protocol with IACUC number 2016/SHS/1170 was approved by Institutional Animal Care and Use Committee of National University of Singapore, Duke-NUS Medical School. Steps were taken to minimize animal suffering. *WWP2^{Mut/Mut} mouse generation*: WWP2^{Mut/wt} mice were generated by CRISPR/Cas9 technology based on C57BL/6J strain in the laboratory of Dr. Weiping Yu at Agency for Science, Technology and Research (A*STAR), Singapore. Briefly, mutant animals were generated by co-injection of Cas9 mRNA and individual gRNAs into one-cell mouse embryos. Founder animals carrying the *indel* mutations were identified first by PCR and T7 endonuclease I assay, and then by deep sequencing of the PCR products. Founders carrying the desired reading frame shift mutations were used to generate mutation-segregated heterozygous F1 animals by crossing with the wild-type animals. Homozygous mutant animals were generated by heterozygote crossing and used for experiments in comparison with the wild-type littermates.

To look into the specific function of individual Wwp2 isoforms, three gRNAs were designed to targeting coding Exon 2, aiming to introduce mutations in individual domains of the protein, which in humans have different functions and are encoded by the three different gene isoforms. Here, a reading frame shift mutation in Exon 2 would render Wwp2-FL and Wwp2-N functionally null, but is unlikely to affect Wwp2-C function (see Supplementary Fig. 2). WWP2^{Mut/wt} mice were crossbred to generate WWP2^{Mut/Mut} and WWP2^{wt/wt} (WT) mice in vivarium at Duke-NUS Medical School, Singapore. *Details on AngII (Angiotensin II) infusion model*: Alzet miniosmotic pump (Model No. 1004, Durect Corporation) was subcutaneously implanted in eight-week-old mice anesthetized with 2% isoflurane. Miniosmotic pumps loaded with saline or Angiotensin II (Sigma Aldrich, #A9525) were implanted to deliver AngII at 500 ng/kg/min for a period of 4 weeks. *Details on myocardial Infarction (MI) model*: MI was induced in 8–10-week-old mice after anesthetizing with ketamine and xylazine and intubated with a 22GX1″ SURFLO Flash I.V. catheter (TERUMO), which was connected to an artificial rodent ventilator MINI VENT type 845 (Harvard Apparatus, USA). After exposing the heart via thoracotomy at the fourth left intercostal space, the left coronary artery was permanently ligated with an 8-0 nylon monofilament suture. The thorax was

closed with 6-0 coated vicryl suture and mice were followed for 4 weeks after surgery.

For both models, mice were sacrificed after weighing them at indicated time points. Hearts were harvested for weight measurement, histological studies, collagen determination, and molecular biology analyses. Samples sizes were determined by power analysis and $n \geq 8$ mice per group were used to account for the inherent variability in the fibrotic response of mice. Mice that died undergoing surgery before the sample collection were excluded from statistical analysis. Data from the animal studies were collected in a blinded manner.

**Cell culture experiments**. Murine cardiac fibroblasts were taken from the LV of mice. Minced LV pieces (1–3 mm$^3$) were placed in 6 cm dishes with DMEM supplemented with 20% fetal bovine serum for less than 10 days to generate mice cardiac fibroblasts (P0) and passaged to P1 and P2 DMEM supplemented with 10% fetal bovine serum for experiments. In each experiment, all the cells from WWP2$^{Mut/Mut}$ and WWP2$^{wt/wt}$ (WT) heart were cultured at the same time with same generation. Human cardiac fibroblasts were isolated from the right atrium (RA) appendage obtained from patients on cardiopulmonary-by-pass during cardiac surgery operations by digesting the tissue with Collagenase II. Cardiac fibroblasts are obtained by growing the homogenized tissue suspended in DMEM supplemented with 20% fetal bovine serum in a humidified atmosphere. However, fibroblast-to-myofibroblast conversion occurs with each cell passage using a common cell culture method[60]. Our primary cardiac cells isolated and cultured presented myofibroblasts features, and thus we refer to them as "(myo)fibroblasts".

C2C12 myoblast cell line, gifted by Dr. Lisa Tucker Kelloggs' laboratory, Duke-NUS Medical School, Singapore, were grown in DMEM medium supplemented with 10% fetal bovine serum. The cells were passaged twice before being used for experiments.

To mimic the in vivo cardiac (myo)fibroblast activity, cells were treated with TGFβ1 human (Sigma Aldrich, #T7039) at a concentration of 5 ng/μl for 16–72 h. For siRNA and plasmid transfection, primary cardiac (myo)fibroblasts were seeded on a six-well plate (~70%) and were transiently transfected with siRNA duplexes (20 nM) designed for targeting 5′ or 3′ in WWP2 mRNA (Qiagen) using Lipofectamine RNAiMAX (Life technologies holdings, #13778075) in a serum-free medium for 48–72 h according to the manufacturer's instructions. In parallel, WWP2-FL and -N plasmids were transiently transfected using Lipofectamine 2000 (Life Technologies Holdings, #11668019) for 48–72 h according to the manufacturer's instructions. For siRNA transfection in human cardiac (myo)fibroblasts, primary human cardiac (myo)fibroblasts were seeded on a 12-well plate (70,000 cells/well) and were transiently transfected with siRNA duplexes (20 nM) designed for targeting 5′ or 3′ in WWP2 mRNA (Qiagen) using Lipofectamine RNAiMAX (Life Technologies holdings, #13778075) in a serum-free medium for 24 h according to the manufacturer's instructions.

For ubiquitination analysis, cells were treated with proteosomal inhibitor MG132 (Sigma Aldrich, #M7449) following stimulation with TGFβ1 for 4 h before harvesting. SB431542 (Stem Cell Technologies, #72234) was used to inhibit TGFβ1 effect.

**Echocardiography analysis**. Transthoracic echocardiography was performed on day 28 after AngII infusion and MI model using Vevo 2100 (VisualSonics, VSI, Toronto, Canada) and a MS400 linear array transducer, 18- to 38-MHz under anesthetized condition. An average of 10 cardiac cycles of standard two dimension (2D) was acquired and stored for subsequent analysis using Vevo Imaging Workstation version 1.7.2 (VisualSonics, VSI, Toronto, Canada). All images acquisition and analysis were performed by a blinded operator. For the AngII infusion model, 2D-guided M-mode of parasternal short-axis short (middle) were selected for visualization of the papillary muscle during end systole and end diastole. For the MI model, the parasternal long axis was analyzed at three levels (basal, mid, and apical) and all measurements were averaged over three consecutive cardiac cycles. LVEF and FS were calculated using the modified Quinone method, using the following formulas: LVEF = (LVIDed$^2$−LVIDes$^2$)/LVIDed$^2$; FS = (LVIDed−LVIDes)/LVIDes, where LVIDed is left ventricular internal diameter at end diastole and LVIDes is left ventricular internal diameter at end systole.

**Single-cell RNA-sequencing analysis in mouse heart**. Single-cell suspension was prepared from the adult LV of one mouse with Angiotensin II infusion for 28 days. After removal of dead cells with MACS dead cell removal kit (Miltenyi Biotec, #130-090-101), cells were lysed and subsequently RNA was reverse-transcribed and converted into cDNA libraries for RNA-seq analysis using a Chromium Controller and a Chromium Single Cell 3′ v2 Reagent kit (Genomics 10×) following the manufacturer's protocol. The library was sequenced using the Illumina Hi-Seq3000 sequencing platform.

The reads were mapped to the mouse genome (m38, Ensembl version 89) and quantified using Cell Ranger 2.1.1 (10x Genomics). We provided to Cell Ranger a custom built reference transcriptome generated by filtering the Ensembl transcriptome (Ensembl file: Mus_musculus.GRCm38.89.gtf) for the gene biotypes: protein coding, lincRNA, and antisense. Cell Ranger was run with the expect number of cells parameter (expect-cells) set to 3000. Cell Ranger out filtered matrices (i.e., genes.tsv and barcodes.tsv) were then input into R and genes with

zero counts in all cells were discarded. Three cell quality control filtering steps were implemented. We removed: (a) cells with less than the 50th percentile of the distribution of the total cells library size, (b) cells with less than the 50th percentile of the distribution of total of number of detected genes, (c) cells with more than 50% of their total gene count coming from mitochondrial genes. This resulted in a final number of 508 cells. After applying these cell filtering steps, we carried out additional gene quality control steps on the remaining cells: (a) we only kept "detectable" genes, defined as genes detected with more than one transcript in at least two cells, (b) we removed genes with low average expression in the data (i.e., genes with an average expression below 0.01, this cutoff was set based on the total distribution of average gene expression across all cells and all genes), (c) we removed genes encoded on the mitochondrial genome and the gene "Malat1" as it was an outlier in the gene expression distribution.

After all the gene quality control steps, the resulting number of genes was 6,728. Gene counts were normalized with scran 1.8.4R package (https://bioc.ism.ac.jp/packages/3.7/bioc/html/scran.html). Scran size factors were computed from cell pools by doing a pre-clustering of the data with the quickCluster function (the output object of this function was provided to the computeSumFactors function and then run the normalize function was run, all of them with default parameterizations). t-SNE was computed by providing the Log$_2$ scran-normalized data to the function plotTSNE from scater 1.8.4R package (http://packages.renjin.org/package/org.renjin.bioconductor/scater). Two tSNE components were computed setting a random seed of 123456 using automatic perplexity (after removing an imposed minimum of 50, i.e., floor (number cells/5)). In the t-SNE graph, each cell was colored by Wwp2 expression level (Log$_2$ scran-normalized gene counts). See this t-SNE graph in Fig. 4b. Cardiac cell subpopulations were identified using established cell marker genes. Specifically, we used: Aplnr and Pecam1 (endothelial cells); Lum (fibroblasts); Ttn (cardiomyocytes); Hbb-bs (eritrocytes); Ccr2, Cd163, and Ptprc (immune cells).

**Analysis of WWP2$^{Mut/Mut}$ mouse RNA-seq data**. Total RNA from tissue was isolated from LVs of nine WWP2$^{Mut/Mut}$ and nine WT mice with 28 days of Angiotensin II infusion. mRNA libraries were constructed from poly(A)-selected RNA using the NEBNext Ultra Directional RNA library prep kit (Illumina, New England BioLabs) and sequenced on Illumina Hi-Seq3000 sequencing (150 × 2 bp, seven samples were loaded per lane).

RNA-seq reads were assessed for quality, aligned to m38 (Ensembl Gene annotation build 89) using STAR 2.5.2b (https://github.com/alexdobin/STAR) and quantified with RSEM 1.2.31 (https://github.com/deweylab/RSEM/). The average mapping rate (unique and multimapping) was 94.5%. Gene annotation was retrieved from Ensembl version 89 (m38) using the R library biomaRt 2.30.0 (https://bioconductor.riken.jp/packages/3.4/bioc/html/biomaRt.html). Ribosomal genes (Ensembl gene biotype "rRNA") and mitochondrial genes were removed (391 genes). Gene counts were rounded using the R function round and differential expression analysis was performed with DESeq2 1.14.1 with a pre-filtering step in which we considered only genes with more than 1 count when summing up across all samples. DESeq2 was run pairwise comparing WWP2$^{Mut/Mut}$ with Angiotensin II against WT mice with Angiotensin II using the Wald test, with the outlier correction parameter cooksCutoff set to false (default parameterizations for the rest of parameters). In the DESeq2 model, we added RNA concentration and sequencing lane as covariates. Functional enrichment analysis of the differential expression results was performed with GSEA[25] software v 2-2.2.2. From all genes included in DESeq2 output, we selected those with one-to-one mouse–human ortholog relationships (as downloaded from Biomart, 14,820 genes) and then we mapped them to human gene symbols. Then ranked all the genes by the corresponding DESeq2 output Wald statistic (i.e., the estimate of the log2 fold change divided by its standard error). GSEA was run to assess the overrepresentation of the following gene sets and pathways derived from the Molecular Signatures Database gene sets 5.1 (http://software.broadinstitute.org/gsea/msigdb/collections.jsp) (gene sets were queried using gene symbols): Hallmark gene sets (i.e., coherently expressed gene signatures derived from the aggregation of many MSigDB gene sets to represent well-defined biological states or processes), Gene Ontology, and Reactome databases. GSEA was run in classic pre-rank mode with 10,000 permutations to assess false discovery rate (FDR). In the GSEA runs, maximum gene set size was set to 5,000 and minimum gene set size was set to 10. In this test, upregulated processes and pathways in the WWP2$^{Mut/Mut}$ will be positively enriched, whereas downregulated processes will be negatively enriched. Gene sets were deemed as enriched if FDR < 0.05. All Hallmark enriched gene sets are displayed in Fig. 3j. The rest of the results can be found in Supplementary Data 4a. In addition, GSEA was run a second time using the same parameterization but this time testing for overrepresentation of all the human co-expression networks. In this last run, we further reduced the background to the common set of genes with one-one mouse–human ortholog relationship that were also present in both the DESeq2 mouse output and in the initial set of human genes considered for network inference in the human data. This background reduction was applied to both the ranked list and the human networks, resulting in a total number of 10,449 genes. Results of this test are included in Supplementary Data 4b.

We tested whether the hECM-network genes displayed differential co-expression upon Angiotensin II infusion when comparing the WWP2$^{Mut/Mut}$ mouse with control mice. To this aim, we followed the same procedure as we

previously did to compute differential co-expression of the hECM-network genes in the two human cohorts (DCM/controls and rTOF/controls, see previous sections). We applied a filtering step and removed lowly expressed genes (i.e., we only retained genes with FPKM > 1 in at least 2 out of the 18 samples), this resulted in 12,659 genes. Then, as the hECM network was inferred in humans, out of these 12,659 genes, we only considered the ones with one-to-one ortholog relationship to humans (10,271 genes). From these genes we took the set of genes included in the hECM-network (415 genes). We added an offset of 1 and computed the $Log_2$ FPKM. Then we computed gene-gene pair Turkey's biweight midcorrelation separately in the WT-Angiotensin II and WWP2$^{Mut/Mut}$-Angiotensin II mice and carried out the same test for differential co-expression used for the human networks. The $P$ value and the heatmap showing the correlation in the hECM-network genes in these mice can be seen in Fig. 3i.

**Histology and immunofluorescence.** LVs harvested from the mice were fixed in 10% neutral buffered formalin (NBF) for 24 h at RT, processed with a Leica automatic tissue processor, paraffin-embedded and sectioned with thickness of 5 μm. After dewaxing and rehydration, slides were stained with Sirius Red Collagen kit (Chondrex, Inc., #9046) and Masson's Trichrome staining kit (Sigma Aldrich, #HT15) as per the manufacturer's instructions. Sections were stained using anti-ACTA2 (1:100) and anti-S100A4 (1:100) to identify cell and biochemical features. Bovine Anti Rabbit IgG-CFL 488 (Santa Cruz Biotechnology, #sc-362260) and Bovine Anti Mouse IgG-CFL 488 (Santa Cruz Biotechnology, #sc-362256) were used as secondary antibodies for immunofluorescence. Rhodamine Wheat Germ Agglutinin (WGA, Vector laboratories, #RL-1022) was used to stain the myocytes. Cells were grown in a eight-well chamber slide with a removable silicone chamber (ibidi) up to 70% confluence. After fixation with ice cold acetone and blocking with 1% BSA for 30 min at RT, the slides were incubated with primary antibodies anti-WWP2-FL/N(1:100), anti-ACTA2 (1:100), anti-S100A4(1:100), anti-Vimentin (1:100), and anti-FLAG(1:100) overnight at 4 °C. Following washing steps, the slides were incubated with Bovine Anti Rabbit IgG-CFL 488 (Santa Cruz Biotechnology, #sc-362260) and Bovine Anti Mouse IgG-CFL 488 (Santa Cruz Biotechnology, #sc-362256) for 2 h at RT. VectaShield Mounting Medium (Vector laboratories, #H-1200) with DAPI was used to stain the nuclei and the slides were covered by coverslip. Slides were imaged on a Leica fluorescence microscope and image was processed using ImageJ software with the Fiji package.

With merge function, the positive Sirius red staining in the whole section of the LV was quantified using custom semiautomated image analysis routine. To measure the fluorescence intensity of ACTA2 and COL1A1 in different cellular sections, images were taken using ×20 Plan Fluor objective. Fluorescence intensity was measured by taking the integrated intensity of a region of interest and subtracting the background intensity, and normalized to cell number. For each group, at least six fields were analyzed per section. To measure fluorescence intensities of SMAD2, 1-μm Z-stacks through cells of fields interested were acquired. A region was drawn around each cell and nucleus to be measured, and background without fluorescence was subtracted. The nuclear/cellular fluorescence intensity ratio was calculated. Each field represented around 8–10 cells and at least four fields were analyzed for each section.

**Hydroxyproline assay.** The amount of total collagen in the LV was quantified using the Quickzyme Total Collagen assay kit (Quickzyme Biosciences). The assays were performed according to the manufacturer's protocol.

**Luciferase assay.** Cells were transfected with a luciferase reporter gene plasmid with SMAD binding sites (Yeasen, SMAD-Luc, #11543ES03) and co-transfected with pGMLR-TK (Yeasen, #11557ES03) as a normalization control. Thirty hours after transfection, cells were treated with vehicle or TGFβ1 for 16 h and harvested. Luciferase assays were performed using the Dual-Luciferase Reporter Assay System (Yeasen, #11402ES60).

**Cell proliferation and migration assays.** Cell proliferation was quantified by MTS assay (Promega) according to the manufacturers' protocol. For migration assay, cells were seeded at a density of 10,000 cells/well in a 96-well plate. A uniform, reproducible wound was created using Incucyte, Essen Bioscience (USA). The 96-well plate was placed in the Incucyte ZOOM apparatus and the images of cell migration was captured every 2 h for up to a total of 48 h.

**RT-qPCR analysis.** Total RNA was extracted from snap-frozen fibrotic cardiac tissue and primary cardiac (myo)fibroblasts using the RNeasy mini kit (Qiagen, #74106) and cDNA was prepared using the iScript cDNA synthesis kit (primer specific, BIORAD, #170-8897) according to the manufacturer's instructions. Fast SYBR-Green master mix (BIORAD, #170-8880AP) was used for the analysis of gene expression using the BIORAD CFX RT- PCR system. The primers used in the experiment are listed the in Supplementary Table 2. 18S was used to normalize the relative gene expression and the $2^{-\Delta\Delta Ct}$ method was used to measure the fold change.

**Western blotting.** Protein extracts were isolated from heart tissue and cells using RIPA buffer (Thermofischer, #89900) supplemented with protease (Sigma Aldrich, #11836170001) and phosphatase inhibitors cocktails (ROCHE, #PHOSS-RO). Nuclear and cytoplasmic extracts were obtained using NE-PER kit (Pierce, #78833) according to the manufacturer's instructions. Co-immunoprecipitation was performed with the cell lysates subjected to different treatment conditions with Pierce Direct Magnetic IP/CO-IP kit (Pierce, #88828) according to the manufacturer's protocol. Immunoprecipitates were washed from conjugated beads and boiled in 5× SDS-PAGE buffer for further WB analysis. After quantification with the Bradford method, protein lysates were loaded onto a 4–12% acrylamide gel subjected to SDS-PAGE and then transferred onto a nitrocellulose membrane. After blocking in 5% nonfat dry milk, blotting was performed with anti-WWP2 targeting N-terminal region (Santa Cruz Biotechnology, #sc30052,1:500), anti-WWP2 targeting C-terminal region (Aviva Systems Biology, #ARP43089_P050,1:500), anti-TGFβ1 (Santa Cruz Biotechnology, #sc52893,1:500), anti-ACTA2 (Sigma Aldrich, #A5228,1:10,000), anti-S100A4 (Abcam, #ab41532, 1:500), anti-Vimentin (Abcam, #ab45939,1:500), anti-Periostin (Novus Bio, # NBP1-30042, 1:500), anti-Fibronectin (Sigma, #SAB4500974, 1:500), anti-p-SMAD2 (CST, #18338,1:500), anti-SMAD2/3 (CST, #3102, 1:500), anti-SMAD-4 (Santa Cruz Biotechnology, #sc-7966, 1:500), anti-Ubiquitin (CST, #3933, 1:500), and anti-FLAG (Sigma Aldrich, #F7425, 1:1000). Loading control was blotted with anti-tubulin (Sigma Aldrich, #T5168, 1:5000) and anti-GAPDH (Abcam, #ab8245, 1:5000). Anti-Lamin A/C (Abcam, #ab8984, 1:5000) and anti-PARP (Abcam, #ab6079,1:5000) were used as nuclear controls. Blots were visualized by labeling with anti-Rabbit HRP (Bethyl Laboratories, #A120-101P, 1:5000 or Thermo Fisher # 101023, 1:1000) and anti-Mouse HRP (Bethyl laboratories, #A90-116P, 1:5000) and developed on a Kodak automated developer with the ECL and Femto Detection Systems (Pierce) and quantified using densitometry with ImageJ (version 2.0.0-rc-43). The unprocessed scans of the western blots are shown in Supplementary Fig. 18.

**Statistical analyses.** Data are expressed as mean ± standard deviation (SD). The applied statistical tests were dependent on the number of groups being compared and the study design, and are detailed in each figure legend. Unless otherwise indicated, a two-tailed Mann–Whitney $U$ test was used to compare two groups, with * denoting $P < 0.05$, and ** denoting $P < 0.01$. When comparing mice groups with different genotypes, male littermate mice were assigned to the WT and Mut/Mut mice groups according to the results of genotyping (Supplementary Fig. 2) and mice with the same genotype were randomly assigned to the control, AngII infusion, or MI group using a simple random-sampling approach. All experiments requiring the use of animals, directly or as a source of cells, were subjected to randomization. The experimenters were blinded to the grouping information. All in vitro experiments were independently replicated at least three times as indicated in the figure legends.

**Reporting Summary.** Further information on research design is available in the Nature Research Reporting Summary linked to this article.

## Data availability

All the data generated in this study supporting the main findings have been deposited to NCBI's Gene Expression Omnibus (GEO) and accessible through GEO Series accession number GSE133017, including GSE130468 (bulk RNA-seq data from mouse heart) and GSE133015 (single-cell RNA-seq data from mouse heart). The rest of the data are available from the authors on reasonable request, please refer to author contributions for specific data sets.

## Code availability

We used published algorithms. See Methods section for full description of each analysis including input data, library, and algorithm version used, for which we provide corresponding web links.

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

## Acknowledgements

The research was primarily supported by National Medical Research Council (NMRC) Singapore grant NMRC/CBRG/0106/2016 (to E.P.) and the British Heart Foundation (BHF) Ph.D. Studentship grant FS/11/25/28740 (to E.P). We acknowledge additional funding support from European Union FP7 CardioNeT-ITN-289600 (to E.L.-P., S.A.C., and P.J.R.B.), Heart Research UK (to P.J.R.B.), NIHR CV BRU of Royal Brompton and Harefield, NHS Foundation Trust (to S.A.C. and P.J.R.B.), BHF (to S.A.C.), Leducq Foundation (to S.A.C.), MRC UK (to S.A.C.), BHF Program Grant no. RG/15/5/31446 (to C.E. and E.P.). M.P. was supported by Praemium Academiae award of the Czech Academy of Sciences and grant 14-36804G from the Czech Science Foundation. We wish to thank Dr. Jacques Behmoaras for contributing critical and constructive comments to the manuscript.

## Author contributions

E.P. conceived and supervised the study, obtained and managed funding for the project. Mouse animal and cell experiments were carried out by H.C. and N.D. with the assistance of E.T. A.M.-M. conceived and carried out Systems Genetics and Bioinformatics analyses with the assistance of M.R., P.K.S., N.H., K.S., O.J.R., S.S., and L.B. F.P. processed and carried out the network analyses in the DCM cohort with the assistance of A.M.-M. X.S. and S.A.C. generated human fibroblasts data. G.A., C.B., and C.E. collected human primary fibroblasts and carried out the experiments in human cells. M.M. generated rat histomorphometric data. M.P. provided rat data. E.L.H. generated human rTOF data. S.A.C., P.B., N.H., S.S., and L.E.F. provided DCM patient data. E.L.-P. generated heart cells data in mice. N.G.Z.T. carried out the mice heart echocardiography measurements. W.P.Y. generated the WWP2$^{Mut/wt}$ mouse. E.P., H.C., and A.M.-M. designed all analyses, experiments, and wrote the manuscript, which was critically revised by C.E. and S.A.C., and with inputs from co-authors.

## Additional information

**Competing interests:** E.P. and A.M.-M. have filed a patent (application number 1908544.8; status: "provisional application"; date 14 June 2019) concerning the treatment and/or prevention of disease through inhibition of the pro-fibrotic functions of WWP2. The remaining authors declare no competing interests.

