## [Peer Review File · Nature Communications]

Reviewers' comments:

Reviewer #1 (Remarks to the Author):

In this study, Chen et al. computationally predicted and experimentally validated WWP2 as a key regulator of a gene network associated with cardiac fibrosis. Specifically, they performed the well-established weighted gene coexpression network analysis (WGCNA) on cardiac gene expression data in human and rat to systematically identify gene modules associated with fibrosis levels. The cross-species conservation and differential coexpression analyses pinpointed an extracellular matrix enriched module (hECM-network) as the most important molecular network in dilated cardiomyopathy (DCM). A majority of the genes in the hECM-network are upregulated in both DCM and heart failure. They further performed network-eQTL mapping for the hECM-network in human DCM heart and identified a single regulatory SNP (rs9936589) within WWP2, suggesting that the hECM-network is potentially regulated by genetic variant within the WWP2 locus. WWP2 mutant mice using CRISPR/Cas9 technology were found to have improved cardiac function and reduced myocardial fibrosis. Subsequent investigation of cellular mechanisms revealed that WWP2 regulates the TGF β 1-induced fibrotic response in primary cardiac fibroblasts. Overall, this study presented some compelling evidence to demonstrate that WWP2 is a key regulator of pathological fibrosis though some additional analyses need be done to better understand the hECM-network and its other master regulators.

Major concerns:

1. The hECM module shares only about 10.5% of its member genes with its rat counterpart, suggesting the cross-species conservation analysis greatly limited the discovery of additional, potentially more important, modules and master regulators of cardiac fibrosis in human. More dedicated network and genetic analyses of the human cardiac gene expression data are required to present a comprehensive picture about all the WGCNA modules and their potential genetic regulators associated with cardiac fibrosis.
2. The WWP2 mutant signatures from bulk tissue and single cell RNA-seq data should be compared with the hECM module to validate the predicted role of WWP2 in regulating the hECM network. Moreover, the WWP2 mutant signatures should also be compared with all the other WGCNA modules to see how the WWP2 regulate other relevant pathways.
3. Since the network-eSNP (rs9936589) located within an intron of the WWP2 is not a risk factor for heart diseases by GWAS, the translational perspective of the discovery is questionable. The frequency of the mutations in this locus need be calculated using relevant human genetics data.

Minor concerns:

1. Was the Kruskal–Wallis test p-value reported in Figure 2G corrected for multiple testing?
2. It seems that the DEGs from Figure 3j are not enriched for extracellular matrix genes. The enrichment statistics from the overlap between the DEG signature and the ECM pathway should be provided.

Reviewer #2 (Remarks to the Author):

In this study, Chen and co-workers undertook a highly integrative systems genetics approach to identify a pro-fibrotic gene network in the diseased heart and show that the network is regulated by the N-terminal isoform of the E3 ubiquitin ligase WWP2. The strengths of the study include (1) its translational character with experiments being conducted in both pre-clinical models and in human (diseased) tissue; (2) the fact that the authors anchored the regulation of the profibrotic network to a region of the human genome allowing them to identify WWP2 as a regulator of this network and identify the direction of effect; (3) the delineation of a likely molecular mechanism of action of WWP2. However, the reviewer has some points that require attention or clarification.

1. Throughout the manuscript, and in particular the sections relating to the systems genetics, there needs to be an enhanced transparency regarding the statistics, in particular whether correction for multiple testing has been done at all the different steps. An expert review in statistics is highly recommended.

2 'This might explain why WWP2 passed undetected to GWAS and

other eQTL genetic mapping or cellular screening studies of fibrotic diseases.' Another reason for escaping detection in eQTL studies is that it seems that its expression is restricted to fibroblasts, and eQTL studies have thus far looked at heart tissue as a whole (containing both cardiomyocytes and fibroblasts). In this respect, this reviewer doubts whether this gene network is indeed absent in normal cardiac tissue. It could be that the authors are detecting it in diseased hearts because these hearts simply have a higher proportional representation of fibroblasts compared to controls, with the more diseased hearts containing more fibroblasts than the less diseased ones. This may confound the analysis. Please explain which measures were taken to take care of this.

Minor comments

3. Line 377: (siRNA-C') regions of the WWP2 genomic RNA; what do you mean by 'genomic RNA'?

4. Additional work in human cardiac fibroblasts provided further evidence that WWP2-385 N/FL knockdown is able to reduce pro-fibrotic gene expression (Fig. 4j). Please explain in the text what additional work you mean.

Reviewer #3 (Remarks to the Author):

Nature Communications "WWP2 regulates pathological cardiac fibrosis by modulating SMAD2 signaling" by Chen H, Moreno-Moral A, Pesce F, et al.

Strengths

- Chen et al. use both murine and human specimens for pathological analyses; the genetic analyses were thoroughly executed and implement state-of-the-art approaches (ie. GSEA and RNA-seq) with the two models.
- A thorough, mechanistic approach was taken to confirm the results from GSEA. The mouse model provided greater understanding of SMAD2 shuttling in primary cardiac fibroblasts.
- The identification of WWP2 as a novel player in cardiac fibrosis is significant, as it has become evident that SMAD proteins are subject to much more nuanced levels of regulation that goes well beyond canonical TGF- β signaling.
- The manuscript itself is clear and well-written, with few grammatical/typographical errors. The discussion is frank and does not attempt to embellish the results of the study. The majority of the results present by the authors is of good quality and serve to support their claims.

- Appropriate controls were used when needed; normalizations are adequate. Statistical tests are valid for the complexity of the data sets used

Major weaknesses/corrections

- There is a marked tendency by the authors to rely solely on alpha smooth muscle actin (ACTA2/alpha-SMA) as a marker of fibroblast activation or the existence of myofibroblasts. It is known that ACTA2 is not a reliable marker of the myofibroblast phenotype on face-value: rather, it is more useful when it is examined in context as to whether or not it is incorporated into stress fibers (as shown in Figure 4d). It should also be noted that ACTA2 is also a driver of EMT and phenotypic modulation in mesenchymal cells; so using it as the sole marker may not be truly indicative of the degree of phenotype activation.
- Suggestions of markers to add to the panel for Westerns: Fibronectin extracellular domain A (ED-A or “cellular fibronectin”; see Gulio Gabbiani’s work); periostin (see Jeff Molkentin’s work); SMemb/Non-muscle myosin IIB (work by Nikolaos Frangogiannis), Thy-1/CD90 (work of Gaétan Thibault).
- As the primary cells are isolated from a heterogeneous population, it is recommended that Western blots also be probed for Vimentin as a phenotype control for cells of mesenchymal origin.
- Suggestions of markers to add to panel for qPCR: TCF21, Postn, ED-A fibronectin, CTGF. While not all indicative of fibroblast vs myofibroblast phenotype, it would at least give the audience a better sense of what sub-population of cells the authors were observing.
- The cell culture method for primary cardiac fibroblasts suggests that the group used myofibroblasts, rather than fibroblasts. First, the cells were cultured in DMEM with an excess of serum. Second, the cells were likely cultured on stiff plastic, as the elastic modulus of the culture surface was not indicated (eg. 5 kPa). Third, the cells were allowed to reach 80-90% confluency, which greatly affects the cell phenotype by contact inhibition. Lastly, the cells were cultured for 10 days in these conditions, which would only further promote the pro-fibrotic, myofibroblast phenotype.

- It is well known that conventional cell culture methods are not conducive to the maintenance of a quiescent phenotype. (See Santiago et al. Dev Dyn. 2010 Jun;239(6):1573-84)
- The authors should caution in the discussion that their in vitro results may not have been as evident as the in vivo results as they were limited by their cell culture methods.
- Eg. ACTA2 was always highly-expressed (Fig. 4C, I, M) and the addition of TGF- β yielded modest results.
- The cells were already exhibiting a pro-fibrotic phenotype, thus the treatments did not generate a robust response.

Minor weaknesses/corrections

- Figure 4 is very difficult to follow—perhaps re-organizing the figure or removing some of the diagrams/schematics to make it less dense? In its current form, it is difficult to identify individual components in the figure.
- Line 535: It should say “ARKADIA” to correct “AIKADIA”

Questions of General Interest

- Is WWP2 affected by HECT E3 ubiquitin ligase inhibitors, such as Heclin?

Reviewer #4 (Remarks to the Author):

Chen et al reported here that they have identified a cross-species (human and mouse) profibrotic extracellular matrix (ECM) co-expression gene network in diseased human hearts and mapped a common regulatory cis-element of this network to a gene encoding WWP2, a E3 ubiquitin ligase, particularly to its N-terminal isoform. They went on to show that mice mutant for WWP2 had improved cardiac function and reduced myocardial fibrosis in response to pressure overload, and attributed this regulation to WWP2N becoming nuclear bound under the influence of TGF- β , where it interacts with SMAD2 and promotes SMAD2 mono-ubiquitination.

Overall I find this manuscript very difficult to follow, not because I am not an expert in systems biology, many of their conclusions are not supported experimentally, and some of their experimental methods/designs are outright flawed. Although they have presented an impressive body of experimental data, I simply don't see a coherent story or cannot be enthusiastic in supporting its acceptance for publication.

Specific issues.

1. Based on analyses of RNA-seq data, the authors reported only very small increases of WWP2 expression as low as 1.02 fold in the diseased hearts. Can this be independently verified by qRT-PCR on available samples? I understand the statistical power applied to large data set, but any outcome conclusion has to withstand scrutiny of an independent method.
2. Some of their analyses are very difficult to evaluate, e.g. the authors concluded that only WWP2N was regulated by SNP associated with that gene, but their PCR design for measuring the individual isoforms are flawed. First of all, those "P1, P2, P3" designation in Fig.3a should really refer to PCR products, not the primers, as so stated in the text, which is very confusing and wrong. The drawing in the figure lacks sufficient detail to show how these PCR products could be exon-specific, therefore isoform specific. As they are, P1 is to both N and full length forms, P2 is full length only, whereas P3 can be both full length and C isoform.
3. The 4 bp deletion introduced by CRISPR/Cas9 is expected to cause frameshift mutation that disrupt the expression of full length as well as the N isoforms, the same consequence as in the "global" null mutation reported by others previously. In both cases, the "C" isoform could have been preserved. As such it is not surprising that the new mutants essentially phenocopied the old ones. However, I'd expect the absence of both the full length and the N-isoform by WB, not a mere truncation that only slightly shortened the full length protein as reported in all WB figures. So, none of the WB results can be trusted!
4. Related to above, the immunofluorescence results are also questionable.
5. The authors designed siRNAs against 5' or 3' end regions of WWP2 and claim that these siRNAs decreased FL/N or FL/C isoforms. As pointed above, the WB analysis cannot be trusted, but this design is also flawed.

6. Since all experimental manipulations affect both N and full length WWP2, no demonstration was made to verify that marginal increase of WWP2-N expression could cause any phenotype in either cell culture or animal experiments.
7. Fig. 5e, I could not see any difference in the levels of SMAD2 monoubiquitination between WT and Mut/Mut cells based on the blot the authors provided.
8. Authors showed in Fig. 5g and Fig. 5h that SMAD2 nuclear accumulation was enhanced or cytoplasmic export was delayed, (again I did not see any meaningful changes in SMAD4 nuclear accumulation in Fig. 5g). However, the transcriptional reporter assay indicated that SMAD transcriptional activity was actually decreased, how could this be? Is it possible due to the fact that WWP2 actually affect SMAD7 in these conditions? Regulation of SMAD7 by WWP2 was reported previously. The authors should look into this possibility in their manuscript.
9. In theory, mono-ubiquitination would disrupt SMAD2/3 interaction with SMAD4, or disrupt SMAD3 binding to DNA, thus hampering the transcriptional activities of SMADs. As such, WWP2 via its ubiquitin E3 ligase activity should negatively regulate TGF- β signaling. Moreover, WWP2N lacks the HECT domain, therefore the ligase activity. If the regulation is mediated by the N-isoform, as the authors implied, it cannot be done through a direct ubiquitin modification. As is, this manuscript is not clear at all on the nature of N isoform function.

Pont-by-point rebuttal

Reviewer #1 (Remarks to the Author):

In this study, Chen et al. computationally predicted and experimentally validated WWP2 as a key regulator of a gene network associated with cardiac fibrosis. Specifically, they performed the well-established weighted gene coexpression network analysis (WGCNA) on cardiac gene expression data in human and rat to systematically identify gene modules associated with fibrosis levels. The cross-species conservation and differential coexpression analyses pinpointed an extracellular matrix enriched module (hECM-network) as the most important molecular network in dilated cardiomyopathy (DCM). A majority of the genes in the hECM-network are upregulated in both DCM and heart failure. They further performed network-eQTL mapping for the hECM-network in human DCM heart and identified a single regulatory SNP (rs9936589) within WWP2, suggesting that the hECM-network is potentially regulated by genetic variant within the WWP2 locus. WWP2 mutant mice using CRISPR/Cas9 technology were found to have improved cardiac function and reduced myocardial fibrosis. Subsequent investigation of cellular mechanisms revealed that WWP2 regulates the TGFβ1-induced fibrotic response in primary cardiac fibroblasts. Overall, this study presented some compelling evidence to demonstrate that WWP2 is a key regulator of pathological fibrosis though some additional analyses need be done to better understand the hECM-network and its other master regulators.

Major concerns:

1. The hECM module shares only about 10.5% of its member genes with its rat counterpart, suggesting the cross-species conservation analysis greatly limited the discovery of additional, potentially more important, modules and master regulators of cardiac fibrosis in human. More dedicated network and genetic analyses of the human cardiac gene expression data are required to present a comprehensive picture about all the WGCNA modules and their potential genetic regulators associated with cardiac fibrosis.

We thank the reviewer for raising this point. For clarity and space constraints, in the main text of the manuscript, we focused only on the network with strongest rat-human conservation and replicated genetic mapping in both species (i.e., the hECM-network). However, in our analysis we had found five human networks that met our first selection criteria: (1) conserved in the rat and intersecting a rat network associated with fibrosis, (2) differentially co-expressed in DCM. These human modules were: Hs M28, Hs M39, Hs M44, Hs M47 -hECM-network- and Hs M48, and they turned out to be intersecting with only three fibrosis-associated rat networks (M1 -rat ECM-network-, M2 -an *Insulin signaling pathway enriched network*- (M2) and M12 -an *Adherence junctions enriched network*-). These analyses suggested the presence of other networks potentially associated with fibrosis and with genetic regulators in the rat. We carried out genetic mapping in both rat and human. We did not find any control point in the rat genome for the rat network M12, but we found a signal for M12 (locus in rat chromosome 4q12, spanning 4.39Mb). However, this locus did not replicate in human. While these results are potentially of interest, we did not follow them up, as they were not replicated in humans. Nevertheless, our analyses have been carried out in all the networks and we agree with the reviewer that this information can be of use to the community. Therefore, in the revised version of the manuscript we have added an **Extended Analyses section (in supplementary)** with the full results for all the networks and some additional analyses including: (1) Enrichment of networks for TGFβ1 and TGFβ2 differentially expressed genes in cardiac fibroblasts (Extended Supplementary Figure S1), (3) conservation of the networks in right ventricle rTOF patients and control samples (Extended Supplementary Figure S2), (2) genetic mapping results of other networks that are also associated with fibrosis (Extended Supplementary Figure S3 and S4), (4) fibroblast and myocyte expression enrichment (Supplementary Figure S13). We believe these additional data will be useful to other investigators who are interested in following up other cardiac gene networks identified in our study.

We also agree with the reviewer that cross-species conservation could potentially limit the discovery of relevant regulators of human cardiac fibrosis. However, in this study there are two main reasons to start from the rat system: (1) this study is focused on cardiac fibrosis and unfortunately, we only had quantitative histomorphometrical measurements of interstitial and perivascular fibrosis in the rat heart, (2) genetic mapping in the rat panel yields increased statistical power than solely mapping in human. The latter has been previously shown in several studies in which we started first from rodents mapping populations (i.e., mapped the *trans*-acting genetic regulator of the network in rodents) and we then successfully translated the findings (i.e., the *trans*-regulator of the network) to humans [1-4]. We believe this is a key advantage of our approach, as it allows us to narrow down the human mapping to specific loci (rather than going genome-wide), which is important since with a cohort of only 96 genotyped disease patients we have limited power for the discovery of *trans*-acting regulators of networks and *trans*-eQTLs in general.

2. The WWP2 mutant signatures from bulk tissue and single cell RNA-seq data should be compared with the hECM module to validate the predicted role of WWP2 in regulating the hECM network. Moreover, the WWP2 mutant signatures should also be compared with all the other WGCNA modules to see how the WWP2 regulate other relevant pathways.

We have now carried out this analysis and tested the enrichment of all human modules (including the hECM network) in the WWP2 mutant signatures profiled from bulk tissue. See the full list of results in the following table.

Human modules enriched in Wwp2 mouse loss of function differentially expressed genes (FDR<0.05)	Human module size in mouse (considering only genes with one-one human-mouse ortholog relationship)	NES	FDR	P or N enrichment
Hs-M44	1500	-5.54	<10E-6	N
Hs-M35	225	-3.80	<10E-6	N
Hs-M47 (hECM-network)	430	-3.68	<10E-6	N
Hs-M23	102	-3.36	<10E-6	N
Hs-M6	84	-3.02	<10E-6	N
Hs-M43	120	-2.83	<10E-6	N
Hs-M48	49	-2.72	0.0001	N
Hs-M36	64	-2.32	0.0023	N
Hs-M27	36	-1.82	0.04	N
Hs-M32	189	-1.79	0.04	N
Hs-M3	910	8.77	<10E-6	P
Hs-M34	152	4.30	<10E-6	P
Hs-M15	34	3.41	<10E-6	P
Hs-M2	228	3.21	<10E-6	P
Hs-M19	111	3.07	0.00001	P
Hs-M17	288	3.03	0.00001	P
Hs-M40	40	3.01	0.00001	P
Hs-M25	99	2.87	0.00002	P
Hs-M24	16	2.27	0.00211	P
Hs-M9	45	1.78	0.03721	P
Hs-M18	137	1.77	0.03517	P

In this table, negative and positive Normalized Enrichment Score (NES) denotes enrichment for genes down- and up-regulated in the mutant. We found that the hECM-network is one of the top downregulated modules in the mutant. The hECM-network (Hs-M47) had the third highest negative NES after Hs-M44 (top pathway enriched in this module: “Ubiquitin mediated proteolysis”) and Hs-M35 (top pathway enriched in this module: “Regulation of actin cytoskeleton”). The top up-regulated modules were Hs-M3 (top pathway enriched in this module: “Oxidative phosphorylation”), Hs-M34 (top pathway enriched in this module: “Ribosome”) and Hs-M15 (top pathway enriched in this module: “Cell cycle”). We now provide these additional results in **Supplementary Table S5**. These findings (above) therefore confirm and strengthen the evidence in support of the regulation of the hECM-network by WWP2 and, as suggested by the reviewer, they shed light into other processes undergoing in the mutant. The processes highlighted by this analysis also agree with the pathways found up- and down-regulated in WWP2 mutant signatures profiled from bulk tissue presented in **Figure 3j**.

Unfortunately, we cannot carry this analysis in the single cell RNA-seq data, as we do not have single-cell RNA-seq data for the WWP2 mutant. The new data and analyses of the networks are reported in **Supplementary Table S5**, and commented in the main text.

3. Since the network-eSNP (rs9936589) located within an intron of the WWP2 is not a risk factor for heart diseases by GWAS, the translational perspective of the discovery is questionable. The frequency of the mutations in this locus need be calculated using relevant human genetics data.

We thank the reviewer for raising this point as it was not clear and we agree (and now clarify) that the network-SNP is not a risk factor for heart disease. We have carried out a thorough search for cardiovascular disease mutations in the locus including (among others) the following databases: The Cardiovascular Disease Knowledge Portal (database with genetic information linked to myocardial infarction, atrial fibrillation, and related traits, <http://broadcvdi.org/home/>), GWAS central (<https://www.gwascentral.org/>), OMIM (<https://www.omim.org/>), dbSNP Short Genetic Variations (<https://www.ncbi.nlm.nih.gov/snp/>), the human gene mutation database (<http://www.hgmd.cf.ac.uk>), the Databases of genomic variation and Phenotype in Humans using Ensembl Resources, DECIPHER (<https://decipher.sanger.ac.uk>). We have only found some evidence of high impact variants at the WWP2 locus for triglycerides, (see the variants below) and other cardiovascular traits (BMI and Type 2 diabetes).

High-risk variants for triglycerides:

Variant ID	dbSNP ID	Major allele	Minor allele	Predicted impact	Protein change	p-Value	Effect	MAF	Gene	Data set
16_69832662_C_T	rs150059700	C	T	missense	p.P50S	0.154	0.136	0.00019	WWP2	300K exome chip analysis
16_69905808_G_A	rs113797383	G	A	missense	p.G226D	0.305	-0.0925	0.00022	WWP2	300K exome chip analysis
16_69905721_C_T	rs143156114	C	T	missense	p.S197L	0.423	0.0420	0.00065	WWP2	300K exome chip analysis
16_69973023_A_G	rs150082808	A	G	missense	p.I813V	0.432	0.0759	0.00019	WWP2	300K exome chip analysis

We have not found any evidence for variants or mutations associated to any heart disease, including myocardial infarction or heart failure. In fact, the role of fibrosis in heart disease is complex as it is a finely tuned system that plays a dual role: fibrosis is necessary for the initial healing of the heart but, when this healing becomes active over time, can lead to over production of ECM components that compromise heart function. Future human studies should focus on sequencing patients after measuring fibrogenic levels. However, to date there are no successful human genetic studies focusing exclusively on heart fibrosis.

Nevertheless, here we want to clarify that (in our opinion) the translational perspective of this study does not come from genetic variation in the WWP2 locus or the regulatory SNP. While we explored the potential role of the regulatory SNP (and other SNPs) at the WWP2 locus and found no obvious links with common heart diseases, we believe that translational interest of this study arises from the fact that the WWP2 gene is a positive regulator of fibrosis in the heart and that the WWP2 protein is a druggable target, potentially to prevent progression to heart failure and improve heart function. In the manuscript (see Discussion), we refer specifically to the WWP2 gene (*not* the regulatory SNP) as a novel antifibrotic and druggable target [5], with the potential to control fibrosis in pathological cardiac remodeling.

Minor concerns:

1. Was the Kruskal–Wallis test p-value reported in Figure 2G corrected for multiple testing?

We did not correct the Kruskal-Wallis test for multiple testing, as the aim of this boxplot was only to show the direction of the effect in relation to the genotype. We clarify that we did not use this to discover this SNP as *trans*-regulator of the network. The discovery of this SNP as regulator of the network was done by using our multivariate Bayesian method, which allows boosting the power to detect *trans*-eQTL effects [6, 7]. Typically, *trans*-eQTLs have small effect sizes and therefore they would not be detected by other, more traditional methodologies (i.e., a Kruskal-Wallis test with multiple correction), as they would not reach significance after multiple testing correction.

2. It seems that the DEGs from Figure 3j are not enriched for extracellular matrix genes. The enrichment statistics from the overlap between the DEG signature and the ECM pathway should be provided.

The DEGs are also enriched for the ECM pathway but we left these results in the **Supplementary Table S5**. Due to space constraints and to avoid showing only pre-selected relevant enriched terms, we chose to show the full list of results for a single database (we chose hallmark gene sets as it had less redundancy and a good coverage of relevant processes). Unfortunately, the hallmark database does not cover the ECM pathway and this is why this pathway was missing in the figure. We agree with the reviewer that it gave the impression that the DEGs were not enriched for the ECM pathway, this is why we have added this into the text and specifically referred to the Supplementary where this was initially reported, as follows: “*TGFβ* signaling” and “*Extracellular matrix*” were two major downregulated pathways in *WWP2^{Mut/Mut}* mice following *Ang II*-infusion, (Normalized Enrichment Score (NES) of -2.71 and -2.8, **Fig. 3i** and **Supplementary Table S5a**, respectively). Also, we keep the full list of enriched terms in **Supplementary Table S5**.

Reviewer #2 (Remarks to the Author):

In this study, Chen and co-workers undertook a highly integrative systems genetics approach to identify a pro-fibrotic gene network in the diseased heart and show that the network is regulated by the N-terminal isoform of the E3 ubiquitin ligase WWP2. The strengths of the study include (1) its translational character with experiments being conducted in both pre-clinical models and in human (diseased) tissue; (2) the fact that the authors anchored the regulation of the profibrotic network to a region of the human genome allowing them to identify WWP2 as a regulator of this network and identify the direction of effect; (3) the delineation of a likely molecular mechanism of action of WWP2. However, the reviewer has some points that require attention or clarification.

1. Throughout the manuscript, and in particular the sections relating to the systems genetics, there needs to be an enhanced transparency regarding the statistics, in particular whether correction for multiple testing has been done at all the different steps. An expert review in statistics is highly recommended.

We thank the Reviewer for pointing out the need to more clarity in the statistics; we agree that this information was not clearly stated in the previous version of the manuscript. Now we have thoroughly revised the text and added these details whenever statistical tests are carried out (see the changes highlighted in the main text). In addition, in the supplementary material we have added full details of the tests carried out, backgrounds and multiple testing corrections used in each case.

2 'This might explain why WWP2 passed undetected to GWAS and other eQTL genetic mapping or cellular screening studies of fibrotic diseases.' Another reason for escaping detection in eQTL studies is that it seems that its expression is restricted to fibroblasts, and eQTL studies have this far looked at heart tissue as a whole (containing both cardiomyocytes and fibroblasts).

Our immunofluorescence experiments and single cell data results (Figure 4a-b, and below) consistently show that WWP2 is not expressed in cardiomyocytes. In addition, single cell data show WWP2 expression in a range of FSP1 positive cells including endothelial and immune cells. These data indicate that WWP2 expression is not restricted to fibroblasts.

Figure legend. (4a) Immunofluorescence images of LV section staining from WT LV with Ang II showed that WWP2 (green, arrow) is expressed in non-myocytes (Left), and co-localized with part of FSP1 positive cells (red, arrow; Right). Scale bar: 40 μm . (4b) t-SNE displaying single-cell RNA-seq data in the LV of WT mice after Ang II infusion. A total of 508 cells were detected. Each dot corresponds to a single cell, which is coloured by WWP2 expression level (Log_2 normalized counts). Cells belonging to the different subpopulations identified in these mouse heart data are indicated.

In this respect, this reviewer doubts whether this gene network is indeed absent in normal cardiac tissue. It could be that the authors are detecting it in diseased hearts because these hearts simply have a higher proportional representation of fibroblasts compared to controls, with the more diseased hearts containing more fibroblasts than the less diseased ones. This may confound the analysis. Please explain which measures were taken to take care of this.

We thank the Reviewer for raising this point. We agree with the Reviewer that the relatively different cellular composition in the disease and control heart tissues could be one of the reasons for the detection of the network and this might also have affected the detection of the network's genetic control points. This is an important point and highlights a limitation of bulk tissue studies.

To investigate this, we inspected the relationship between gene expression level and our ability to detect gene co-expression in the hECM network, separately in the DCM and the control samples. We could not find any association between higher expression and increased gene co-expression in the network in either the DCM or control samples (the Spearman's ranked correlation coefficient was 0.054 and 0.162 in the DCM

and controls cohorts respectively). So this factor (network genes' expression levels in DCM and control hearts) does not appear to affect the detection of the network.

We went on and inspected data from two of the main subpopulations of cells in the heart, fibroblasts and cardiomyocytes, and carried out an additional analysis to investigate the possible “cellular origin” (i.e., fibroblasts or cardiomyocytes “origin”) of all human gene co-expression modules detected in our study. To this aim, for each gene, we (first) to define fibroblasts- or cardiomyocytes-expressed genes, computed the Log₂ fold change of expression between primary fibroblasts and cardiomyocytes (i.e. for each gene a fold change is computed between fibroblast and cardiomyocytes expression levels), (second) ranked all genes by this fold change and (third) ran a formal Gene Set Enrichment Analysis (GSEA) test to assess whether each of the human modules was enriched for fibroblast-like or cardiomyocyte-like genes. Figure below shows the results where we can see that the hECM-network (Hs-M47) (regulated by *WWP2*) is the top enriched gene module for containing fibroblasts like genes. In keeping with the Reviewer hypothesis, this suggests that the DCM-associated hECM-network is indeed enriched for fibroblasts genes (see Figure below, left hand side). Other networks show similarly high enrichment for cardiomyocyte-like genes (see Figure below, right hand side).

This is important observation does not come completely unexpected as diseased tissues (especially in chronic conditions such as DCM) underwent strong remodeling and the relative cellular composition is also affected, in this case with fibroblasts and myofibroblasts having a different contribution to different networks. It is possible that the high fibroblast-like gene enrichment observed in Hs-M47 might contribute to our ability to map the genetic regulation of the network to the *WWP2* locus, but we doubt this is major confounder in the analysis as this specific gee network was (i) significantly conserved between rat and human heart, and (ii) significantly conserved across two different human heart conditions and tissue types, and perhaps more importantly here, (iii) the *trans*-acting genetic regulation of the network by the *WWP2* locus was also conserved between rat and human heart as none of the other rat loci regulated this network in either rat or human heart.

We consider this observation and the analysis shown in the figure below useful and we thank the Reviewer for raising this point and suggesting this additional analysis of the networks. We now report these results in the revised manuscript (**Supplementary Figure S13**) and discuss their implications in the Discussion.

Minor comments

3. Line 377: (siRNA-C') regions of the *WWP2* genomic RNA; what do you mean by 'genomic RNA'?

Thanks for spotting this type, which has been corrected to “*Wwp2* mRNA” (Line 385).

4. Additional work in human cardiac fibroblasts provided further evidence that *WWP2*-N/FL knockdown is able to reduce pro-fibrotic gene expression (Fig. 4j). Please explain in the text what additional work you mean.

We have now clarified in the text that we referred to “Additional siRNA targeting 5'-terminal of *WWP2* mRNA (siRNA-*WWP*-N') experiments in primary human cardiac fibroblasts confirmed that *WWP2*-N/FL knockdown”. (Line 393-394)

Reviewer #3 (Remarks to the Author):

Strengths

- *Chen et al. use both murine and human specimens for pathological analyses; the genetic analyses were thoroughly executed and implement state-of-the-art approaches (ie. GSEA and RNA-seq) with the two models.*
- *A thorough, mechanistic approach was taken to confirm the results from GSEA. The mouse model provided greater understanding of SMAD2 shuttling in primary cardiac fibroblasts.*
- *The identification of WWP2 as a novel player in cardiac fibrosis is significant, as it has become evident that SMAD proteins are subject to much more nuanced levels of regulation that goes well beyond canonical TGF- β signaling.*
- *The manuscript itself is clear and well-written, with few grammatical/typographical errors. The discussion is frank and does not attempt to embellish the results of the study. The majority of the results present by the authors is of good quality and serve to support their claims.*
- *Appropriate controls were used when needed; normalizations are adequate. Statistical tests are valid for the complexity of the data sets used*

Major weaknesses/corrections

- *There is a marked tendency by the authors to rely solely on alpha smooth muscle actin (ACTA2/alpha-SMA) as a marker of fibroblast activation or the existence of myofibroblasts. It is known that ACTA2 is not a reliable marker of the myofibroblast phenotype on face-value: rather, it is more useful when it is examined in context as to whether or not it is incorporated into stress fibers (as shown in Figure 4d). It should also be noted that ACTA2 is also a driver of EMT and phenotypic modulation in mesenchymal cells; so using it as the sole marker may not be truly indicative of the degree of phenotype activation.*

We thank the Reviewer for raising this point. We agree that relying on ACTA2 alone is not sufficient despite it being a well-recognized marker of fibroblast activation and myofibroblast existence. Our data analysis revealed that the ECM network of genes regulated by WWP2 was enriched for genes differentially expressed after fibroblast activation with TGF β , therefore being relevant for ECM activation (**Figure 1g** and **Extended Supplementary Figure S1**). The hECM-network was also significantly enriched for DE genes between Ang II-treated WT and WWP2^{Mut/Mut} mice (**Supplementary Table 5b**). "TGF β signaling" and "extracellular matrix" were two of the major downregulated pathways in WWP2^{Mut/Mut} mice following Ang II-infusion, (Normalized Enrichment Score (NES) of -2.71 and -2.8, **Fig. 3j** and **Supplementary Table S5a**, respectively). In the primary cardiac fibroblasts, we also show that the TGF β 1-stimulated WT fibroblasts presented a clear organization of ACTA2 into stress fibers, while WWP2^{Mut/Mut}-derived cells showed a diffuse expression of ACTA2 with rare incorporation into stress fibers (**Fig. 4f**).

We thank the reviewer for his/her comment and now we have carried out additional experiments and analyzed more markers both *in vivo* and *in vitro* to understand the fibroblast activation and ECM protein synthesis regulated by WWP2. The details are showed in the following.

- **Suggestions of markers to add to the panel for Westerns: Fibronectin extracellular domain A (ED-A or "cellular fibronectin"; see Gulio Gabbiani's work); periostin (see Jeff Molkentin's work); SMemb/Non-muscle myosin IIB (work by Nikolaos Frangogiannis), Thy-1/CD90 (work of Gaétan Thibault).**

Thank you very much for this constructive suggestion. TGF β 1 enhances the synthesis of extracellular matrix (ECM) molecules including ED-A fibronectin (FN-EDA), an isoform de novo expressed during wound healing and fibrotic changes [8]. Periostin (POSTN) is predominantly expressed in collagen-rich fibrous connective tissues that are subjected to constant mechanical stresses in hearts [9, 10]. Following this suggestion, we have measured by western blot the protein expression of FN-EDA and POSTN in heart after Ang II infusion (**Fig. 3m**). As expected, Ang II-infusion resulted in increased expression of POSTN and FN-EDA in heart tissue, especially POSTN. Moreover, WWP2^{Mut/Mut} mice produced significant less POSTN (P=0.040) and relatively less although no significant change in FN-EDA (P=0.182) in the heart after Ang II-infusion. These data are now included in the revised manuscript (**Fig. 3m**).

Figure legend.

Figure 3m. Representative western blot showing Fibronectin extracellular domain A (FN-EDA, ~220 kD) and Periostin (POSTN, ~94 kD) in the heart of WT and WWP2^{Mut/Mut} mice following Ang II infusion.

- As the primary cells are isolated from a heterogeneous population, it is recommended that Western blots also be probed for Vimentin as a phenotype control for cells of mesenchymal origin.

We thank the Reviewer for raising this point. Vimentin has been extensively used to label cardiac fibroblasts [11]. Santiago *et al* [12] showed that vimentin was expressed in cultured primary cardiac fibroblasts and even increased fibroblast-myofibroblast conversion in cells from passages P0 to P3.

We measured the expression of Vimentin in our primary cardiac fibroblasts. We cultured P1 cardiac fibroblasts and detected the expression of in the both WT and WWP2^{Mut/Mut} quiescent cells. After treatment with TGFβ1 for 72 hrs, Western Blot assay and fluorescence staining showed the expression vimentin protein was slightly increased in WT cells, but this increase was not clear in WWP2^{Mut/Mut} cells. However, Vimentin in mRNA levels were similar between WT and WWP2^{Mut/Mut} cardiac fibroblasts whether with or without TGFβ1 stimulation.

Taking Vimentin as a marker, dysfunction of WWP2 protected the conversion of fibroblast to myofibroblast after TGFβ1 treatment. We have added these results in the revised version of our manuscript (**Supplementary Figure S8a-c**).

Supplementary Figure S8. Vimentin expression level in primary cardiac fibroblasts (WT and WWP2^{Mut/Mut}). (a) Western blot of Vimentin in primary cardiac fibroblasts revealed that TGFβ1 treatment (5ng/ml, 72h) relatively increased the protein levels of vimentin, which was prevented in the WWP2^{Mut/Mut} cells. Representative image (top) and corresponding relative levels of Vimentin (down) (Mann-Whitney U test, n=3 for each group; means ± SD) (b) Representative immunostaining showing Vimentin in cardiac fibroblasts (WT and WWP2^{Mut/Mut}) after TGFβ1 treatment (5ng/ml, 72h). (c) Relative mRNA expression of vimentin in cardiac fibroblast (WT and WWP2^{Mut/Mut}) after TGFβ1 treatment (5ng/ml, 72h). mRNA expression was normalized to 18S level. (Mann-Whitney U test, n=5 for each group; means ± SD)

- **Suggestions of markers to add to panel for qPCR: *TCF21*, *Postn*, *ED-A fibronectin*, *CTGF*. While not all indicative of fibroblast vs myofibroblast phenotype, it would at least give the audience a better sense of what sub-population of cells the authors were observing.**

Following this suggestion, we measured the mRNA expression of these four genes (*Tcf21*, *Postn*, *Fn1* and *Ctgf*) and added the data in our qPCR panel. *Postn*, ED-A fibronectin (*Fn1*) and *Ctgf* are all extracellular matrix associated proteins that are upregulated by TGF β 1 [8-10, 13], while *TCF21* is a transcript factor considered essential for the development of cardiac fibroblasts [14, 15]. TGF β 1 stimulation significantly increased the expression of *Postn*, *Fn1* and *Ctgf* in mRNA levels in cardiac fibroblasts, and the dysfunction of WWP2 protected this effect. Regarding *Tcf21*, TGF β 1 treatment did not change the mRNA levels of *Tcf21* in cardiac fibroblast, which was similar between WT and WWP2^{Mut/Mut} cells whether with or without TGF β 1 stimulation. This new data has been added in the revised version of the manuscript (**Figure 4d**).

Figure legend

Figure 4d. Relative mRNA expression of genes referred to ECM and fibroblast markers in primary cardiac fibroblasts, including *Tcf21*, *Tcf21*, *Postn* and *Fn1*. mRNA expression was normalized to the level of 18S. Mann-Whitney U test, n=5-6 for each group; means \pm SD.

- **The cell culture method for primary cardiac fibroblasts suggests that the group used myofibroblasts, rather than fibroblasts. First, the cells were cultured in DMEM with an excess of serum. Second, the cells were likely cultured on stiff plastic, as the elastic modulus of the culture surface was not indicated (eg. 5 kPa). Third, the cells were allowed to reach 80-90% confluency, which greatly affects the cell phenotype by contact inhibition. Lastly, the cells were cultured for 10 days in these conditions, which would only further promote the pro-fibrotic, myofibroblast phenotype.**

- **It is well known that conventional cell culture methods are not conducive to the maintenance of a quiescent phenotype. (See Santiago et al. Dev Dyn. 2010 Jun;239(6):1573-84)**

- **The authors should caution in the discussion that their in vitro results may not have been as evident as the in vivo results as they were limited by their cell culture methods. Eg. ACTA2 was always highly-expressed (Fig. 4C, I, M) and the addition of TGF- β yielded modest results.**

- **The cells were already exhibiting a pro-fibrotic phenotype, thus the treatments did not generate a robust response.**

We thank the reviewer for raising this point. In the method used by Santiago et al and cited by the reviewer [12], the ventricular cells were isolated after Langendorff perfusion and collagenase digestion for 20–25 min, resuspended in fresh medium containing 10% FBS and plated on 75 cm² culture flasks for ~3 hr. The adherent cells mainly were fibroblasts labeled as P0 and the cells were passaged three times accordingly (P1–P3) for experiments.

In our studies, we prepared cardiac fibroblasts from mouse ventricle following another published protocol [16]. In detail, minced LV pieces (1-3 mm³) were placed in 6-cm dishes with DMEM supplemented with 20% FBS. The medium was renewed every 2-3 days. The fibroblasts began to crawl out from the tissues at ~5 days and reached to 80-90% confluence around the tissues up to ~10 days. These cells were labeled as P0 and passaged two times accordingly (P1–P2) for experiments.

In Santiago et al's protocol [12], cardiac fibroblasts (P0) are collected only 3 hrs after digestion. But, in the method we used, the cardiac fibroblasts (P0) began to crawl out from the tissues only at around 5 days after seeding the tissues. The cells gradually came out from each tissue piece, and then reached 80-90% confluence around the remaining tissue in the following 3-5 days. Thus, this method of cell culture takes a long time (~ 10 days) and the cells were allowed to reach local 80-90% confluence surrounding the tissues. In our experience, DMEM supplemented with 20% FBS is better to promote cell crawling compared with 10% FBS. But, we did not pay much attention to the elastic modulus of the culture surface as the elastic modulus of the culture surface seemed normal, so we did not indicate it.

We agree that some of the cardiac fibroblasts that we cultured and used for experiments had been already converted into myofibroblasts before the TGF β 1 treatment. It is not conducive to the maintenance of a

quiescent cardiac fibroblasts phenotype with currently used cell culture methods [12]. We detected the expression of ACTA and vimentin in cardiac fibroblasts before the TGF β 1 treatment and the TGF β stimulation yielded only modest results.

We considered this during all our experiments. To limit the pro-fibrotic, myofibroblast phenotype in the cultured fibroblasts, we only used the P1 and P2 cells in all the experiments, as it has been shown that fibroblast-to-myofibroblast conversion occurs with cell passage [12]. In certain experiments to compare the potential difference both WT and mutant cells, both cells were isolated in parallel from different mice, cultured, passaged and treated simultaneously at the same passages (P1 or P2). Since the difference between WT and WWP2^{Mut/Mut} cells was tested under same conditions, the comparison between WT and WWP2^{Mut/Mut} cells, showed that mutant WWP2 resulted in a reduced development of the pro-fibrotic phenotype, suggesting that WWP2 regulate cardiac fibrosis. This point has been discussed in the revision of the manuscript (**Page 16, Line 531-537**).

Minor weaknesses/corrections

- **Figure 4 is very difficult to follow—perhaps re-organizing the figure or removing some of the diagrams/schematics to make it less dense? In its current form, it is difficult to identify individual components in the figure.**

We are sorry for this and have now reorganized Figure 4 in the revised version to show the cellular localization of WWP2 in heart (**Fig. 4a-b**), WWP2 dysfunction in TGF β 1-stimulated cardiac fibroblasts show decreased fibrogenic response (**Fig. 4c-f**), the protein expression levels of WWP2 isoforms in TGF β 1-stimulated cardiac fibroblasts (**Fig. 4g**), siRNA (**Fig. 4h-k**) and rescuing (**Fig. 4l-n**) experiments showing the fibrogenic effect of WWP2 isoforms that contain the N-terminal region. We hope that now the individual components of the figure are easier to identify and follow.

- **Line 535: It should say “ARKADIA” to correct “AIKADIA”**

Thanks for spotting this type, which has been corrected. (**Line 566**)

Questions of General Interest

- **Is WWP2 affected by HECT E3 ubiquitin ligase inhibitors, such as Heclin?**

We did not test this experimentally, but after reviewing the literature we found that Heclin can inhibit WWP2 [17]. Interestingly, Heclin allowed clear distinction between RING and HECT-mediated ubiquitination, but it did not seem to have any selectivity for different HECT E3 ligases. Although Heclin was identified from a bicyclic peptide library screening targeting bicycle–Smurf2 interaction, it could inhibit Smurf2, Nedd4, WWP1 and WWP2 [17]. Another study showed that all inhibitors tested demonstrated some degree of inhibition of all three E3 ubiquitin ligases, Nedd4, WWP2 and WWP1 [5].

Reviewer #4 (Remarks to the Author):

Chen et al reported here that they have identified a cross-species (human and mouse) profibrotic extracellular matrix (ECM) co-expression gene network in diseased human hearts and mapped a common regulatory cis-element of this network to a gene encoding WWP2, a E3 ubiquitin ligase, particularly to its N-terminal isoform. They went on to show that mice mutant for WWP2 had improved cardiac function and reduced myocardial fibrosis in response to pressure overload, and attributed this regulation to WWP2N becoming nuclear bound under the influence of TGF- β , where it interacts with SMAD2 and promotes SMAD2 mono-ubiquitination.

Overall I find this manuscript very difficult to follow, not because I am not an expert in systems biology, many of their conclusions are not supported experimentally, and some of their experimental methods/designs are outright flawed. Although they have presented an impressive body of experimental data, I simply don't see a coherent story or cannot be enthusiastic in supporting its acceptance for publication.

We address each point and criticism raised by the Reviewer as detailed below.

Specific issues.

1. Based on analyses of RNA-seq data, the authors reported only very small increases of WWP2 expression as low as 1.02 fold in the diseased hearts. Can this be independently verified by qRT-PCR on available samples? I understand the statistical power applied to large data set, but any outcome conclusion has to withstand scrutiny of an independent method.

We believe that qRT-PCR analysis is valuable and cost-effective to analyze the expression of few genes with a known sequence. However, we and others [18] deem RNA-seq the preferred gold-standard method for genome-wide gene expression quantification because of its high reproducibility, it does not rely on internal control/genes for normalization, the large dynamic range and the requirement of less sample RNA.

As rightly mentioned by the Reviewer, we have high statistical power in our RNA-seq analysis in human heart due to the large sample size (126 DCM vs 92 control hearts) and therefore we consider our detection of small fold changes (FC), as in the case of *WWP2*, to be robust. However, to check if sample size and method can influence our *WWP2* FC quantification (reported in our manuscript, *WWP2* fold change (FC) = 1.02), we retrieved a separate (publicly available) heart gene expression data, where gene expression was quantified in subendocardial left ventricular tissue samples from 7 patients with dilated cardiomyopathy (DCM) and in those from 5 patients with non-failing (NF) hearts by an independent method, namely chip microarray analysis [19]. In this considerably smaller sample where gene expression levels were quantified by an independent method, we also found *WWP2* being marginally (and not significantly) upregulated in DCM (*WWP2* FC = 1.03, P-value = 0.111 by t-test). This microarray-based analysis shows a *WWP2* FC (by microarray) that is remarkably consistent with the *WWP2* FC reported in our DCM cohort (by RNA-seq). In addition, we highlight that in our manuscript we have reported three independent quantifications of *WWP2* overexpression in separate disease contexts (in human DCM, in human rTOF and in mouse heart failure, respectively), which showed similar and consistent results.

2. Some of their analyses are very difficult to evaluate, e.g. the authors concluded that only WWP2N was regulated by SNP associated with that gene, but their PCR design for measuring the individual isoforms are flawed. First of all, those "P1, P2, P3" designation in Fig.3a should really refer to PCR products, not the primers, as so stated in the text, which is very confusing and wrong. The drawing in the figure lacks sufficient detail to show how these PCR products could be exon-specific, therefore isoform specific. As they are, P1 is to both N and full length forms, P2 is full length only, whereas P3 can be both full length and C isoform.

We thank the reviewer for raising this point. The original **Fig. 3a** showed a schematic diagram based on the gene sequence. We appreciate this might lead to some confusion. We apologize if the schematic and designation used in **Fig. 3a** were not clear and might have led the Reviewer to conclude that the "PCR design for measuring the individual isoforms are flawed".

In the revised manuscript, we have redrawn a schematic based on the opening read frame (ORF) sequence of the different isoforms. We also clarified that the "P1, P2, P3" designation referred to the PCR products generated from different primer pairs (as indicated in the following figure). It is not possible to design primers that are specific to each individual isoform, as the *WWP2*-FL spans the entire ORF and overlaps with *WWP2*-N or *WWP2*-C.

By comparative analysis of the PCR products P1-P2-P3 we can assess the relative expression of isoform combinations. P1 PCR production referred to the expression of *WWP2*-N and *WWP2*-FL together, P2 PCR production referred to the *WWP2*-FL only and P3 PCR production referred to the expression of *WWP2*-FL

and WWP2-C together. We believe the new schematic in **Fig. 3a** - that is based on the ORF sequence of the different isoforms – clarifies this point; we also amended the text in the revised manuscript accordingly.

(New) Figure 3a. Schematic representation of the open reading frame (ORF) of *Wwp2* isoforms (*Wwp2*-FL, *Wwp2*-N and *Wwp2*-C). The position of the 4bp deletion introduced in the *Wwp2*^{Mut/Mut} mouse is shown as well as the position of the primer pairs (orange triangles) and PCR products used for the qPCR analysis of the *Wwp2* isoforms (i.e. P1 tags both *Wwp2*-N and *Wwp2*-FL, P2 tags *Wwp2*-FL only and P3 tags both *Wwp2*-C and *Wwp2*-FL).

3. The 4 bp deletion introduced by CRISPR/Cas9 is expected to cause frameshift mutation that disrupt the expression of full length as well as the N isoforms, the same consequence as in the “global” null mutation reported by others previously. In both cases, the “C” isoform could have been preserved. As such it is not surprising that the new mutants essentially phenocopied the old ones. However, I’d expect the absence of both the full length and the N-isoform by WB, not a mere truncation that only slightly shortened the full length protein as reported in all WB figures. So, none of the WB results can be trusted!

The Reviewer refers to the WB where we compared WWP2 isoforms in WT and mutant (**Fig. 3e**), and which shows a lower band for WWP2-FL, WWP2-N, suggesting a lack of proper protein with the right molecular weight. We reasoned two possible explanations for the presence of a secondary lower band in **Fig. 3e**: (1) the 4 bp deletion results in a truncated protein. This is what we *initially* proposed in our manuscript without a direct evidence or (2) the 4 bp deletion results in the absence of both the full length and the N-isoform (shown by the lack of bands at the right molecular weight, i.e., WWP2-FL at ~110 kD, WWP2-N at ~50 kD), despite the fact that our antibody (from Santa Cruz) detected a lower band.

We followed the reviewer’s suggestion “*the absence of both the full length and the N-isoform*” and we investigated this second hypothesis. To this aim, we acquired a different WWP2 Ab (from Bethyl Lab.), which indeed confirmed the presence of a lower band in a WWP2 KO model (see *Figure (left)*, below [20]). When this Ab was tested on our mutant mice cells, this showed the presence of a lower band (see *Figure (right)*, below and **Supplementary Figure 2e**). Therefore, we are not the only ones to report an additional band in WWP2 KO cells. Since the presence of a lower band in WWP2 KO mice has been previously reported and replicated in our lab (see *Figure below*), we now reason that we have generated a full WWP2 KO, rather a mouse model with truncated versions of both WWP2-FL and WWP2-N proteins, as initially hypothesized.

We have therefore amended the manuscript text accordingly and now indicate “lack of WWP2-FL and WWP2-N proteins” in our mouse model. We believe this investigation of WWP2 antibodies addresses the Reviewer point on the nature of our WWP2 mutant mice, which, as also acknowledged by the Reviewer, is a phenocopy of previously published WWP2 KO models [21].

We thank the reviewer for urging us to elucidate this point.

Figure legend: *Left*, Western blot analysis from website showing lack of proper WWP2-FL but aberrant short-band in WT, Knockdown (KD) & Knockout (KO) SHSY5Y by Biocompare.com. *Right*, Western blot analysis showing lack of proper WWP2-FL instead detection of an aberrant short-band in cardiac fibroblasts from *Wwp2*^{mut/mut} mice. This

experiment was carried out using two different anti-WWP2 antibodies bought from two companies. The arrow indicates the expected right-weight band for the WWP2-FL protein (~110 kD).

4. Related to above, the immunofluorescence results are also questionable.

The issue of the WWP2 Ab has been addressed above and we believe our immunofluorescence data are solid. For the expression and localization of WWP2 in heart section with immunofluorescence staining, we used two different antibodies: anti-WWP2 targeting N-terminal (Santa Cruz biotechnology, sc-30052) and anti-WWP2 targeting C-terminal (Aviva Systems Biology, #ARP43089_P050). As indicated in the datasheet, the former is specific for an epitope mapping between amino acids 30-57 near the N-terminus of WWP2 and the latter immunogen is a synthetic peptide directed towards the C terminal region of WWP2. As expected, we confirmed that the former antibody could detect WWP2-FL and WWP2-N isoforms and the latter could detect WWP2-FL and WWP2-C isoforms by WB. The immunofluorescence results with both antibodies showed similar upregulated WWP2 in heart after AngII-infusion, which was detected in non-myocytes (Figure 4a and Supplementary Fig. S6a).

In addition, the immunofluorescence staining showed that the induced WWP2 was localized predominantly in the nucleus upon TGF β 1 simulation with antibody for WWP2 (Fig. 5a). The nuclear localization of WWP2 was confirmed with immunofluorescence staining for the exogenously expressed WWP2-FLAG, which was detected with the antibody targeting FLAG (Supplementary Fig. S11a).

5. The authors designed siRNAs against 5' or 3' end regions of WWP2 and claim that these siRNAs decreased FL/N or FL/C isoforms. As pointed above, the WB analysis cannot be trusted, but this design is also flawed.

Multiple siRNA experiments have been carried out in murine cardiac fibroblasts as well as in human cardiac fibroblasts, which provided consistent results. The Reviewer's concern about our WB analysis has been addressed earlier. To clarify the point of the siRNA experiment, we now provide additional details on the design of the two siRNAs. The siRNA is a synthetic RNA duplex designed to specifically target a particular mRNA for degradation. As showed in the following figure (Supplementary Fig. S10a), the two siRNAs were designed to target the 5'-terminal (SiRNA-Wwp2-N') and 3'-terminal (SiRNA-Wwp2-C') of *Wwp2* mRNA, respectively. In line with the experimental design, the siRNA-Wwp2-N' led to the knockdown of Wwp2-FL and Wwp2-N, while the siRNA-Wwp2-C' led to the knockdown of WWP2-FL and WWP2-C (Fig. 4h-i).

We further highlight that the results of the siRNA experiments in mouse WT cells are consistent with the results of independent siRNA experiments in human cardiac cells with SiRNA-WWP2-N' (Fig. 4j, only target 5'-terminal of WWP2 mRNA and knockdown the FL/N isoforms). These results are also consistent with the data generated in primary cardiac fibroblasts from our mutant mouse model, which lacks WWP2-FL and WWP2-N (Fig. 4c-d). Both *in vivo* and *in vitro* (mouse and human) results therefore consistently showed that decreased expression WWP2 isoforms containing the N-terminal region results in decreased expression of fibrotic markers.

Therefore, we believe that multiple and independent experiments support the validity of our siRNA experimental design and the findings presented in these siRNA experiments.

(New) Supplementary Figure S10a. Schematic representation of siRNAs (SiRNA-Wwp2-N' and SiRNA-Wwp2-C') targeting different parts of *Wwp2* gene. Position of the three primer pairs (P1, P2 and P3, tagging *Wwp2*-N/FL, *Wwp2*-FL and *Wwp2*-C/FL, respectively), used for the quantification of *Wwp2* expression levels are also labeled.

6. Since all experimental manipulations affect both N and full length WWP2, no demonstration was made to verify that marginal increase of WWP2-N expression could cause any phenotype in either cell culture or animal experiments.

In our study we generated a new WWP2 mutant mice (WWP2^{Mut/Mut}) using CRISPR/Cas9 and carried out knockdown WWP2 expression by siRNA (see above). Technically, we could not manipulate WWP2-N without affecting WWP2-FL, and as WWP2-N overlaps with WWP2-FL in the DNA and mRNA sequence. The DNA deletion or sequence-based siRNA experiment also affected WWP2-N and WWP2-FL at the same time.

However, not all experimental manipulations affect both N and full length WWP2 isoforms. We carried out experiments in which cardiac fibroblasts from WWP2^{Mut/Mut} were *separately* transfected in two independent

experiments with WWP2-FL or WWP2-N plasmid expression (**Fig. 4l-m**, revised manuscript). We apologize if this was not clearly indicated in our manuscript. Notably, consistently with exogenous WWP2-FL, exogenous WWP2-N can also enhance (moderately but significantly) the fibrotic response (*Acat2*, *Col1a1*) in *WWP2^{Mut/Mut}* cells upon TGF β 1 stimulation. These data show that re-establishing WWP2-N or WWP2-FL expression in cardiac fibroblasts from our KO mouse (which lacks both WWP2-FL/N) is sufficient to induce expression of fibrotic markers upon TGF β 1 stimulation (data reported in Fig. 4m-n).

In the revised manuscript, we now clearly state that we transfected individually either the WWP2-N or WWP2-FL expression in cardiac fibroblasts.

7. Fig. 5e, I could not see any difference in the levels of SMAD2 monoubiquitination between WT and Mut/Mut cells based on the blot the authors provided.

We thank the reviewer for raising this point. We agree that the SMAD2 monoubiquitination band in cardiac fibroblasts was not clear enough in the previous Western Blot. We repeated this WB several times and we always found a strong lower band in all blots. We proposed that this lower band could be the heavy chain of rabbit IgG (as showed in the following figure, *left*). We initially used the normal HRP conjugated anti-rabbit IgG secondary antibody (Bethyl laboratory, A120-101P), as the antibody for immunoprecipitation (IP) is rabbit and Smad2 (Cell signalling technology, #3102). As the band was close to the expected SMAD2 monoubiquitination, it was difficult to get a clear blot at the expected region. Although the SMAD2 monoubiquitination was stronger in WT cardiac fibroblasts than in mutant cells at the same condition, we acknowledge that the originally presented monoubiquitination band in WT is weak.

In order to improve the blot, we used a new secondary antibody (Thermo Fisher, # 101023), which is the HRP conjugated anti-rabbit IgA, and should not detect the IgG heavy chain. The new Western Blot after IP with Smad2 is clearer and shows ubiquitination bands in WT cardiac fibroblasts (with *), including monoubiquitination (~68 kD, closely lower to marker for 72 kD), diubiquitination (~76 kD, closely higher to the marker for 72 Kd) and polyubiquitination of Smad2. Nonetheless, based on these blots we cannot state the complete absence of SMAD2 monoubiquitination band in *WWP2^{Mut/Mut}* cells, which might due to compensatory mechanisms of ubiquitination.

We thank the reviewer for urging us to elucidate this point. In the revised manuscript we present a much clearer blot showing the monoubiquitination band in WT and a much weaker band in *WWP2^{Mut/Mut}* cells (**Fig. 5d** of revised manuscript).

Figure legend: In-cell ubiquitylation of SMAD2 in fibroblasts from both *WWP2^{Mut/Mut}* and WT cells. Cells were treated with MG132 (10 μ M, 3hr) followed by TGF β 1 (5ng/ml, 6 hr). Lysates were prepared from *WWP2^{Mut/Mut}* and WT fibroblasts and then were subject to immunoprecipitation with anti-SMAD2 antibodies, followed by western blotting probed with antibodies as indicated. (a, b) the representative WB image showing the ubiquitinated SMAD2 with primary antibody anti-ubiquitin and the 1st secondary antibody anti-rabbit IgG (Cell signaling technology, #3102). (c) the representative WB image shows the ubiquitinated SMAD2 with primary antibody anti-ubiquitin and the 2nd secondary antibody anti-rabbit IgG (Thermo Fisher, # 101023). (d) Representative WB image showing the detection of SMAD2 in both *WWP2^{Mut/Mut}* and WT fibroblasts.

8. Authors showed in Fig. 5g and Fig. 5h that SMAD2 nuclear accumulation was enhanced or cytoplasmic export was delayed, (again I did not see any meaningful changes in SMAD4 nuclear accumulation in Fig. 5g). However, the transcriptional reporter assay indicated that SMAD transcriptional activity was actually decreased, how could this be?

We first showed that SAMD2 is one substrate that interacts with WWP2-FL/N (**Supplementary Fig. s11d and s11e**). Given that WWP2 is a E3 ubiquitin ligase, we reasoned that the binding of WWP2 to SMAD2 would lead to SMAD2 protein degradation. However, upon TGF β 1 stimulation, the levels of SMAD2 were

similar between WT and WWP2^{Mut/Mut} cardiac fibroblasts (**Fig. 5c**, revised manuscript). This suggested that WWP2 did not regulate SMAD signaling via protein degradation.

Poly-ubiquitination leads to protein degradation in the cytoplasm, in contrast mono-ubiquitination has often been reported to regulate protein location and activity [22]. We further showed that TGFβ1 stimulation (<16h) led to monoubiquitination of SMAD2 in WT cardiac fibroblasts, and this effect was much weaker in WWP2^{Mut/Mut} cells. Moreover, transcriptional reporter assay showed that WWP2 regulates the transcriptional activation of SMAD2. We reasoned that WWP2 interacts with SMAD2 to facilitate its monoubiquitylation, a post-translational modification important for the optimal function of SMAD2 in the nucleus. Similar to what has been previously shown for the regulation of Goosecoid (Gsc) (a paired-like homeobox transcription factor that has an important role in craniofacial development) by WWP2 [21].

Moreover, monoubiquitination has been reported to be essential for intrinsic nuclear import and function of the transcriptional factor [23]. We therefore show how TGFβ1 leads to increased nuclear retention of SMAD2, a transcriptional factor involved in TGFβ1 signaling, in cardiac fibroblasts. It was observed a similar import of SMAD2 in nucleus and delayed exportation of SMAD2 out of nucleus (shown in **Fig. 4g**). The distribution of SMAD4 similar in WT and WWP2^{Mut/Mut} fibroblasts, as rightly highlighted by the Reviewer (and we have amended the text accordingly, **Line 461-463**).

Considering the reduced transcriptional activity of SMAD2 in WWP2^{Mut/Mut} cardiac fibroblasts, we proposed that the retention of SMAD2 in the nucleus hampered the transcriptional activity of this protein. Inman GJ *et al.* reported that the transcriptional activity and SMAD2 levels in nucleus were not simultaneous [24]. The transcriptional activity of SMAD2 began to increase after the nuclear p-SMAD2 began to reduce upon TGFβ1 stimulation. It suggested that cytoplasm and nucleus continuous recycling of Smads is required after TGFβ1 stimulation, to maintain active SMAD complexes in the nucleus.

In summary, collectively our results suggest a mechanism of degradation-independent regulation of SMAD2 activity downstream of TGFβ-signaling activation that is modulated by WWP2. In the revised manuscript **Discussion**, we provided explanations (and literature) on the possible mechanisms on how E3 ligase-mediated monoubiquitination can affect signaling and TF activity regulation (**Line 546-551**). We also stated that further studies are required to prove whether monoubiquitination of SMAD2 by WWP2 disrupts (or directly regulates) the activity SMAD complex in the nucleus, or to prove that this SMAD2 monoubiquitination affects the nuclear export of SMAD2 (**Line 556-559**), as documented for other E3 ubiquitin-protein ligases [25].

Is it possible due to the fact that WWP2 actually affect SMAD7 in these conditions? Regulation of SMAD7 by WWP2 was reported previously. The authors should look into this possibility in their manuscript.

We thank the reviewer for raising this point. Yes, it has been previously reported that SMAD7 is a preferred substrate for WWP2-FL and WWP2-C following TGFβ stimulation [26]. Following this suggestion, we tested the interaction of WWP2 isoforms with SMAD7 in NIH-3T3 mouse embryonic fibroblast cells, and confirmed that Flag-tagged WWP2-FL and WWP2-C co-immunoprecipitated with SMAD7 protein, but WWP2-N showed a weaker binding to SMAD7 (**Supplementary Fig. 11d**, below). We then tested whether SMAD7 was differently binding to WWP2 isoforms in WT and WWP2^{Mut/Mut} cardiac fibroblasts and showed that WWP2-C similarly binds to SMAD7 in WT and WWP2^{Mut/Mut} cardiac fibroblasts (**Supplementary Fig. 11g**, below). Thus, SMAD7 mainly interacted with the C-terminal region of WWP2, in keeping with reported studies [26]. Moreover, our previous genetic analysis in human DCM hearts and experiments *in vivo* and *in vitro* have showed that WWP2 isoforms containing N-terminal region were the positive regulators for fibrogenesis in heart. We, therefore, did not focus on the possible mechanism of regulation of WWP2 on SMAD7 in cardiac fibrosis in the first place. In addition, the expression levels of SMAD7 were very similar with or without TGFβ1 stimulation in both WT and WWP2^{Mut/Mut} cardiac fibroblasts (**Supplementary Fig. 11h**). These new data regarding SMAD7 are included in the revised manuscript.

On the other hand, we confirmed that WWP2 interacted with SMAD2 with its N-terminal region, which was consistent with the published findings [26]. Considering the regulation of WWP2 isoforms with N-terminal regions, we proposed a mechanism of degradation-independent regulation of SMAD2 activity downstream of TGFβ-signaling activation that is modulated by WWP2.

(New) Supplementary Figure S11. (d) NIH-3T3 cells were transfected with WWP2-Flag isoforms and co-immunoprecipitation experiment shows a direct interaction of SMAD2 or SMAD7 with WWP2 isoforms. Lysates were subject to immunoprecipitation with anti-FLAG antibodies, followed by western blotting probed with antibodies as indicated. (g) Co-immunoprecipitation experiment shows a direct interaction between SMAD7 and WWP2-C isoforms in both WT and *WWP2*^{Mut/Mut} fibroblasts. Lysates prepared from fibroblast from both WT and *WWP2*^{Mut/Mut} were subject to immunoprecipitation with anti-SMAD7 antibodies, followed by western blotting probed with antibodies as indicated. (h) Representative Western blot analysis of SMAD7 protein levels in WT and *WWP2*^{Mut/Mut} fibroblasts with/without TGFβ1.

9. In theory, mono-ubiquitination would disrupt SMAD2/3 interaction with SMAD4, or disrupt SMAD3 binding to DNA, thus hampering the transcriptional activities of SMADs. As such, WWP2 via its ubiquitin E3 ligase activity should negatively regulate TGF- β signaling.

The Reviewer is right in his/her comment. In light of our data showing that SMAD2 and p-SMAD2 levels were similar in WT and *WWP2*^{Mut/Mut} cardiac fibroblasts, we reasoned that SMAD2/3 degradation was not the primary mechanisms at play here. Our findings further showed that physiological monoubiquitination of SMAD is associated with the regulation of TGFβ1-signaling in cardiac fibroblasts and this is modulated by WWP2.

The regulatory consequences of E3 ligase-mediated monoubiquitination are complex and context specific. Monoubiquitination can regulate protein location, activity, and protein interactions with binding partners [22]. Monoubiquitination was also shown to be required for the activity and the intrinsic nuclear import of target transcription factor(s) [21, 23]; while in other instances monoubiquitination has been reported to disrupt specific TFs interactions and their transcriptional activity [27].

In addition, monoubiquitination can regulate nuclear accumulation and the nucleocytoplasmic shuttling of SMAD complexes [24, 28]. In our study, we focused on how WWP2 regulates the import and export of SMAD2 in cardiac fibroblasts. Upon TGFβ1 stimulation, SMAD2 showed nuclear import in both WT and *WWP2*^{Mut/Mut} cardiac fibroblasts. We then used SB431542, a selective inhibitor of TGFβ superfamily type I activin receptor-like kinase (ALK) receptors [29], to study the differential nuclear export and nucleocytoplasmic shuttling of SMAD2 [30]. In support of WWP2 regulating nuclear accumulation and the nucleocytoplasmic shuttling of SMAD, we found a clear nuclear retention of SMAD2 in *WWP2*^{Mut/Mut} cells, but not for SMAD4 (Fig. 5g). The nuclear import and export (nucleocytoplasmic shuttling) of SMADS (2/3) is required to maintain active SMAD complexes in the nucleus, and SMAD4 is not necessarily required for the exit of SMAD2/3 from the nucleus after receptor inhibition [24]. In addition, nucleocytoplasmic shuttling is

crucial for transduction of TGF β -superfamily signals [24, 31], and therefore we believe our findings of WWP2 involvement in this process are important in the context of TGF β signaling. However, further studies will be required to prove whether monoubiquitination of SMAD2 by WWP2 disrupts and/or directly regulates the activity SMAD complex in the nucleus [24, 32] or to demonstrate that this monoubiquitination directly affects the nuclear export of SMAD2, as shown for other E3 ubiquitin-protein ligases [25].

Moreover, WWP2N lacks the HECT domain, therefore the ligase activity. If the regulation is mediated by the N-isoform, as the authors implied, it cannot be done through a direct ubiquitin modification. As is, this manuscript is not clear at all on the nature of N isoform function.

Yes, the WWP2-N contains the C2/WW domains which are considered to recognize and bind substrate proteins [33], but the WWP2-N lacks the HECT domain, i.e., the ligase activity. However, Soond *et al.* showed that individual WWP2 isoforms, and particularly the WWP2-N isoform, could play key roles in aberrant TGF β -dependent signaling function in cancer [26][34]. Indeed, it has been demonstrated that WWP2-N can interact with WWP2-FL, this way regulate the autoubiquitination of WWP2-FL, and in turn regulate the ubiquitination of SMADs [26], which is mediated by the WWP2-WW domain.

More generally, in other E3 ubiquitination ligases it has been shown that the C2/WW domains can interact directly with the HECT domain either intra- or intermolecularly [35] and WW domains provide a platform for the assembly of multi-protein networks and complexes [36]. So, previous data suggest that individual E3 ligase gene isoforms containing C2/WW domains – such as the WWP2-N isoform [26] – can have functional consequences (including mediating ligase activity) by direct interaction with the HECT domain.

Our data show that modification of individual WWP2 isoforms containing C2/WW domains, i.e., the WWP2-FL or WWP2-N isoforms, was sufficient to affect profibrotic gene expression downstream of TGF β -receptor activation. Our *in vivo* and *in vitro* data, siRNA and transfection of WWP2-N or WWP2-FL isoforms *individually* in cardiac fibroblasts from our KO mouse supported this hypothesis.

References

1. Heinig, M., et al., *A trans-acting locus regulates an anti-viral expression network and type 1 diabetes risk*. Nature, 2010. **467**(7314): p. 460-4.
2. Kang, H., et al., *Kcnn4 is a regulator of macrophage multinucleation in bone homeostasis and inflammatory disease*. Cell Rep, 2014. **8**(4): p. 1210-24.
3. Johnson, M.R., et al., *Systems genetics identifies Sestrin 3 as a regulator of a proconvulsant gene network in human epileptic hippocampus*. Nat Commun, 2015. **6**: p. 6031.
4. Bagnati, M., et al., *Systems genetics identifies a macrophage cholesterol network associated with physiological wound healing*. JCI Insight, 2019. **4**(2).
5. Watt, J.E., et al., *Discovery of Small Molecule WWP2 Ubiquitin Ligase Inhibitors*. Chemistry, 2018. **24**(67): p. 17677-17680.
6. Bottolo, L., et al., *Bayesian detection of expression quantitative trait loci hot spots*. Genetics, 2011. **189**(4): p. 1449-59.
7. Lewin, A., et al., *MT-HESS: an efficient Bayesian approach for simultaneous association detection in OMICS datasets, with application to eQTL mapping in multiple tissues*. Bioinformatics, 2016. **32**(4): p. 523-32.
8. Serini, G., et al., *The fibronectin domain ED-A is crucial for myofibroblastic phenotype induction by transforming growth factor-beta1*. J Cell Biol, 1998. **142**(3): p. 873-81.
9. Fu, X., et al., *Specialized fibroblast differentiated states underlie scar formation in the infarcted mouse heart*. J Clin Invest, 2018. **128**(5): p. 2127-2143.
10. Norris, R.A., et al., *Periostin regulates collagen fibrillogenesis and the biomechanical properties of connective tissues*. J Cell Biochem, 2007. **101**(3): p. 695-711.
11. Zeisberg, E.M. and R. Kalluri, *Origins of cardiac fibroblasts*. Circ Res, 2010. **107**(11): p. 1304-12.
12. Santiago, J.J., et al., *Cardiac fibroblast to myofibroblast differentiation in vivo and in vitro: expression of focal adhesion components in neonatal and adult rat ventricular myofibroblasts*. Dev Dyn, 2010. **239**(6): p. 1573-84.

13. Chen, M.M., et al., *CTGF expression is induced by TGF- beta in cardiac fibroblasts and cardiac myocytes: a potential role in heart fibrosis*. J Mol Cell Cardiol, 2000. **32**(10): p. 1805-19.
14. Acharya, A., et al., *The bHLH transcription factor Tcf21 is required for lineage-specific EMT of cardiac fibroblast progenitors*. Development, 2012. **139**(12): p. 2139-49.
15. Furtado, M.B., et al., *View from the heart: cardiac fibroblasts in development, scarring and regeneration*. Development, 2016. **143**(3): p. 387-97.
16. Schafer, S., et al., *IL-11 is a crucial determinant of cardiovascular fibrosis*. Nature, 2017. **552**(7683): p. 110-115.
17. Mund, T., et al., *Peptide and small molecule inhibitors of HECT-type ubiquitin ligases*. Proc Natl Acad Sci U S A, 2014. **111**(47): p. 16736-41.
18. Everaert, C., et al., *Benchmarking of RNA-sequencing analysis workflows using whole-transcriptome RT-qPCR expression data*. Sci Rep, 2017. **7**(1): p. 1559.
19. Barth, A.S., et al., *Identification of a common gene expression signature in dilated cardiomyopathy across independent microarray studies*. J Am Coll Cardiol, 2006. **48**(8): p. 1610-7.
20. <https://www.biocompare.com/Product-Reviews/350601-WWP2-polyclonal-antibody-for-Western-Blot/>
21. Zou, W., et al., *The E3 ubiquitin ligase Wwp2 regulates craniofacial development through mono-ubiquitylation of Goosecoid*. Nat Cell Biol, 2011. **13**(1): p. 59-65.
22. Schnell, J.D. and L. Hicke, *Non-traditional functions of ubiquitin and ubiquitin-binding proteins*. J Biol Chem, 2003. **278**(38): p. 35857-60.
23. Trotman, L.C., et al., *Ubiquitination regulates PTEN nuclear import and tumor suppression*. Cell, 2007. **128**(1): p. 141-56.
24. Inman, G.J., F.J. Nicolas, and C.S. Hill, *Nucleocytoplasmic shuttling of Smads 2, 3, and 4 permits sensing of TGF-beta receptor activity*. Mol Cell, 2002. **10**(2): p. 283-94.
25. Brooks, C.L., M. Li, and W. Gu, *Mechanistic studies of MDM2-mediated ubiquitination in p53 regulation*. J Biol Chem, 2007. **282**(31): p. 22804-15.
26. Soond, S.M. and A. Chantry, *Selective targeting of activating and inhibitory Smads by distinct WWP2 ubiquitin ligase isoforms differentially modulates TGFbeta signalling and EMT*. Oncogene, 2011. **30**(21): p. 2451-62.
27. Inui, M., et al., *USP15 is a deubiquitylating enzyme for receptor-activated SMADs*. Nat Cell Biol, 2011. **13**(11): p. 1368-75.
28. Tang, L.Y., et al., *Ablation of Smurf2 reveals an inhibition in TGF-beta signalling through multiple mono-ubiquitination of Smad3*. EMBO J, 2011. **30**(23): p. 4777-89.
29. Inman, G.J., et al., *SB-431542 is a potent and specific inhibitor of transforming growth factor-beta superfamily type I activin receptor-like kinase (ALK) receptors ALK4, ALK5, and ALK7*. Mol Pharmacol, 2002. **62**(1): p. 65-74.
30. Schmierer, B. and C.S. Hill, *Kinetic analysis of Smad nucleocytoplasmic shuttling reveals a mechanism for transforming growth factor beta-dependent nuclear accumulation of Smads*. Mol Cell Biol, 2005. **25**(22): p. 9845-58.
31. Hill, C.S., *Nucleocytoplasmic shuttling of Smad proteins*. Cell Res, 2009. **19**(1): p. 36-46.
32. Tang, L.Y. and Y.E. Zhang, *Non-degradative ubiquitination in Smad-dependent TGF-beta signaling*. Cell Biosci, 2011. **1**(1): p. 43.
33. Gong, W., et al., *Structure of the HECT domain of human WWP2*. Acta Crystallogr F Struct Biol Commun, 2015. **71**(Pt 10): p. 1251-7.
34. Soond, S.M., et al., *Novel WWP2 ubiquitin ligase isoforms as potential prognostic markers and molecular targets in cancer*. Biochim Biophys Acta, 2013. **1832**(12): p. 2127-35.
35. Yao, W., et al., *WW domain-mediated regulation and activation of E3 ubiquitin ligase Suppressor of Deltex*. J Biol Chem, 2018. **293**(43): p. 16697-16708.
36. Ingham, R.J., et al., *WW domains provide a platform for the assembly of multiprotein networks*. Mol Cell Biol, 2005. **25**(16): p. 7092-106.

Reviewers' comments:

Reviewer #1 (Remarks to the Author):

All my concerns have been well addressed.

Reviewer #2 (Remarks to the Author):

No further comments.

Reviewer #3 (Remarks to the Author):

The paper is improved by the careful response to the points we raised. There are still a few minor problems to be considered, along with one major point of clarification.

1. Figure 4D legend; CTGF is misspelled.

2. Major concern - Figure 4D specifies FN1 for the fibronectin product. If this is a pan-specific gene product it is entirely possible that the experimental results under represent the actual change in ED-A isoform of Fn expression. In other words, the experiment should be designed to be specific for ED-A-Fn. This is important because fibronectin appears as up to 12 splice variants in plasma and tissues. The experiment needs to be redone to specify the correct isoform.

3. Major concern - The authors wrote: "In our experience, DMEM supplemented with 20% FBS is better to promote cell crawling compared with 10% FBS. But, we did not pay much attention to the elastic modulus of the culture surface as the elastic modulus of the culture surface seemed normal, so we did not indicate it." If they used plastic plates, the elastic modulus values will be 2000 - 5000 X what they are in a specially prepped 5 kPa plate. This always results in rapid and complete activation of fibroblasts to myofibroblasts. The text needs to be rewritten to identify this fact within the context of their results and their discussion. There is unlikely to be partial conversion under these conditions.

4. Major concern - The authors wrote: " To limit the pro-fibrotic, myofibroblast phenotype in the cultured fibroblasts, we only used the P1 and P2 cells in all the experiments, as it has been shown

that fibroblast-to-myofibroblast conversion occurs with cell passage [12]." P1 cells are passaged and P2 cells are again passaged. On plastic, this guarantees that most if not all of the fibroblasts are activated to myofibroblasts in the presence of high serum (10 - 20%). Again, the likelihood that all of the cells in this study are activated and phenoconverted myofibroblasts is given. These cells may slide from mature to super mature myofibroblasts, but they will all be myofibroblasts nonetheless. This is likely the actual case within the cells studied in this experiment.

Reviewer #4 (Remarks to the Author):

The authors addressed most of my issues except one. I didn't make my point clear about the siRNA experiment previously. The isoform-specific design would not work if the three different WWP2 isoforms are produced from a single pre-mRNA transcript as products of alternative splicing regulation, unless they are products of different RNA transcripts, even then WWP2N could still potentially be targeted by the so-called siWWP2C if the mRNA message does not terminate before the C-form portion. Please 1) specify if these 3 isoforms are produced from different transcripts in the revision and 2) show the actual WB in which designated isoforms were indeed specifically targeted.

Pont-by-point rebuttal

Reviewer #1 (Remarks to the Author):

All my concerns have been well addressed.

Reviewer #2 (Remarks to the Author):

No further comments.

Reviewer #3 (Remarks to the Author):

The paper is improved by the careful response to the points we raised. There are still a few minor problems to be considered, along with one major point of clarification.

1. Figure 4D legend; CTGF is misspelled.

Thanks for spotting this typo, which has been corrected.

2. Major concern - Figure 4D specifies FN1 for the fibronectin product. If this is a pan-specific gene product it is entirely possible that the experimental results under represent the actual change in ED-A isoform of Fn expression. In other words, the experiment should be designed to be specific for ED-A-Fn. This is important because fibronectin appears as up to 12 splice variants in plasma and tissues. The experiment needs to be redone to specify the correct isoform.

In keeping with the reviewer recommendation, we have reanalyzed the expression of specific fibronectin forms containing the EDA segment (the primers for EDA-FN used were also listed in the **Supplementary Table S5**). We now show that in primary (myo)fibroblasts, TGFβ1 stimulation increased EDA-FN production of mRNA (~2 folds), and this was significantly prevented in WWP2^{Mut/Mut} cells (P=0.008; see revised **Fig. 4d**).

3. Major concern - The authors wrote: "In our experience, DMEM supplemented with 20% FBS is better to promote cell crawling compared with 10% FBS. But, we did not pay much attention to the elastic modulus of the culture surface as the elastic modulus of the culture surface seemed normal, so we did not indicate it." If they used plastic plates, the elastic modulus values will be 2000 - 5000 X what they are in a specially prepped 5 kPa plate. This always results in rapid and complete activation of fibroblasts to myofibroblasts. The text needs to be rewritten to identify this fact within the context of their results and their discussion. There is unlikely to be partial conversion under these conditions.

As detailed by the reviewer, it is expected that most of the cultured fibroblasts would be activated to myofibroblasts in our experimental setup. In keeping with this, we also noticed a pro-fibrotic phenotype in the "fibroblasts" before TGFβ1 treatment. We acknowledge this point and to make it explicit to the reader, we have revised the manuscript text and, when appropriate, re-named the cultured cells used in our experiments as "(myo)fibroblasts". These changes have been applied throughout the whole text and detailed in the methods, and are highlighted using red font in the revised manuscript. We also amended the Discussion, where we now recognize this specific limitation (**Page 15, line 537-546**).

4. Major concern - The authors wrote: " To limit the pro-fibrotic, myofibroblast phenotype in the cultured fibroblasts, we only used the P1 and P2 cells in all the experiments, as it has been shown that fibroblast-to-myofibroblast conversion occurs with cell passage [12]." P1 cells are passaged and P2 cells are again passaged. On plastic, this guarantees that most if not all of the fibroblasts are activated to myofibroblasts in the presence of high serum (10 - 20%). Again, the likelihood that all of the cells in this study are activated and phenoconverted myofibroblasts is given. These cells may slide from mature to super mature myofibroblasts, but they will all be myofibroblasts nonetheless. This is likely the actual case within the cells studied in this experiment.

As suggested by the reviewer, under our culture conditions TGFβ1 stimulation would increase the maturation of the myofibroblasts to super mature myofibroblasts, and we also observed further induction of the expression of pro-fibrotic genes and synthesis of extracellular matrix, which was reduced in WWP2 loss-of-function cells. In the revised manuscript, **(1)** we now clearly acknowledge that the large majority the of cultured fibroblasts are myofibroblasts (see **Supplementary Information page 42-43**, description of **Cell Culture and treatment**) and therefore these cells are now referred to as (myo)fibroblasts (these changes applied throughout the text and figure legends), and **(2)** we recognize and discuss this specific limitation in the revised Discussion (**Page 15, line 537-546**).

Reviewer #4 (Remarks to the Author):

The authors addressed most of my issues except one. I didn't make my point clear about the siRNA experiment previously. The isoform-specific design would not work if the three different WWP2 isoforms are produced from a single pre-mRNA transcript as products of alternative splicing regulation, unless they are products of different RNA transcripts, even then WWP2N could still potentially be targeted by the so-called siWWP2C if the mRNA message does not terminate before the C-form portion. Please 1) specify if these 3 isoforms are produced from different transcripts in the revision and 2) show the actual WB in which designated isoforms were indeed specifically targeted.

We thank the reviewer for his positive comments on our revisions. We further clarify details on our siRNA experiment as follows. (1) It has been proposed that alternative promoters within the Wwp2 gene can give rise to different isoforms (see schematic Figure 1a, below) [1, 2]. In particular, the full length mRNA generated from promoter 1 (P1) produces Wwp2-FL transcript (NM_025830.4) and then WWP2-FL protein isoform. The N-terminal transcript (AK141281.1) is likely generated from the full length mRNA by failure to splice-out intron 9-10 and then WWP2-N protein isoform. Notably, it has been proposed that the C-terminal transcript (AK159248.1) is likely generated from a second internal promoter P2 within intron 10-11. We have revised the manuscript text and specified the origins of the three gene isoforms in keeping with the literature (Page 11, line 384-386).

Figure 1. Schematic representation of WWP2 gene locus and three WWP2 transcripts and protein isoforms. **a.** The position of the promoters on mouse chromosome 8. **b.** The *Wwp2-FL* and *Wwp2-N* are generated from a common promoter and *Wwp2-N* isoform presumably generated by failure to splice-out intron 9-10. **c.** *Wwp2-C* is likely to be generated from a second internal promoter P2 within intron 10-11, as previously suggested [1].

(2) We apologize if this was not clearly defined in the legend of Figure 4. The expression of each different WWP2 isoform targeted by the different siRNAs (SiRNA-Wwp2-N' and SiRNA-Wwp2-C') was reported in the WB shown in Fig. 4j. We have amended the legend of Fig. 4j to elucidate this.

Figure 2. Schematic representation of SiRNAs (SiRNA-Wwp2-N' and SiRNA-Wwp2-C') targeting different *Wwp2* transcripts (*left*) and the WBs for each targeted Wwp2 isoform (*right*), which are presented in full in Fig. 4j.

Here, to further clarify this point for the reviewer, we report the WBs (from Fig. 4j) showing the expression of each targeted isoform (**right**), alongside with a description and annotation of each siRNA used to target each isoform (**left**).

References

1. Soond, S.M. and A. Chantry, *Selective targeting of activating and inhibitory Smads by distinct WWP2 ubiquitin ligase isoforms differentially modulates TGFbeta signalling and EMT*. *Oncogene*, 2011. **30**(21): p. 2451-62.
2. Zou, W., R. Shao, and D. Jones, *Reply to 'Dissecting the role of miR-140 and its host gene'*. *Nat Cell Biol*, 2018. **20**(5): p. 519-520.

REVIEWERS' COMMENTS:

Reviewer #3 (Remarks to the Author):

No further comments for the authors, they have answered the concerns raised.

Reviewer #4 (Remarks to the Author):

Authors have addressed all my concerns. No further comments.